



# Islet: Interpolation semi-Lagrangian element-based transport

Andrew M. Bradley[1], Peter A. Bosler[1], and Oksana Guba[1]

[1]Sandia National Laboratories, Albuquerque, New Mexico

**Correspondence:** Andrew M. Bradley (ambradl@sandia.gov)

**Abstract.** Advection of trace species (tracers), also called tracer transport, in models of the atmosphere and other physical domains is an important and potentially computationally expensive part of a model's dynamical core (dycore). Semi-Lagrangian (SL) advection methods are efficient because they permit a time step much larger than the advective stability limit for explicit Eulerian methods. Thus, to reduce the computational expense of tracer transport, dycores often use SL methods to advect passive tracers. The class of interpolation semi-Lagrangian (ISL) methods contains potentially extremely efficient SL methods. We describe a set of ISL bases for element-based transport, such as for use with atmosphere models discretized using the spectral element (SE) method. An ISL method that uses the natural polynomial interpolant on Gauss-Legendre-Lobatto (GLL) SE nodes of degree at least three is unstable on the test problem of periodic translational flow on a uniform element grid. We derive new alternative bases of up to order of accuracy nine that are stable on this test problem; we call these the *Islet bases*. Then we describe an atmosphere tracer transport method, the *Islet method*, that uses three grids that share an element grid: a dynamics grid supporting, for example, the GLL basis of degree three; a physics grid with a configurable number of finite-volume subcells per element; and a tracer grid supporting use of our Islet bases, with particular basis again configurable. This method provides extremely accurate tracer transport and excellent diagnostic values in a number of validation problems. We conclude with performance results that use up to 27,600 NVIDIA V100 GPUs on the Summit supercomputer.

## 1 Introduction

Trace atmosphere species, or tracers, are an important part of atmosphere models for the study of climate. Thus, tracer transport is an important computational subcomponent of an atmosphere dynamical core (dycore). Because of the large number of tracers in climate models, tracer transport can be computationally very expensive. To address this cost, often semi-Lagrangian (SL) methods are used to carry out passive tracer transport. The Lagrangian form of the advective form of the sourceless tracer transport equation is $Dq/Dt = 0$, where $Dq/Dt \equiv \partial q/\partial t + \boldsymbol{u} \cdot \nabla q$, $q$ is a mixing ratio, and $\boldsymbol{u}$ is the flow velocity.

There are a number of types of SL methods; see, e.g., Giraldo et al. (2003); Natarajan and Jacobs (2020); Bradley et al. (2019) for a review of these. This article focuses on remap-form interpolation methods. An interpolation SL (ISL) method discretizes the tracer transport equation by $q(x_i, t_2) = q(x_i^*, t_1)$, where $x_i$ is an Eulerian *arrival* grid point and $x_i^*$ is its *departure* point at $t_1 < t_2$. $q$ has values only at nodal points $x_i$; thus, evaluating $q(x_i^*, t_1)$ requires interpolation. A method in *remap form* directly remaps the tracer field on the Eulerian grid at the previous time step to the Lagrangian grid, in contrast to flux-form methods that integrate over swept regions (Lauritzen et al., 2010; Lee et al., 2016) or characteristic curves (Erath and Nair, 2014). The





computational and communication costs of a remap-form method are very nearly independent of time step, whereas a flux-form method's costs grow roughly linearly with time step.

Interpolation is in contrast to *cell-integrated* methods; these latter integrate the basis of a target (e.g. Lagrangian) element

against those of the source. This integration entails substantially greater cost for two reasons: first, larger computational cost due to sphere-to-reference point calculation and interpolant evaluations at many quadrature points; second, larger communication volume because all data from the target element must be made available to a source element. However, in trade for these additional costs, flux-form and exact cell-integrated remap-form methods have the benefits of, first, local mass conservation and, second, stability obtained from the $L^2$ projection.

In this work, first, we give up local, but not global, mass conservation, and in trade seek maximum computational efficiency, where *computational efficiency* is the ratio of solution accuracy to computational or communication cost, each of these having multiple means of measurement. See Bradley et al. (2019, Sect. 1) for a more detailed discussion of these matters. Second, we seek ISL methods that satisfy a necessary condition for stability.

### 1.1 Motivation and applications

Our primary objective in developing efficient tracer transport methods is to provide one for the Dept. of Energy Exascale Earth System Model (E3SM) (E3SM Project, 2018) Atmosphere Model's (EAM) dycore, HOMME (Dennis et al., 2005, 2012). HOMME uses the spectral element (SE) method (SEM) to discretize the equations governing atmospheric flow. *Element-based* methods permit extremely flexible discretization of domains: for example, E3SM's regionally refined model (RRM) configurations (Tang et al., 2019), in which the atmosphere element grid is refined in a particular region of the earth. Our

methods are fully element based.

In addition to developing ISL methods that can share a spectral element grid, we develop methods to remap among multiple subelement grids. Our motivation is to enable extremely highly resolved tracer filamentary structure for a fixed dynamics resolution, to support the modeling of strongly tracer-dependent models such as those of aerosols. A recent report from the National Academies of Sciences, Engineering, and Medicine (Field et al., 2021) includes the recommendation to "explore

whether global aerosol optical depth (AOD) distribution is significantly affected by plume-scale effects" and asks: "Are nested grids needed to represent plume processes? What spatial resolution is needed to faithfully represent the radiative forcing and impact outcomes?"

Because the Islet method includes algorithms to remap among subelement grids to address this first motivation, it can in addition couple to element-based discretizations other than SE.

### 1.2 Related work


Baptista and Oliveira (Baptista, 1987; Oliveira and Baptista, 1995) analyzed the stability of a number of SL methods, including ISL. They differentiate between compact and non-compact interpolants. A compact interpolant is one whose support is only the nodes of an element; a non-compact one includes nodes in the interiors of adjacent elements in its support. In this article, we focus on compact interpolants. Let a *natural* interpolant be the unique polynomial interpolant that uses all available nodes in an





element as its support, and similarly for a natural basis. Our findings are consistent with theirs: an ISL method using a compact natural interpolant is unstable for elements having degree three and higher. The ISL method on spectral elements, in particular, has been explored in the past (Giraldo, 1998; Giraldo et al., 2003; Bochev et al., 2015), but these studies did not completely analyze the stability implications of using high-order bases. In Sect. 2, we study the stability of compact interpolants and devise new interpolation bases founded on this analysis.

Remapping data among multiple component grids is common in many applications of PDE-based modeling because it is a direct means to permit each component to run in its most efficient configuration. For example, most whole-earth, fully coupled earth system models use different grids for ocean, land, and atmosphere components. Our focus is on using multiple grids within a component to enhance efficiency of each subcomponent model. In atmosphere models, the term *physics parameterizations* is often used to refer to the collection of microphysics, macrophysics, chemistry, aerosols, radiation, and other sub-element-

grid parameterizations, also called subgrid closures (e.g. Stensrud, 2009). These act together as the physics parameterizations subcomponent, and the dycore is the other subcomponent of the atmosphere component. Recent examples in atmosphere models of remapping data among multiple subcomponent grids include separate physics parameterizations and dynamics grids in the atmosphere (Herrington et al., 2019; Hannah et al., 2021); adaptive mesh refinement (AMR) of a tracer (Chen et al., 2021; Semakin and Rastigejev, 2020); and local vertical refinement in physics parameterizations relative to the shared background

vertical grid, the Framework for Improvement by Vertical Enhancement (FIVE) (Yamaguchi et al., 2017; Lee et al., 2020). We use the ideas in Hannah et al. (2021) directly in this article. An important part of remapping fields between grids is conserving mass, not introducing new mixing ratio extrema, and – as a specific case of not introducing new extrema – maintaining a constant mixing ratio. We summarize these requirements as *property preservation*; sometimes *shape preservation* is used to refer to the properties other than mass conservation. The Islet method uses both local-element and global-grid versions of

algorithms described and analyzed in Bradley et al. (2019); see that article for a detailed discussion of the problem of property preservation.

This article focuses on the horizontal dimensions or, more generally, two-dimensional Lagrangian levels in a three-dimensional discretization. Future work using methods similar to those in Sect. 4 will address the vertical dimension. For example, it is necessary to increase simultaneously both the horizontal and vertical resolutions of a plume to capture and maintain its structure

(Eastham and Jacob, 2017).

### 1.3 Outline

The rest of this article is structured as follows. Section 2 describes a necessary condition for stability of an ISL transport method on a test problem. In Sect. 3, we combine this necessary condition with two quality heuristics and a search procedure to derive element basis sets, the *Islet bases*, for ISL element-based transport. In Sect. 4, we use these basis sets as part of the overall *Islet*

*method* to couple accurate tracer transport to a spectral element dynamical grid and a finite-volume physics parameterization grid. Section 5 describes numerical validation problems and presents results for the Islet method. Section 6 shows performance data on the GPU-powered supercomputer Summit. Finally, Sect. 7 concludes.





## 2 Stability

A necessary and sufficient condition for stability of a linear method is uniform power boundedness (Strikwerda, 2004): the norm
of the method's space-time operator raised to any integer power $n \geq 0$, and independent of (i.e. "uniform in") discretization
parameters, is bounded by a constant.

Consider the pure advection *test problem* of periodic uniform translation on a uniform element grid with no source terms. A
necessary condition for stability of a space-time operator for this problem is that the maximum amplitude of an eigenvalue of
the space-time operator is at most one. A discrete space-time operator for pure advection in general has at least one eigenvalue
of magnitude one.

For the class of discretizations we develop in this article, we do not attempt to prove satisfaction of this necessary condition
analytically. For doing so is equivalent to finding a bound of one, analytically, on the magnitudes of eigenvalues of an infinite
set of $d \times d$ matrices for $d$ as large as 12, where the matrices have at least one eigenvalue with magnitude one. We are not
aware of a means to do this. Thus, our approach is purely numerical: we describe a standard method to compute eigenvalues
numerically efficiently, and then we apply this method to numerically test methods at a large but finite number of points in
parameter space and with finite-precision results.

Let $\lambda_{\max}$ denote the maximum eigenvalue magnitude we find in our search. In infinite precision, the necessary condition
implies $\lambda_{\max} = 1$ for the test problem. In finite precision, we expect error in the computation and require $\lambda_{\max} \leq 1 + \varepsilon$, with
$\varepsilon$ a tolerance near machine precision. We say that a method is *test-problem stable* (t.p.s.) if this numerical bound holds for
the method applied to the test problem. For brevity, we also say that a basis is or is not t.p.s., by which we mean that the ISL
method that uses the basis is or is not t.p.s.

### 2.1 Instability in the classical cubic ISL method

We neither seek nor expect to find methods that satisfy the necessary condition in more general settings. For already classical
methods do not. In this subsection, we provide two examples in which the classical cubic ISL method is not stable. To find each
example, we construct a parameterized problem and discretization, then run a randomized search over parameter values to find
non-t.p.s. instances. Finally, to simplify reproducibility, we round the parameter values to not more than five digits while still
obtaining a maximum eigenvalue magnitude substantially greater than one.

First, consider the classical cubic ISL method on a nonuniform one-dimensional grid for the problem of periodic uniform
translational flow. Let the periodic grid have the five unique points $\{0, 0.11242, 0.44817, 0.78392, 0.88737, 1\}$, where 0 is
identified with 1 by periodicity. The cubic interpolant uses the four source points nearest a target departure point as its support.
Let the flow speed and time step be such that in each time step, the field translates by distance 0.33575. The space-time operator
for this configuration has maximum eigenvalue magnitude exceeding $1 + 10^{-3}$, violating the necessary condition for stability.

Second, consider again the classical cubic ISL method, but now on a doubly periodic, two-dimensional grid $[0,1]^2$, where
again 0 is identified with 1 by periodicity. The grid consists of uniformly sized rectangles. The flow velocity, while still
nondivergent and constant in time, is shear in space: $s \equiv 1 + \cos(2\pi(0.342 + x - y))$, $(u,v) = (s,s)$. With 15 unique grid





points in the $x$ direction and 13 in the $y$ and time step 0.2761, the space-time operator has maximum eigenvalue magnitude exceeding $1 + 10^{-2}$.

Figure 2, described in Sect. 3.7, plots $\log_{10}(\lambda_{\max} - 1)$ as a function of step size for a set of bases. We need to proceed further in this article before we can describe this figure in detail. However, one qualitative fact is apparent now. In one basis, with
corresponding line pattern red with small circles, $\lambda_{\max}$ is substantially above 1 for almost all step sizes. In the other two, green with $\times$ markers and black with large circle marker, $\lambda_{\max} - 1$ is at machine precision for almost all values, with one or a small number of spikes to larger values. We refer to this second pattern of $\lambda_{\max}$ values as *spiky*. In our two examples, the classical cubic ISL method's $\lambda_{\max}$ is spiky with respect to step size and other parameters. We speculate all high-order ISL methods have at least a spiky $\lambda_{\max}$ pattern when applied to some problems, and practical high-order ISL methods must have no worse
than a spiky pattern or else be unstable in practice. Sect. 5.1 compares the consequences of the two $\lambda_{\max}$ patterns on validation problems.

## 2.2 Notation and definitions

Vectors and matrices are written in boldface; continuum fields are not. Thus, $\boldsymbol{f}$ could be the nodal values that, combined with a basis, represent the continuum field $f$. Indexing starts at 0. To minimize nested subscripting, in some cases we denote entries of
a vector parenthetically, and we use colon notation to indicate an index list. For example, $\boldsymbol{x}(i)$ is element $i$ of $\boldsymbol{x}$, and $\boldsymbol{x}(i:i+2)$ produces the 3-element vector containing elements $i$ through $i+2$ inclusive. The list $n-1:-1:0$ decrements the index; thus, $\boldsymbol{x}(n-1:-1:0)$ is a vector with elements in the opposite order of those in $\boldsymbol{x}$. It is sometimes useful to annotate a matrix or vector with its size; for example, each of the matrix of zeros $\mathbf{0}^{m \times n}$ and the identity matrix $\mathbf{I}^{m \times n}$ has size $m \times n$, and $\boldsymbol{x}^n$ is an $n$-vector. Often in equations we annotate only the first instance of the vector or matrix to provide useful data while minimizing
clutter. Let $\boldsymbol{e}_i$ be column $i$ of the identity matrix.

We refer to various grids. An *element grid* is distinguished from the *point* or *subelement* grid. The point grid is derived from the element grid by populating each element with either subelement grid points associated with the method's basis or finite-volume subcells. When *grid* is used without qualification, we mean the point grid.

## 2.3 Maximum eigenvalue magnitude

The ISL space-time operator for an element-based method applied to the test problem has the following structure. Let a matrix $\mathbf{B}^{d \times (d+1)} \equiv \left( \bar{\mathbf{B}}^{d \times d} \quad \mathbf{b}^{d \times 1} \right)$. Let $\mathbf{C}_k$ be the permutation matrix that circularly shifts its left operand $k$ columns to the right and its right operand $k$ rows up. Let

$$\mathbf{A} \equiv \mathbf{C}_{-r'} \left[ \left( \mathbf{I}^{N \times N} \otimes \bar{\mathbf{B}} \right) + \left( \mathbf{I}^{N \times N} \otimes (\mathbf{b} \boldsymbol{e}_{d-1}^T) \right) \mathbf{C}_1 \right], \tag{1}$$

where $\otimes$ denotes the Kronecker product, $T$ the matrix transpose, and $N$ the number of elements.

This structure arises as follows. Consider a continuous discretization using a nodal $n_p$-basis, $n_p = d+1$, with $n_p$ the number of nodes. The grid has $N$ elements. Each row of the space-time matrix corresponds to a target node. The target node's departure point is in a source element, and the source element's $n_p$ nodes provide the interpolant's support for that target node. Because





the test problem has uniform flow and the grid is uniform, each source element contains $n_p - 1$ target node departure points. Thus, for each source element, there is a $d \times (d+1)$ block $\mathbf{B}$ in the matrix. Again because the test problem has uniform flow and the grid is uniform, this block is the same for each source element. Translation distance determines the number of rows $r'$ these blocks are circularly shifted. The final column of each block, $\boldsymbol{b}$, overlaps the first column of the next because elements share element-boundary nodes.

We seek the maximum eigenvalue magnitude of the matrix $\mathbf{A}$. Diagonalization of a matrix having this form can be understood in a number of ways: the discrete Fourier transform diagonalizes a circulant matrix, and block diagonalization of a block circulant matrix follows with some algebra (e.g. Vichnevetsky and Bowles, 1982; Idelsohn et al., 1995); by Bloch-wave decomposition (e.g. Cohen, 2001; Ainsworth, 2004); and by consideration of the problem's symmetries. The result is that the spectrum of $\mathbf{A}$ can be obtained by solving eigenvalue problems associated with $\mathbf{B}$. For fixed $N$, one must solve $N$ independent eigenvalue problems of size $d \times d$. Thus, they can be solved in parallel, and each has cost that is a function of just $d$ rather than $Nd$. Appendix A provides details.

## 2.4 Test-problem stability

The free parameter in the test problem is the scalar flow speed times time step, or simply *translation distance*. A block $\mathbf{B}$ corresponds to one particular flow speed. Let $\mathcal{N}$ be the collection of data describing the basis according to which $\mathbf{B}$ is formed. To approximate $\lambda_{\max}(\mathcal{N})$, the maximum eigenvalue magnitude corresponding to basis $\mathcal{N}$, we must discretize not only the search over values of number of elements $N$, but also the translation distance.

We can use two symmetries to reduce the translation distance set. First, on a uniform element grid having element size one, any translation $\Delta x + n$ for integer $n$ yields the same maximum eigenvalue magnitude as simply $\Delta x$. Integer increments to $\Delta x$ correspond to row shifts of the space-time operator, to which the maximum eigenvalue magnitude is invariant. Second, basis symmetry implies that only $\Delta x \in (0, 1/2]$ needs to be searched, for $\lambda_{\max}$ is the same for translations $\pm \Delta x$, and then it is the same for $-\Delta x + 1$ by the first point.

In summary, computing an approximation to $\lambda_{\max}(\mathcal{N})$ requires solving eigenvalue problems for a discrete subset of the two-dimensional space $\Delta x \in (0, 1/2]$, $\theta \in [0, 2\pi)$, where, as explained in Appendix A, $\theta$ is the parameter in the eigenvalue computation corresponding to $N$. If in the course of this search it is found that $\lambda_{\max}(\mathcal{N}) > 1 + \varepsilon$, with $\varepsilon$ a tolerance near machine precision, then $\mathcal{N}$ is determined not to be t.p.s.

If a basis is t.p.s. on the one-dimensional test problem and has tensor-product structure, then the same holds on the multidimensional test problem, too, because the multidimensional test problem decouples by dimension. Our interpolants have tensor-product structure.

## 3 The Islet bases

The natural basis at Gauss-Lobatto-Legendre (GLL) nodes – for brevity, subsequently the *natural GLL basis* – is not t.p.s. for $n_p \geq 4$, as we illustrate in Sects. 3.5 and 5.1. In this section, we design compact modified bases that are t.p.s.





### 3.1 Definitions

The polynomial interpolant through values $\boldsymbol{y}^n$ at points $\boldsymbol{x}^n$, evaluated at $x$, in Lagrange form is

$$\mathcal{L}(x; \boldsymbol{y}^n, \boldsymbol{x}^n) \equiv \sum_{i=0}^{n-1} \boldsymbol{y}(i) \prod_{j=0, j \neq i}^{n-1} \frac{x - \boldsymbol{x}^n(j)}{\boldsymbol{x}^n(i) - \boldsymbol{x}^n(j)}.$$

If the second argument to $\mathcal{L}$ is a matrix $\mathbf{Y}^{n \times k}$ rather than a vector $\boldsymbol{y}^n$, then the output is a vector whose element $j$ corresponds to column $j$ of $\mathbf{Y}^{n \times k}$. If the first argument is a vector $\boldsymbol{z}^m$, then the output is an $m \times k$ matrix whose $i$th row corresponds to $\boldsymbol{z}^m(i)$ and $j$th column corresponds to column $j$ of $\mathbf{Y}^{n \times k}$. $\mathcal{L}$ provides a basis for degree-$d$ polynomials. These are supported by $n = d + 1$ points, each an element in the $n$-vector $\boldsymbol{x}^n$. The natural basis functions associated with $\boldsymbol{x}^n$ are $\mathcal{L}(x; \mathbf{I}^{n \times n}, \boldsymbol{x}^n)$. In this work we take $\boldsymbol{x}$ to be points in the reference domain $[-1, 1]$. Often $\boldsymbol{x}^n = \boldsymbol{x}_G^n$, the GLL points, or a subset of these. *Region* $r$ of the domain $[-1, 1]$ is $[\boldsymbol{x}^n(r), \boldsymbol{x}^n(r+1)]$. A *middle region* is region $r = n/2 - 1$ if $n$ is even; if $n$ is odd, then there is no middle region.

### 3.2 Interpolants

The coefficients of the ISL space-time operator correspond to interpolation at departure points. We consider piecewise-polynomial interpolants having the following constraints. In Appendix B we state these constraints precisely; here, we summarize them sufficiently to motivate the subsets we choose to study in this article.

1. Given a departure point $x$, the interpolant is a linear operator in the function values $\boldsymbol{y}$ at basis points $\boldsymbol{x}$.

2. Within each region, the interpolant is a polynomial.

3. The interpolant must recover a degree-$p$ polynomial with specified $p = s - 1$ and $2 \leq s \leq n$. We call this the *order constraint*.

4. The interpolant must interpolate nodes $r$, $r + 1$; that is, it must have values $\boldsymbol{y}(r)$, $\boldsymbol{y}(r+1)$, respectively, at these nodes. We call this the *region interpolation constraint*. One consequence is that the resulting interpolant is continuous.

5. The nodal basis function associated with node $i$ of $n$ must have mirror symmetry around 0 with the basis function associated with node $n - i - 1$.

### 3.3 Nodal subset bases

In this article, with one exception, we limit our attention to two nested proper finite subsets of the infinite set of all basis sets that satisfy constraints 1–5: *nodal subset bases* (n.s. bases) and, a proper subset of these, *offset nodal subset bases* (o.n.s. bases), each with $\boldsymbol{x}^{n_p} = \boldsymbol{x}_G^{n_p}$.

A nodal subset $n_p$-basis $\mathcal{N}$ consists of two data sets. First, it has $n_p - 1$ sets of support nodes, $\mathcal{I}_r = \mathcal{N}(r)$, one for each region $r \in \{0, \ldots, n_p - 2\}$, with $(n_p^{\text{sub}})_r = |\mathcal{I}_r|$ the number of nodes in region $r$'s support. For brevity, we call a set of support nodes $\mathcal{I}_r$ a *support*. The region interpolation constraint requires $r, r+1 \in \mathcal{I}_r$. Second, it has the points $\boldsymbol{x}_G^{n_p} = \mathcal{N}_{\boldsymbol{x}}$.

We write the vector of reference domain coordinates of the support nodes for a region, the region's *support points*, as $\boldsymbol{x}^{n_p}(\mathcal{I}_r)$. Given these data, the basis functions take values $\mathcal{L}(x; \mathbf{I}^{n_p \times n_p}(\mathcal{I}_r, :), \boldsymbol{x}^{n_p}(\mathcal{I}_r))$ in region $r$, i.e. $x \in [\boldsymbol{x}^{n_p}(r), \boldsymbol{x}^{n_p}(r+1)]$.



A software implementation can use the barycentric form of the polynomial interpolant by first precomputing coefficients for each region.

*Offset* nodal subset bases in addition have consecutive indices in each $\mathcal{I}_r$, and thus a region's support can be defined by its number of support nodes and an offset index giving the smallest index node in the support.

So far, these definitions satisfy the first four constraints. Using the symmetry constraint, basis data for regions $r \geq \lfloor n_p/2 \rfloor$ are constructed by symmetry of the basis set from regions $r < \lfloor n_p/2 \rfloor$. In addition, when $n_p$ is even, the symmetry constraint implies that the middle region support, $\mathcal{I}_r$ with $r = n_p/2 - 1$, must correspond to support points that are symmetric about $x = 0$.

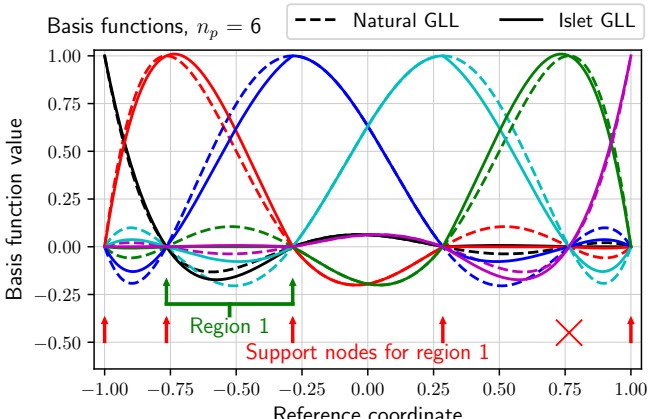

**Figure 1.** Basis functions for the Islet $n_p = 6$ GLL nodal subset basis listed in Table 1. Each curve's color corresponds to a basis function. Each line pattern corresponds to a basis type, as listed in the legend. The green span shows region 1. The red arrows point to the nodes in the support of region 1; the red $\times$ is beneath the one node not in region 1's support.

Figure 1 shows an example of a n.s. basis for $n_p = 6$. Each color corresponds to a basis function. Dashed curves are the
natural GLL basis functions. Solid curves are the n.s. basis functions. This basis corresponds to the entry for $n_p = 6$ in Table 1. There are five regions. The green span shows region 1. The red arrows point to the nodes in the support of region 1; the red $\times$ is beneath the one node not in region 1's support. Referring now to Table 1, regions 0 to 2 have $(n_p^{\text{sub}})_r$ values, in region order, of $\{5, 5, 6\}$, implying the basis provides order of accuracy 4. Region 1 has support $\mathcal{I}_1 = \{0, 1, 2, 3, 5\}$. Because this support omits node 4 and includes nodes 3 and 5, this basis is not an o.n.s. basis; $\mathcal{I}_1$ does not contain fully consecutive nodes. By
symmetry, region 3 has support $\mathcal{I}_3 = \{0, 2, 3, 4, 5\}$. Region 2 has support $\mathcal{I}_2 = \{0, 1, 2, 3, 4, 5\}$ and thus uses all nodes. The support nodes for region 2 are symmetric around the element center, thus satisfying the symmetry constraint. Consider now the basis function values as they pass through region 1. Each basis function can be identified with the node at which it takes value 1. Basis functions 0, 1, 2, 3, and 5 all have nonzero values in region 1 because these nodes are in $\mathcal{I}_1$. Basis function 4 has function values 0 throughout region 1 because node 4 is not in $\mathcal{I}_1$. As an example of one basis function, basis function 2 has
the values $\mathcal{L}(x; \begin{pmatrix} 0 & 0 & 1 & 0 & 0 \end{pmatrix}^T, \boldsymbol{x}_G^6([0, 1, 2, 3, 5]))$ in region 1.

| $n_p$ | OOA | $n_p^{\text{sub}}$ | Supports |
|---|---|---|---|
| 4 | 2 | see text | see text |
| 5 | 2 | $\{3, 4\}$ | offsets $\{0, 0\}$ |
| 6 | 4 | $\{5, 5, 6\}$ | nodal subsets $\{\{0, 1, 2, 3, 4\},$ $\{0, 1, 2, 3, 5\},$ $\{0, 1, 2, 3, 4, 5\}\}$ |
| 7 | 4 | $\{5, 5, 6\}$ | offsets $\{0, 0, 0\}$ |
| 8 | 5 | $\{6, 6, 7, 6\}$ | offsets $\{0, 0, 0, 1\}$ |
| 9 | 6 | $\{7, 8, 8, 7\}$ | nodal subsets $\{\{0, 1, 2, 3, 4, 5, 8\},$ $\{0, 1, 2, 3, 4, 5, 7, 8\},$ $\{0, 1, 2, 3, 4, 5, 6, 8\},$ $\{1, 2, 3, 4, 5, 6, 7\}\}$ |
| 10 | 6 | $\{7, 7, 7, 8, 8\}$ | offsets $\{0, 0, 0, 0, 1\}$ |
| 11 | 7 | $\{8, 9, 8, 9, 8\}$ | offsets $\{0, 0, 0, 0, 1\}$ |
| 12 | 8 | $\{9, 9, 10,$ $10, 9, 10\}$ | offsets $\{0, 0, 0, 0, 1, 1\}$ |
| 13 | 9 | $\{10, 10, 10,$ $10, 11, 10\}$ | offsets $\{0, 0, 0, 0, 0, 1\}$ |

**Table 1.** Islet GLL nodal subset bases. Each row provides a formula for the row's $n_p$ value. Columns are $n_p$, order of accuracy (OOA), the support sizes $n_p^{\text{sub}}$ for each region ordered left to middle, and the supports. For offset nodal subset bases, supports are given by offsets. For general nodal subset bases, supports are given by nodal subsets, again ordered from left region to middle. The case $n_p = 4$ is described in Sect. 3.8. In all cases, the support points are GLL points.

Let $r = \text{region}(x; \boldsymbol{x}^n)$ be the region containing $x$. If $x = \boldsymbol{x}^n(r)$, $\text{region}(x; \boldsymbol{x}^n)$ can return $r$ or $r - 1$; the value does not affect calculations because the interpolant is continuous. Let $n_p^{\text{submin}}$ be the minimum value of $|\mathcal{I}_r|$ over $r$. When present, a subscript $\mathcal{N}_{n_p}$ gives the total number of unique support nodes for the basis, and a superscript $\mathcal{N}_{n_p}^{n_p^{\text{submin}}}$ gives the minimum value of $|\mathcal{N}(r)|$ in the basis. Thus, for example, $\mathcal{N}_4^4$ denotes the natural GLL basis.

### 3.4 Accuracy heuristic

Let $p_d(x)$ be a degree-$d$ polynomial interpolant defined by support points $\boldsymbol{x}^n$, $n = d + 1$, in $[-1, 1]$ and including the endpoints, and nodal values $f(\boldsymbol{x}^n)$, with $f(x) \in C^{d+1}$ for $x \in [-1, 1]$. A standard result (e.g. Lozier, 2003, Sect. 3.3(i)) is that the





approximation error is bounded by

$$|p_d(x) - f(x)| \le \max_{\xi \in [-1,1]} \left| \frac{f^{(d+1)}(\xi)}{(d+1)!} \prod_{i=0}^{d} (x - \boldsymbol{x}^n(i)) \right|. \tag{2}$$

Based on this inequality, we define the following accuracy heuristics for the $l_1$, $l_2$, and $l_\infty$ norms. Let

$$e(x; \mathcal{N}) \equiv \left| \frac{1}{n!} \prod_{i=0}^{n-1} (x - \boldsymbol{x}(\mathcal{I}_r(i))) \right|,$$

$$\text{where } r = \text{region}(x; \boldsymbol{x}^n), \ n = |\mathcal{I}_r|.$$

The error heuristics for the basis are

$$a_1(\mathcal{N}) \equiv \int_{-1}^{1} e(x; \mathcal{N}) \, \mathrm{d}x,$$

$$a_2(\mathcal{N}) \equiv \left( \int_{-1}^{1} e(x; \mathcal{N})^2 \, \mathrm{d}x \right)^{1/2},$$

$$a_\infty(\mathcal{N}) \equiv \max_{x \in [-1,1]} e(x; \mathcal{N}).$$

The relation Eq. (2) is insensitive to Runge's phenomenon. For this reason, the Lebesgue constant is often used when designing interpolants, as it bounds worst-case error. However, in our application, we also have the requirement of test-problem stability, thus filtering out node placements that would produce Runge's phenomenon.

## 3.5   Metric of instability on a perturbed uniform mesh

Based on the results in Sect. 2, for any t.p.s. $\mathcal{N}$, we expect $\lambda_{\max}(\mathcal{N}) > 1 + \varepsilon$ for some nonuniform element grids. To quantify this instability, we create the following *perturbed uniform mesh metric*, denoted $\lambda_{\max}^{\text{PUM}}(\mathcal{N})$.

To approximate $\lambda_{\max}(\mathcal{N})$ for the test problem, recall that we create the matrix $\mathbf{B}$ corresponding to $\mathcal{N}$ and use the methods described in Sect. 2.3. In this case, $\mathbf{B}$ represents the action of the ISL operator on one target element. To approximate $\lambda_{\max}^{\text{PUM}}(\mathcal{N})$,

we construct $\mathbf{B}$ to represent the action of the ISL operator on multiple contiguous nonuniform target elements, as follows.

First, choose an element subgrid having $n_e^{\text{sub}}$ elements. Second, perturb each element boundary by $U(-\delta/2, \delta/2)$, where $U(a,b)$ is the uniform probability distribution over $[a,b]$ and $\delta$ is fraction of the subelement size. Thus, a subelement has size $U(1-\delta, 1+\delta)$ relative to the unperturbed size. Third, compute $\mathbf{B}$ as one matrix for all subelements. The result corresponds to periodic translation on a nonuniform element grid tiled by the $n_e^{\text{sub}}$-subelement grid. Fourth, approximate $\lambda_{\max}(B)$ almost

the same as usual. The one modification is the set of $\Delta x$ values; we explain this modification in Sect. 3.7. To approximate $\lambda_{\max}^{\text{PUM}}(\mathcal{N})$, repeat these four steps multiple times for a set of $n_e^{\text{sub}}$ values and take the maximum over the $\lambda_{\max}$ values. In practice we choose $n_e^{\text{sub}} \in \{3, \dots, 15\}$. The other parameter we must determine is $\delta$. In practice we choose a fixed $\delta = 0.01$, the choice of and insensitivity to which we explain in Sect. 3.7.





## 3.6 Search

We wrote software to search for t.p.s. n.s. bases. Because there are many of these, especially as $n_p$ grows, we use $a_{1,2,\infty}(\mathcal{N})$ and $\lambda_{\max}^{\text{PUM}}(\mathcal{N})$ to choose among the t.p.s. ones.

For a given $n_p$, the procedure first enumerates all possible o.n.s. bases, grouped in sets by decreasing $n_p^{\text{submin}}$. All bases use GLL node points. Once it finds a value of $n_p^{\text{submin}}$ for which there are t.p.s. bases and has tested all bases in this set, the procedure switches to enumerating all possible n.s. bases having this or larger value of $n_p^{\text{submin}}$.

In both phases, bases are filtered as follows. Three sets of uniformly spaced bins are constructed over a range of $\log_{10}(\lambda_{\max}^{\text{PUM}} - 1)$ values, roughly 0 down to $\log_{10}$ of machine precision, one set of bins for each of $a_{1,2,\infty}$. Each bin starts empty. Consider a candidate basis. A sequence of analysis steps is performed. If a step fails, the analysis stops and the procedure moves to the next candidate.

1. The basis's nodal weights, where a node's weight is the integral of its associated basis function, are computed; a non-285 positive weight is a failure. This step is fast.

2. Its $a_{1,2,\infty}$ values are computed. If no value is better than the best in each bin, the step fails. An empty bin implies success. This step is fast.

3. The basis is tested for test-problem stability. As we described previously, many small eigenvalue problems are solved in parallel. If any has $\lambda_{\max} > 1 + \varepsilon$, the step terminates and fails. A speedup is to search $\Delta x$ so that the range $(0, 1/2]$ is covered 290 quickly, coarsely at first, and then filled in at a roughly uniform rate everywhere, and the same for $\theta$. This enumeration of these sets encourages fast failures if a basis is not t.p.s. This step is slow if it succeeds but almost always is fast if it fails.

4. The largest $\lambda_{\max}^{\text{PUM}}$ value that would permit acceptance of the candidate based on its $a_{1,2,\infty}$ values is computed. This step is fast.

5. $\lambda_{\max}^{\text{PUM}}$ is computed, given the threshold from step 4. If a value is found that exceeds this threshold, this step terminates and 295 fails. This step is slow.

6. The basis is added to the bin corresponding to its $\lambda_{\max}^{\text{PUM}}$ value and to every bin with larger $\lambda_{\max}^{\text{PUM}}$ value for which the corresponding $a_{1,2,\infty}$ value is better than the best in that bin. This step is fast.

As the program proceeds, it can filter candidate bases more quickly, since the bins are populated with increasingly good bases. In addition, by starting with o.n.s. bases, the bins get populated quickly with good bases, making the search through the 300 much more numerous general n.s. bases faster.

We have completed this procedure for bases through $n_p = 10$. For bases having $n_p > 10$, we have completed this procedure for o.n.s. bases but only incompletely for general n.s. bases.

The final step is to select a basis from the set of bases in the bins. We have not developed a rule to select the basis since there is a trade-off between $\lambda_{\max}^{\text{PUM}}$ and accuracy. Instead, we follow a few guidelines to select the final one to recommend. First, we 305 always choose a basis having the maximum $n_p^{\text{submin}}$ we have found. Second, within that set, we prefer small $\lambda_{\max}^{\text{PUM}}$ values. A factor of roughly three difference in $\lambda_{\max}^{\text{PUM}}$ values does not matter, but a factor of roughly ten does. We then choose the most accurate basis within the tight cluster of $\lambda_{\max}^{\text{PUM}}$ values we have selected, weighting $a_2$ more heavily than $a_{1,\infty}$. For $n_p \leq 10$,





only $n_p = 6$ and 9 have significantly better non-offset than offset bases. In these two cases, $\lambda_{\max}^{\mathrm{PUM}}$ for the best o.n.s. basis is substantially higher than for the best non-offset one.

## 3.7 Results

Table 1 lists the final set of Islet GLL n.s. bases, subsequently *Islet bases*, one for each value of $n_p$ between 5 and 13. Section 3.8 describes the basis for $n_p = 4$, which is not a n.s. basis. Each row provides a formula for the row's $n_p$ value. Columns are $n_p$, order of accuracy (OOA) $n_p^{\mathrm{submin}} - 1$, the support sizes $n_p^{\mathrm{sub}}$ for each region ordered left to middle, and the supports. For o.n.s. bases, supports are given by offsets. For general n.s. bases, supports are given by nodal subsets, again ordered from left region to middle. Supports for the right half of the reference element are determined by the symmetry constraint. The bases for $n_p = 6$ and 9 are not offset.

As an example of an o.n.s. basis, consider the $n_p = 8$ basis. Regions 0 to 3 have $(n_p^{\mathrm{sub}})_r$ values, in region order, of $\{6, 6, 7, 6\}$. There are seven regions total, and region 3 is a middle region because $n_p$ is even. By symmetry, the $(n_p^{\mathrm{sub}})_r$ values for regions 0 to 6 are $\{6, 6, 7, 6, 7, 6, 6\}$. Because $n_p^{\mathrm{submin}}$ is 6, corresponding to a degree-5 interpolating polynomial, the basis yields an ISL method having order of accuracy 5. The offsets for regions 0 to 3 are $\{0, 0, 0, 1\}$. Thus, region 0 has support $\{0, 1, 2, 3, 4, 5\}$, and the middle region, region 3, has support $\{1, 2, 3, 4, 5, 6\}$. The support nodes for region 3 are symmetric around the element center to satisfy the symmetry constraint. By symmetry with region 0, region 6 has support $\{2, 3, 4, 5, 6, 7\}$.

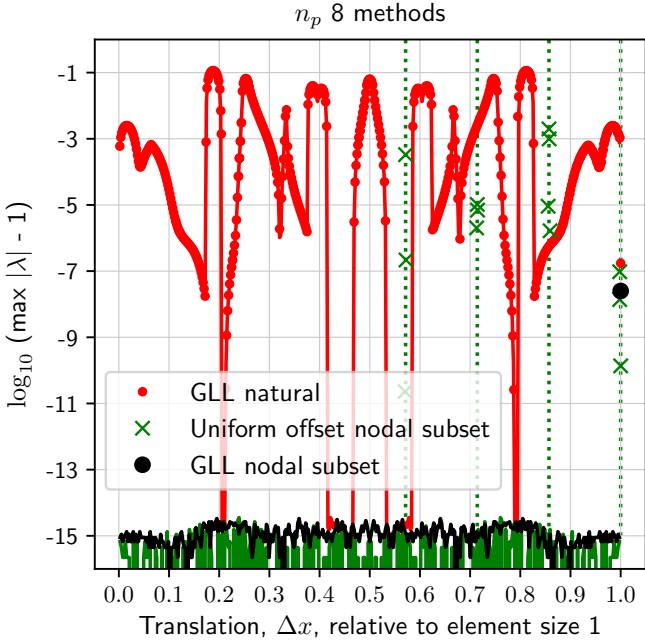

**Figure 2.** $\lambda_{\max}(\Delta x) - 1$ (solid lines) and $\lambda_{\max}^{\mathrm{PUM}}(\Delta x) - 1$ (markers) for the natural GLL (red, small circles), uniform-points offset nodal subset (green, $\times$), and Islet GLL nodal subset (black, large circle) $n_p = 8$ bases. Green dotted vertical lines mark multiples of $1/(n_p - 1) = 1/7$.



Figure 2 illustrates $\lambda_{\max}$ and $\lambda_{\max}^{\mathrm{PUM}}$ for three $n_p = 8$ bases. Color corresponds to basis: natural GLL, red; an o.n.s. basis with uniform points, green; Islet, black. The uniform-points basis is an o.n.s. basis with $n_p^{\mathrm{sub}}$ values $\{4, 4, 4, 4\}$ and offsets $\{0, 0, 1, 2\}$.

Solid lines shows the maximum eigenvalue magnitude of the space-time operator as a function of translation $\Delta x \in (0, 1]$, where again a translation of 1 corresponds to one full element. Note the symmetry around $\Delta x = 1/2$ of these curves. Markers show $\lambda_{\max}^{\mathrm{PUM}}$ as a function of translation $\Delta x$, with $\delta = 0.01$.

The natural GLL basis is not t.p.s.; thus, $\lambda_{\max}^{\mathrm{PUM}}$ values do not add new information and agree closely with $\lambda_{\max}$ values except at $\Delta x = 1$, where $\lambda_{\max} = 1$ but $\lambda_{\max}^{\mathrm{PUM}} > 1$. For the Islet basis, the only $\lambda_{\max}^{\mathrm{PUM}}$ value above 1 occurs at $\Delta x = 1$. In the case of

the t.p.s. basis having uniform points, $\lambda_{\max}^{\mathrm{PUM}}$ values above 1 occur at translations that are clustered around multiples of 1/7 and larger than 1/2. In general, we observe $\lambda_{\max}^{\mathrm{PUM}} > 1$ clustered around values of $\Delta x$ at which most of the translated nodes overlie the Eulerian nodes in a different element. For the Islet bases, these are integer values of $\Delta x$ other than 0. For a t.p.s. uniform-points basis, these are multiples of $1/(n_p - 1) \geq 1/2$. Thus, for t.p.s. bases, we compute $\lambda_{\max}^{\mathrm{PUM}}$ at $\Delta x$ values that follow these observations, rather than the values $\Delta x \in (0, 1/2]$ used when computing $\lambda_{\max}$. Since the discrete set size is much smaller than

for the computation of $\lambda_{\max}$, we can run more trials of randomly perturbed subelements in each test.

T.p.s. uniform-points n.s. bases having OOA 3 exist for $n_p \geq 8$, and in terms of accuracy, these are potentially useful. However, because uniform-points n.s. bases exhibit much larger $\lambda_{\max}^{\mathrm{PUM}}$ than GLL-node n.s. bases, we do not consider them further in this article.

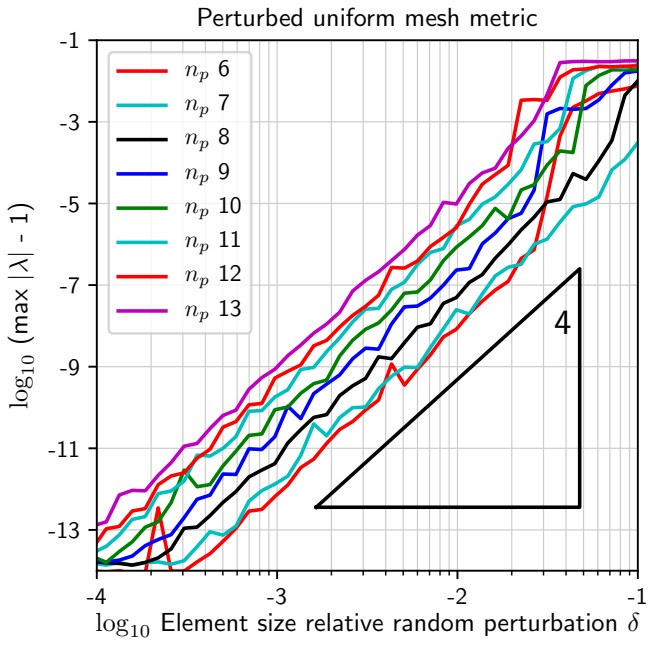

**Figure 3.** $\lambda_{\max}^{\mathrm{PUM}}(\delta) - 1$ for the bases in Table 1 with $n_p \geq 6$. The triangle provides a $\delta^4$ reference slope.





Figure 3 plots $\lambda_{\max}^{\text{PUM}} - 1$ vs. subelement perturbation $\delta$ for the bases in Table 1, excluding those for $n_p \leq 5$ because those

curves have a number of drops to machine precision values. Empirically, $\lambda_{\max}^{\text{PUM}} - 1$ is proportional to $\delta^k$, $k = 4$, for these bases, and for a fixed value of $\delta$, it tends to increase with $n_p$. Not all t.p.s. GLL n.s. bases fall off at this rate $k = 4$. Some fall off more slowly, $k < 4$, indicating greater instability on nonuniform grids. We have not specifically focused a search on attempting to find a t.p.s. basis with rate $k > 4$, but we have not observed any. Based on this plot, when computing $\lambda_{\max}^{\text{PUM}}(\mathcal{N})$ in the basis search procedure, we choose the specific value $\delta = 0.01$, with search results extremely insensitive to this value over a large

range of $\delta$.

### 3.8 Optimized interpolant

For the $n_p = 4$ GLL nodes, the t.p.s. n.s. basis has $n_p^{\text{sub}}$ values $\{3, 4\}$ with offsets $\{0, 0\}$, referred to subsequently as $\mathcal{N}_4^3$. In this subsection, we find a t.p.s. interpolant more accurate than this n.s. one.

The middle region already has $n_p^{\text{sub}} = n_p$ and thus is not modified. In the left region, a convex combination $\alpha(x)$ is sought

so that the basis functions are

$$(1 - \alpha(x)) \, \mathcal{L}(x; \mathbf{I}^{4 \times 4}(0\!:\!2, :), \boldsymbol{x}_G^4(0\!:\!2)) +$$
$$\alpha(x) \ \mathcal{L}(x; \mathbf{I}^{4 \times 4}, \boldsymbol{x}_G^4)$$

for $x \in [\boldsymbol{x}_G^4(0), \boldsymbol{x}_G^4(1)]$. The right region is symmetric to the left, as usual.

We set $\alpha(x)$, $x \in [\boldsymbol{x}_G^4(0), \boldsymbol{x}_G^4(1)]$, to be a quadratic polynomial having values $\boldsymbol{c} \in [0, 1]^3$ at left, middle, and right points of

the region and use the heuristics described in Sections 3.4 and 3.5, as well as the necessary condition for stability, to search for the optimal value of $\boldsymbol{c}$. For the left region, the accuracy heuristic is modified to be $\bar{e}(x) \equiv (1 - \alpha(x)) e(x; \mathcal{N}_4^3) + \alpha(x) e(x; \mathcal{N}_4^4)$, where $\mathcal{N}_4^4$ is the standard GLL 4-basis, and similarly for the right region. Results are insensitive to small perturbations to $\boldsymbol{c}$, so the final numbers are rounded to $\boldsymbol{c} = (1, 0.306, 0)$.

Figure 4 illustrates this basis. The top panel shows $\alpha(x)$ in the left region and its mirror image in the right region. The bottom

panel shows the natural (dotted), o.n.s. (dashed), and optimized (solid) basis functions.

## 4 The Islet method

Our primary application of the Islet bases is tracer transport $p$-refinement (TTPR). In $p$-refinement, the element grid is fixed, while the element basis parameter $n_p$ is increased relative to a baseline value to increase solution accuracy. Importantly, both the dynamical equations and the physics parameterizations use their own original discretizations and basis sets. We refer to

the set of algorithms composing this application as the *Islet method*. In some configurations we study, not every algorithm is active.

The tracer transport equation in conservation form is

$$\frac{\partial(\rho q_i)}{\partial t} + \nabla \cdot (\boldsymbol{u} \rho q_i) = \rho f_i(q_1, \ldots, q_n; \ldots) \tag{3}$$



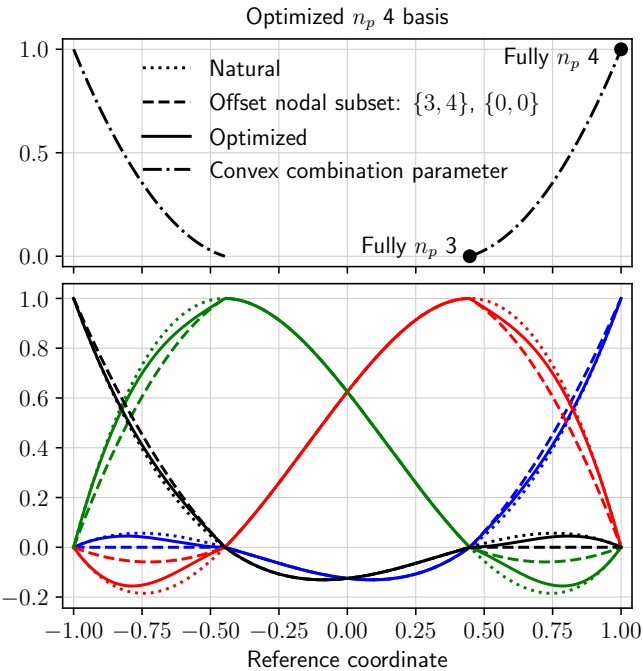

**Figure 4.** Illustration of the optimized Islet GLL $n_p = 4$ basis (solid line) compared with the natural (dotted) and the best nodal subset (dashed) $n_p = 4$ bases. Each basis function in a basis has its own color. The top panel shows the convex combination parameter value as a function of reference coordinate that is used to combine the natural and best nodal subset bases to form the optimized basis.

and in advective form is

$$\frac{Dq_i(\boldsymbol{y},t)}{Dt} \equiv \frac{\partial q_i}{\partial t} + \boldsymbol{u} \cdot \nabla q_i = f_i(q_1,\ldots,q_n;\ldots) \tag{4}$$

for $i \in \{1,\ldots,n\}$, where $\rho$ is total density, $\boldsymbol{y}$ is the spatial coordinate on the sphere, $\boldsymbol{u}$ is the flow velocity provided by the dynamics solver, there are $n$ mixing ratios $q_i$, and $f_i$ are the physical parameterization source terms. Each $f_i$ is a function of the mixing ratios as well as other quantities such as time, space, and variables from the dynamical equations, indicated by the trailing ellipses in the argument list to $f_i$. The two forms of the equation are related by the mass continuity equation $\partial \rho / \partial t + \nabla \cdot (\boldsymbol{u}\rho) = 0$. For convenience, we use base-1 indexing when labeling the mixing ratios and source terms.

The Islet method operates on three point grids sharing one element grid. Figure 5 shows one element, outlined by the solid blue line. The *dynamics grid*, black large circles, belongs to the dynamical core, excluding tracer transport, which solves the equations of motion for the atmosphere to provide $\boldsymbol{u}$ and $\rho$ in Eq. (3). Occasionally in figures we shorten the name to *v grid*. It is shown using the standard $n_p = 4$ GLL grid; we refer to this value subsequently as $n_p^{\mathrm{v}}$. The Islet method modifies only tracer transport in the dynamical core. The tracer transport module uses the *p*-refined *tracer grid*, small red circles, and the Islet basis from Table 1 corresponding to these GLL nodes to evolve the mixing ratios $q_i$. In Fig. 5, the tracer-grid $n_p$ value is 8; subsequently, we refer to the tracer-grid $n_p$ as $n_p^{\mathrm{t}}$. Finally, following Hannah et al. (2021), we couple the dynamics and tracer



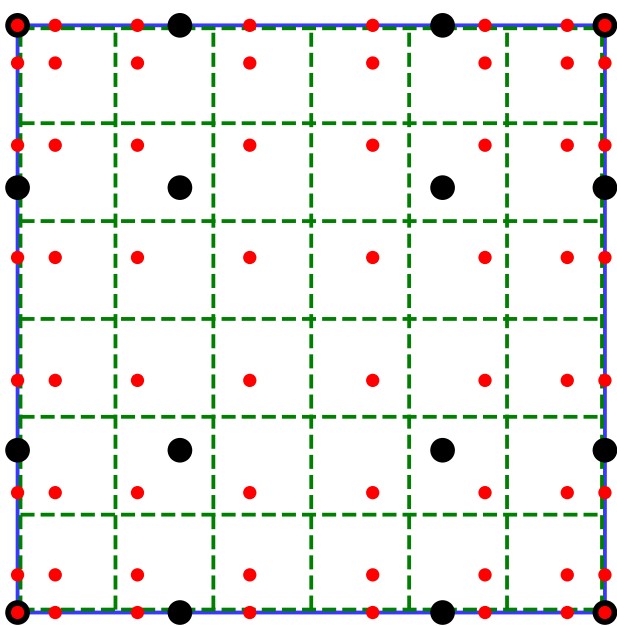

**Figure 5.** One spectral element (blue solid line outlining the full square) with dynamics (black large circles), tracer (small red circles), and physics (green dashed lines) subelement grids.

transport modules to a separate finite-volume (FV) *physics grid*, shown by the green dashed lines outlining an $n_f \times n_f$ grid of squares, $n_f = 6$. The subgrid-scale physics and chemistry parameterizations compute the source terms $f_i$ on the physics grid.

The ISL algorithm and the grid transfers are property preserving. First, global, and in some steps also element-local, mass are conserved. Second, tracer mass mixing ratio extrema are locally (and therefore also globally) constrained from growing. Thus, a constant mixing ratio is preserved, implying in turn that the method is *mass-tracer consistent*. Property preservation steps apply the local or global algorithms described in Bradley et al. (2019); see that reference for further details of property preservation.

In this section, we describe each algorithm that runs in a tracer transport time step, advancing the simulation from time step $n$ to $n+1$. In summary, first, total mass density $\rho$ and trajectory data are remapped from the dynamics grid to the tracer grid. Second, tracer tendencies $f_i \Delta t$ are remapped from the physics grid to the tracer grid, where $\Delta t$ is the physics parameterization time step. Third, the remapped tracer tendencies are added to the tracer-grid tracers. Fourth, the tracers are advected on the tracer grid. Fifth, tracer states are remapped to the physics and, optionally, dynamics grids. We also describe interprocess
communication procedures to maximize efficiency.

## 4.1   Preliminaries

This subsection describes notation, definitions, and mathematical details that we use subsequently.





Arithmetic between vectors applies entry-wise. All arithmetic applies to a single element of the element grid unless stated otherwise. Sub- or superscript letters in the set {v,f,t} indicate which grid the quantity is on, respectively dynamics, finite-volume physics, or tracer grids. Location of this letter has no meaning; it is placed where there is room and may switch position for the same quantity depending on context.

$\mathcal{I}^{n_p^{\mathrm{v}} \to n_p^{\mathrm{t}}}$ interpolates a field in an element on the dynamics grid, represented by the natural GLL $n_p^{\mathrm{v}}$-basis, to one on the tracer grid, represented by the Islet $n_p^{\mathrm{t}}$-basis, using the natural $n_p^{\mathrm{v}}$-basis functions to interpolate. $\mathcal{I}^{n_p^{\mathrm{t}} \to n_p^{\mathrm{v}}}$ does the opposite, using the Islet $n_p^{\mathrm{t}}$-basis functions.

The weight associated to each node in the subelement grid is, as usual, the integral of its associated basis function, whether natural or not, over the reference element; we denote it as $w_i$. Unless otherwise indicated, sums of the form $\sum_i$ are over the grid points in an element.

The interpolation of a field $f$ from the dynamics to the tracer grid, $\boldsymbol{f}^{\mathrm{t}} = \mathcal{I}^{n_p^{\mathrm{v}} \to n_p^{\mathrm{t}}} \boldsymbol{f}^{\mathrm{v}}$, has the useful property that it is conservative, $\sum_i w_i^{\mathrm{t}} f_i^{\mathrm{t}} = \sum_i w_i^{\mathrm{v}} f_i^{\mathrm{v}}$, despite not being in all cases an $L^2$ projection. First, for any Islet basis for which $n_p^{\mathrm{submin}} \geq n_p^{\mathrm{v}}$, the continuum field $f$ is the same on each grid, since the tracer grid can exactly represent polynomials of degree $n_p^{\mathrm{v}} - 1$. Second, for $n_p^{\mathrm{v}}$ even, such as the standard $n_p^{\mathrm{v}} = 4$, if $n_p^{\mathrm{submin}} = n_p^{\mathrm{v}} - 1$, then the continuum field $f$ is in general different on each grid but the integral of each over an element has the same value. This is because for polynomial degree $d$ odd and $f(x) \equiv \sum_{i=0}^{d} a_i x^i$, $\int_{-1}^{1} f(x)\,\mathrm{d}x = \int_{-1}^{1}(f(-x)+f(x))/2\,\mathrm{d}x$, and $g(x) \equiv (f(-x)+f(x))/2$ has degree at least one less than $f(x)$. $g(x)$ is exactly represented on the tracer grid, and thus the integral of $f$, which is the same as the integral of $g$, is the same on the tracer grid as on the dynamics grid. These two reasons assure that $\mathcal{I}^{n_p^{\mathrm{v}} \to n_p^{\mathrm{t}}}$ is conservative for the Islet bases in Table 1 when $n_p^{\mathrm{v}} = 4$.

Each grid point needs a value for the Jacobian determinant of the map from reference element to sphere, denoted $J$. On the dynamics grid, in practice this value is the usual isoparametric-element Jacobian determinant, possibly modified so that the sum over the whole unit sphere is $4\pi$. However, note that what follows is independent of the definition of $J^{\mathrm{v}}$. On the tracer grid, we instead interpolate the Jacobian determinant values from the dynamics grid,

$$\boldsymbol{J}^{\mathrm{t}} = \mathcal{I}^{n_p^{\mathrm{v}} \to n_p^{\mathrm{t}}} \boldsymbol{J}^{\mathrm{v}}, \tag{5}$$

for two reasons. First, this operation conserves the area of the element, the sum $\sum_i w_i^{\mathrm{v}} J_i^{\mathrm{v}}$, as we just discussed. We describe the second reason when discussing $\rho$ in Sect. 4.2. On the physics grid, we follow a similar procedure, explained in Hannah et al. (2021, Sect. 2.2.1), that is specialized to the finite-volume physics grid. Here again, the area of an element on the physics grid is the same as on the dynamics grid. Thus, all three subgrid definitions of the element area agree.

In a SEM, most operations are performed independently in each element, often leading to discontinuities in a field across element boundaries. The global direct stiffness summation (DSS) operator (Dennis et al., 2012) restores continuity. At each grid point on an edge of an element, the multi-valued solution is restored to a single value by weighted summation of contributions from each element sharing the grid point. An element's weight at that grid point is the value $w_i J_i$ normalized by the sum of all contributing elements' values.





### 4.2 Algorithms


We name each algorithm using the format **Step::algorithm-name** and sometimes **Step::algorithm-name::sub-algorithm-name**. Unless stated otherwise, each operation acts within a single element.

**Step::density-d2t**. Interpolate $J\rho^{n+1}$ from the dynamics grid to the tracer grid using the natural GLL $n_p^{\mathrm{v}}$-basis interpolant: $\boldsymbol{\rho}_{\mathrm{t}}^{n+1} = (\mathcal{I}^{n_p^{\mathrm{v}} \to n_p^{\mathrm{t}}}(\boldsymbol{J}_{\mathrm{v}}\boldsymbol{\rho}_{\mathrm{v}}^{n+1}))/\boldsymbol{J}_{\mathrm{t}}$. This quantity will not be needed for several steps, but we describe it here because $J_{\mathrm{t}}\rho_{\mathrm{t}}^n$ is

needed in **Step::tendency-f2t**. This grid transfer is conservative because it uses $\mathcal{I}^{n_p^{\mathrm{v}} \to n_p^{\mathrm{t}}}$. $J_{\mathrm{t}}\rho_{\mathrm{t}}^{n+1}$ is not continuous across element boundaries because $J$ is not, but continuity is not needed. In the case that $\rho$ is constant on the dynamics grid, it is constant on the tracer grid, as follows from Eq. (5). This is not a necessary property to have, but since it is possible, we use it; it is the second reason to define $J^{\mathrm{t}}$ according to Eq. (5).

**Step::tendency-f2t**. Map the tracer tendencies $\Delta q^n$ from the physics grid to the tracer grid. This step involves multiple

sub-algorithms.

**Step::tendency-f2t::bounds**. In each element, compute and store the minimum and maximum mixing ratio state $q_{\mathrm{f}}^n$ values, subsequently *extrema*. Let an element's neighborhood contain itself and its immediate element neighbors. Augment an element's extrema with the extrema over its neighborhood. Finally, augment an element's extrema again with the extremal values of the tracer-grid mixing ratio state $q_{\mathrm{t}}^n$. This final extrema update assures that if $\Delta q_{\mathrm{f}}^n = 0$, then $q_{\mathrm{t}}^n$ is unmodified. These final

extrema in an element are the bounds used in subsequent property preservation corrections.

**Step::tendency-f2t::linear-remap**. Apply the linear, element-local, conservative, panel-reconstruction (PR) remap operator described in Hannah et al. (2021, Sect. 2.2.3) to map the tendency from the physics grid to the tracer grid. In the Islet method, but unlike in Hannah et al. (2021), the basis used in the mass matrix of the $L^2$ projection is the Islet basis. From the previous step's application of **Step::density-d2t**, we have $\rho_{\mathrm{t}}$. The operator uses this quantity and $\rho_{\mathrm{f}}\Delta q_{\mathrm{f}}^n$ to compute $\rho_{\mathrm{t}}\Delta q_{\mathrm{t}}^n$.

**Step::tendency-f2t::CAAS**. In each element, apply CLIPANDASSUREDSUM, subsequently CAAS, Algorithm 3.1 in Bradley et al. (2019), to $\bar{q}_{\mathrm{t}}^n \equiv q_{\mathrm{t}}^n + \Delta q_{\mathrm{t}}^n$. Each node weight is the integral of the corresponding Islet basis function times $J$. In this application of the element-local CAAS algorithm, the same upper and lower bounds apply to each GLL node in the element. In this step, the bounds that **Step::tendency-f2t::bounds** computed are relaxed in each direction by 1% of the difference between upper and lower bounds. This relaxation is one part of obtaining good toy chemistry diagnostic values, as we shall

describe in Sect. 5.4.2. The exact bounds will be enforced in **Step::CEDR**. The constraint set of mass conservation and bounds nonviolation is non-empty because the constant mixing ratio is a solution, as explained in Hannah et al. (2021, Sect. 2.3).

**Step::tendency-f2t::DSS**. At this point, $\bar{q}_{\mathrm{t}}^n$ is discontinuous across element boundaries. Neither order of accuracy nor property preservation requires continuity, but we find the toy chemistry diagnostic value (see Sect. 5.4.2) is large without restoration of continuity prior to the ISL step. $\rho_{\mathrm{t}}^n$ can remain discontinuous. Because of this, the DSS arithmetic is slightly different than

usual. Consider an element boundary node indexed by global ID $g$ and having DSS weights $w_g^e$ – the product of reference-to-sphere Jacobian determinant and Islet basis function weights – for contributing elements indexed by $e$. In element $e$, $(\rho_{\mathrm{t}}^n)_g^e$ is the restriction of $\rho_{\mathrm{t}}^n$ to node $g$. Then the post-DSS value at the node is

$$(\hat{q}_{\mathrm{t}}^n)_g^e \equiv \frac{\sum_e w_g^e (\rho_{\mathrm{t}}^n)_g^e (\bar{q}_{\mathrm{t}}^n)_g^e}{\sum_e w_g^e (\rho_{\mathrm{t}}^n)_g^e},$$



where the sum $\sum_e$ is over elements that contain node $g$. In contrast, in the standard DSS, $(\rho_t^n)_g^e$ is absorbed into the DSSed
quantity, so that the tracer density, rather than the mixing ratio, is made continuous.

**Step::advect-interp**. Compute and apply the linear space-time ISL operator. This step involves two substeps: computing the
grid point trajectories and computing the interpolants.

**Step::advect-interp::trajectory**. The dynamics component supplies velocity data at the dynamics GLL grid points. Any
of a number of algorithms can compute departure points at time $n$ backward in time from dynamics-grid arrival GLL points
at time $n+1$. The Islet method takes as input these departure points as 3D Cartesian departure points. In each element, the
natural GLL interpolant $\mathcal{I}^{n_p^v \to n_p^t}$ is applied to each of the Cartesian components separately to obtain departure points on the
tracer grid. This procedure implies that, first, adjacent elements compute identical departure points at shared boundaries and,
second, at GLL points common to the $n_p^v$- and $n_p^t$-bases, the resulting departure points are identical. Finally, for simplicity in
the subsequent sphere-to-reference map computations, the 3D Cartesian points are normalized to the sphere. For $n_p^v = 4$, the
$n_p^t$-basis departure points are obtained at OOA 4.

**Step::advect-interp::mixing-ratio**. Each departure tracer-grid GLL point is mapped to the containing element, subse-
quently the *source* element. Details of finding the source element depend on host-model implementation details and are omitted
here; possibilities include octree search, $O(1)$ arithmetic for quasiuniform cubed-sphere element grids having certain reference-
to-sphere maps, and search within a predefined element neighborhood whose size is proportional to maximum wind speed times
advection time step. Then the corresponding reference coordinates within the source element are computed using Newton's
method. The mixing ratio value is computed at the departure point using the Islet basis interpolant and the source element's
$\hat{q}_t^n$ values. In addition, the source element's stored extrema are associated to this point as bounds. Finally, the mixing ratio
value and bounds are assigned to the target GLL point on the arrival tracer grid. Departure points and interpolant weights are
calculated once and then are reused for each tracer.

**Step::CEDR**. Apply local and then global Communication-Efficient Density Reconstructors (CEDR) (Bradley et al., 2019)
on the tracer grid. As usual, node weights correspond to Islet basis functions and not to the natural GLL basis.

**Step::CEDR::local**. First, apply the element-local CAAS algorithm. Neither order of accuracy nor property preservation
requires this step. But this step reduces the amount of global mass redistribution in **Step::CEDR::global**, is computationally
inexpensive, and involves no interprocess communication, and thus is worth including in the overall procedure. Again we relax
bounds in each direction by 1% of the difference between upper and lower bounds. Unlike in **Step::tendency-f2t::CAAS**, in
this application of the element-local CAAS algorithm, any two target GLL nodes in an element may have different bounds;
the bounds depend on the source element for the target GLL node, as detailed in **Step::advect-interp::mixing-ratio**. At
this point, the global mass has not yet been corrected; thus, this local CAAS application's mass constraint is to maintain
the element's current tracer mass. The constraint set is not assuredly feasible. Thus, Algorithm 3.4 in Bradley et al. (2019),
RECONSTRUCTSAFELY, wraps the call to CAAS. This algorithm relaxes the bounds according to degree of infeasibility before
calling CAAS.

**Step::CEDR::global**. In this work, we use global CAAS applied at the tracer-grid GLL grid point level; we refer to the
procedure as *CAAS-point*. The exact, rather than relaxed, bounds are applied to each node. Continuity across element bound-





aries was restored to the mixing ratio field in **Step::tendency-f2t::DSS** and was maintained in subsequent steps; this step also

maintains it. In finite precision, continuity does not hold to machine precision. However, first, no step of the overall algorithm is sensitive to this level of error; second, the error does not grow because, in each step, **Step::advect-interp::mixing-ratio** restores exact continuity in finite precision. The inputs to the global CAAS algorithm are as follows: $J_t\rho_t^n$ and $\hat{q}_t^n$ to provide the global tracer mass after the tendency update, $J_t\rho_t^{n+1}$ as part of the global tracer mass at time step $n+1$, the mixing ratio bounds computed in **Step::tendency-f2t::bounds**, and the current mixing ratio values from **Step::CEDR::local**. The final

tracer-grid values are $q_t^{n+1}$.

**Step::state-t2f**. Remap the mixing ratio state to the physics grid. This is a purely element-local operation. First, the linear operator described in Hannah et al. (2021, Sect. 2.2.1) is applied to the tracer density. Second, the element-local CAAS algorithm is applied on the physics grid, with the extremal mixing ratio values in the element on the tracer grid as the bounds. For the same reason as in **Step::tendency-f2t::CAAS**, the constraint set is assuredly non-empty.

**Step::state-t2v**. The dynamics solver needs one or more mixing ratios on the dynamics grid, e.g., specific humidity. In addition, in our numerical results, we compute all errors, except as indicated, on the dynamics grid, so we use this step to obtain those errors. First, in an element, the Islet basis is used to interpolate the tracer-grid mixing ratio to the $n_p^v$-basis. Second, the element-local CAAS algorithm is applied on the dynamics grid to preserve shape and conserve mass; details are as in **Step::state-t2f** but with GLL nodes instead of FV subcells. Third, the standard DSS is applied to obtain continuous tracer

density and mixing ratio fields.

**Step::state-v2t**. In validation problems in Sect. 5, we need to remap a mixing ratio initial condition from the dynamics grid to the tracer grid. The algorithm is the same as **Step::state-t2v** except that the DSS follows the procedure in **Step::tendency-f2t::DSS** since the mass density on the tracer grid is and remains discontinuous at element boundaries.

If $n_p^t = n_p^v$, then tracer transport $p$-refinement is not enabled. In this case, identity maps replace a subset of the algorithms

described in this subsection: **Step::density-d2t**, the interpolation part of **Step::advect-interp::trajectory**, **Step::state-t2v**, and **Step::state-v2t**. When describing numerical experiments, we indicate when TTPR is not enabled.

### 4.3 Interprocess communication

To realize the full computational efficiency of an ISL method, we must limit the interprocess communication to the union of the discrete domains of dependence of the grid points on an owning process. In addition, our methods must run on GPU archi-

tectures; therefore, the assembly and parsing of messages must be highly parallel within a process. This subsection discusses interprocess communication for the Islet method, including some high-level details relevant to an efficient GPU implementation.

A process *owns* one or more elements and the grid points within those elements. A process communicates with remote processes, or *remotes*. We say that an element that is owned by a process or remote is *on* that process or remote. Each element

has a *halo*. A 0-halo is the element; an $(h+1)$-halo, $h \geq 0$, is the union of the $h$-halo and the elements adjacent to elements in the $h$-halo. The time step is restricted according to a configurable maximum $h$; in our implementation, we set $h = 2$. Recall





that a grid point's *source* element is the element containing its Lagrangian departure point; its *target* element is the element containing its Eulerian arrival point.

### 4.3.1 Advection step

**Step::advect-interp::mixing-ratio** has two communication rounds.

The first communication round exchanges lists of departure points. Each process prepares one message for each of its potential partners, where a second process is a potential partner if it owns at least one element in the $h$-halo of the first process's owned elements. We call this message an $x$-message.

An $x$-message contains two sections: first, metadata to identify the source element for each departure point on the remote;
second, the bulk data list of departure points whose source elements are on that remote. For efficiency, during model initialization, each process computes its remotes' local IDs for each element in its halo; then during time stepping, the source element identification data in the $x$-message is the remote's local ID for the element. This procedure means that only direct array lookups rather than hash- or tree-based map lookups are needed during time stepping. If a potential remote has no source elements in a step, then the message is essentially empty but is still sent since it is expected. A process is a potential partner of
itself. In this special case, the list is implicit in data structures reserved for departure points that remain on the owning process; we call this list the $x$-self-list.

In a GPU implementation, assembling these lists involves parallel-scan and parallel-for loops as well as atomic-access writes to bookkeeping data structures. The parallel-scan loop writes the $x$-message metadata and computes pointers into the $x$-message and future received $q$-message bulk data. The parallel-for loop copies departure points into the $x$-message bulk data.
While a process waits to receive its $x$-messages, it computes extrema in each owned element.

The second communication round exchanges interpolated mixing ratios and source-element extrema. A process receives one $x$-message from each of its potential remotes. For each departure point in an $x$-message, it computes the interpolated mixing ratio. Then it writes two sets of data to a new message, the $q$-message corresponding to the $x$-message. The first set contains mixing ratio extrema for each active source element. It writes these data once per active source element per remote. The second
set contains the interpolated mixing ratio data, one for each departure point. If an $x$-message is empty, the process neither writes nor sends a corresponding $q$-message.

In a GPU implementation, a parallel-scan loop, one per potential remote, is used to parse an $x$-message's metadata and compute pointers into the $q$-messages. Then one parallel-for loop over all requested departure points is used to compute and write the $q$-messages, using these pointers. Atomic accesses are not needed in this communication round.

After a process sends all of its $q$-messages to remotes, and while it is waiting to receive its $q$-messages, it computes interpolated mixing ratio data corresponding to entries in its $x$-self-list. This step requires one parallel-for loop and no atomic accesses.

When a process receives all of its $q$-messages, it copies the data to its internal data structures. Bookkeeping that was done while forming and writing $x$-messages permits this step to be done with one parallel-for loop and no atomic accesses.





Finally, the process waits on its sent $q$-messages to clear the MPI buffers. When the waits are done, **Step::advect-interp::mixing-ratio** is complete.

### 4.3.2    Communication volume

*Communication volume*, sometimes just *volume* when the context is clear, is the volume of data transmitted during interprocess communication. Consider the case of one element per process. The worst-case volume per element of both the $x$-messages
and $q$-messages is proportional to the number of grid points, $(n_p^{\rm t})^2$. For example, at most $(n_p^{\rm t})^2$ total interpolated mixing ratio values must be communicated from the source elements to the target element, regardless of halo depth. In contrast, a naive $h$-halo exchange of full-element mixing ratio data would communicate $c_h(n_p^{\rm t})^2$ values, $c_h = (2h+1)^2 - 1$; accounting for continuity, the number could be reduced to a little more than $c_h(n_p^{\rm t}-1)^2$. Thus, the volume reduction factor is at least $[(2h+1)^2 - 1](n_p^{\rm t}-1)^2/(n_p^{\rm t})^2$. As examples, for $h=1$, $n_p^{\rm t}=4$, the factor is 4.5; for $h=2$, $n_p^{\rm t}=4$, 13.5; for $h=2$, $n_p^{\rm t}=8$,
over 18.

The volume of extrema data depends on the number of source elements a departure element overlaps and very little on $n_p^{\rm t}$, since only one pair of extrema data per source element is communicated to a partner. Thus, the proportion of the $q$-message that contains extrema data decreases with increasing $n_p^{\rm t}$.

Other communication rounds require only standard SEM halo exchanges that are already part of a SEM host model. These
include exchanging element-edge data, proportional to $n_p^{\rm v}$, when computing departure points; element single-scalar data, thus independent of subgrid parameters, to share source element extrema in **Step::tendency-f2t::bounds**; and another element-edge exchange, proportional to $n_p^{\rm t}$, in the DSS in **Step::tendency-f2t::DSS**.

CAAS-point requires a reduction. Many climate host models expect answers to be invariant to the number of MPI processes and thus provide a reproducible reduction routine. CAAS-point can maintain reproducibility while keeping the reproducible
reduction independent of $n_p^{\rm t}$ by reducing arrays within an element, then calling the global reproducible reduction routine with element-level values. See Bradley et al. (2019, Sect. 6.1) for further details on CEDR communication.

In summary, only $x$- and $q$-messages have volume proportional to $(n_p^{\rm t})^2$; all other messages have volume proportional to $n_p^{\rm t}$ or $n_p^{\rm v}$ or independent of subgrid parameters. In addition, $x$- and $q$-messages have volume substantially smaller than would occur with full halo exchanges.

## 5    Numerical results

This section presents results for a number of validation problems. Except in Sect. 5.4, the equation is the sourceless advection equation, $f_i = 0$ in Eq. 4. Time-dependent flow $\boldsymbol{u}$ is imposed.

In most figures, we show results for $n_p^{\rm t} = 4, 6, 8, 9, 12$. The $n_p^{\rm t} = n_p^{\rm v} = 4$ case provides a reference because it does not use the TTPR algorithms. $n_p^{\rm t} = 6$ is of interest because it is the smallest value of $n_p^{\rm t}$ with basis OOA greater than 2, in this case
4. $n_p^{\rm t} = 8$ has OOA 5 and has four times as many nodes as the $n_p^{\rm t} = 4$ basis in two dimensions. $n_p^{\rm t} = 9$ has OOA 6. Finally, $n_p^{\rm t} = 12$ has OOA 8 and four times as many nodes as the $n_p^{\rm t} = 6$ basis.





Tests follow the procedures detailed in Lauritzen et al. (2012). Results can be compared with those from many models described in Lauritzen et al. (2014). We refer to these articles frequently and thus abbreviate them as TS12 ("test suite") and TR14 ("test results"), respectively. Initial conditions are generated on the dynamics grid. Similarly, error diagnostics are

computed on the dynamics grid in most cases; we state the exceptions when they occur. In most cases we omit results for simulations without property preservation, as tracer transport modules in earth system models are expected to be property preserving. We have not attempted to make this section self-contained, as describing details of the large number of validation problems would take too much space. We recommend the reader not familiar with these problems read TS12. In addition, we refer to specific figures in TR14 and sometimes TS12 so the reader can compare our results with those from previously

documented methods.

We briefly summarize the key characteristics of the validation problems. There are two prescribed flows: a nondivergent one and a divergent one. Each prescribes a flow that lasts for $T = 12$ days and such that at time $T$, the exact solution is the same as the initial condition (IC). This 12-day prescribed flow can be run for multiple cycles to lengthen the simulation. The nondivergent flow creates a filament of maximum aspect ratio at time $T/2$. The divergent flow tests treatment of divergence.

There are four initial conditions that share the feature of placing two circular shapes at two points along the equator: the $C^\infty$ Gaussian hills, the $C^1$ cosine bells, the correlated cosine bells used in the mixing diagnostic, and the discontinuous slotted cylinders. Assessing the behavior of a transport method on tracers having various degrees of continuity is important because atmosphere tracers can be smooth or nonsmooth.

This article does not study time integration methods to generate the dynamics-grid departure points. Thus, to remove tempo-

ral errors due to time integration algorithms, we use an adaptive Runge-Kutta method (Dormand and Prince, 1980; Shampine and Reichelt, 1997) with tight tolerance ($10^{-8}$, with an exception noted later) to integrate trajectories very accurately. All tests for $n_p^{\mathrm{t}} > 4$ use TTPR with $n_p^{\mathrm{v}} = 4$ unless we state otherwise. Since $n_p^{\mathrm{v}} = 4$ in these validation problems, the $n_p^{\mathrm{t}} = 4$ configuration does not use TTPR.

We use a quasiuniform cubed-sphere element grid. Let a cube face of the cubed-sphere element grid have $n_e \times n_e$ elements.

Long tracer time steps correspond to $6n_e$ steps per $T$; short, $30n_e$. These are the same time step settings as the two CSLAM (Lauritzen et al., 2010) model configurations used in TR14.

An alternative to running CAAS-point is to run the global CAAS method over elements rather than grid points, then to apply element-local CAAS to each element; we refer to this procedure as CAAS-CAAS. See Bradley et al. (2019, Sect. 7.2) for further details. CAAS-CAAS tends to give slightly more accuracy than CAAS-point. But it does not permit the bound relaxation in

the element-local CAAS application that we find is necessary to obtain good results for the toy-chemistry diagnostic. However, because it provides slightly more accurate results, we use it for $n_p^{\mathrm{t}} = 4$, except where noted, while using CAAS-point for all other $n_p^{\mathrm{t}}$ values, thus maximizing the accuracy obtained with $n_p^{\mathrm{t}} = 4$ to provide the best baseline performance.

The total mass density $\rho$ is not needed for advection of mixing ratios, but it is needed for property preservation of these. In these validation problems, we have no independent computation of density $\rho$, as occurs when transport is coupled to a full

dynamical core. For the nondivergent flow, we could of course just set $\rho$ to a constant, but for the divergent flow, that would be incorrect. Thus, we need a means to compute $\rho$ on the dynamics grid. We discretize the Lagrangian formulation of the continuity





equation, $\int_{A_e(t_2)} \rho(x, t_2)\,\mathrm{d}x = \int_{A_e(t_1)} \rho(x, t_1)\,\mathrm{d}x$, using $\rho(x_i, t_2) J_e(x_i, t_2) = \rho(x_i^*(t_1), t_1) J_e(x_i^*(t_1), t_1)$. In these expressions, $x_i$ is an Eulerian grid point, $x_i^*$ is its departure point at $t_1$, $A_e(t)$ is the domain of Lagrangian element $e$, $A_e(t_2)$ is the domain of the element in its arrival Eulerian position, and $J_e$ is the isoparametric determinant of the map from reference element to

spherical element $e$. $J_e$ is a function of time because $J_e(\cdot, t_2)$ corresponds to its arrival Eulerian configuration and $J_e(\cdot, t_1)$ to its departure Lagrangian configuration. First, $\rho$ is advected on the dynamics grid using **Step::advect-interp::mixing-ratio**. Second, the resulting value at grid point $i$ in element $e$ is multiplied by the density factor $J_e(x_i^*(t_1), t_1)/J_e(x_i, t_2)$. Third, because this discretization is not mass conserving, the mass is corrected by adding $\Delta m / a_{\text{total}}$ to each grid point, where $\Delta m$ is the global mass discrepancy after advection and $a_{\text{total}}$ is the total area of the grid. Negative density does not occur in these

validation problems.

### 5.1    Unstable and stable integration

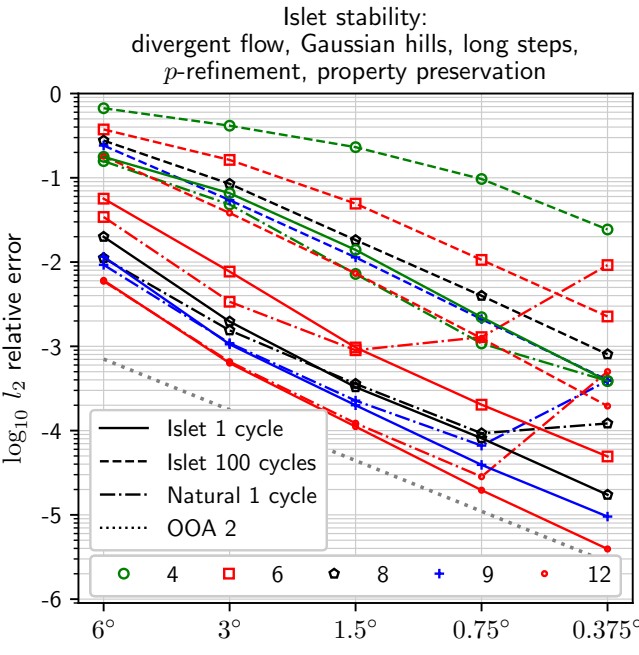

**Figure 6.** Stability of the Islet method with the Islet GLL bases, compared with the instability of the method with the natural GLL bases. The $x$-axis is average dynamics grid point spacing at the equator in degrees for the quasiuniform cubed-sphere grid. The $y$-axis is $\log_{10} l_2$ relative error. A curve's line pattern corresponds to basis type and number of cycles, as listed in the top legend. A curve's marker corresponds to $n_p^{\text{t}}$, as listed in the bottom legend. The case is divergent flow, Gaussian hills ICs, property preservation, TTPR, and long time steps.

Figure 6 compares accuracy and stability between natural and Islet bases. The divergent flow, Gaussian hills IC, property preservation, TTPR, and long time steps are used. A curve's marker corresponds to $n_p^{\text{t}}$, as listed in the legend. To maximize font size and minimize notational clutter in figures, in figures we use $n_p$ rather than $n_p^{\text{t}}$ and omit "=", e.g., $n_p^{\text{t}} = 8$ is written $n_p$ 8;



additionally, sometimes we omit $n_p$ entirely, as in the legend of this figure. The $x$-axis is average dynamics-grid point spacing at the equator in degrees for the cubed-sphere grid. Thus, for example, $n_e = 5$ corresponds to $360°/(4\,\text{faces} \times n_e\,\text{elements/face} \times (n_p^{\mathrm{v}} - 1)\,\text{regions/element}) = 6°/\text{region}$. The $y$-axis is $\log_{10} l_2$ relative error. A curve's line pattern corresponds to basis type and number of cycles: solid, Islet for 1 cycle; dashed, Islet for 100 cycles (or $12 \times 100 = 1200$ days); dash-dotted, natural for 1 cycle. The dotted straight line is a reference for OOA 2. For each $n_p^{\mathrm{t}}$, the $l_2$ norm of the solution using the natural basis diverges

with increasing resolution within the first cycle, demonstrating that the basis leads to an unstable Islet method. In the case of $n_p^{\mathrm{t}} = 4$, we see the start of the curve's divergence, but further element-grid refinement is needed to see the curve fully diverge. In contrast, the curves for the Islet method with the Islet bases converge at OOA 2.

## 5.2 Accuracy for $C^\infty$ and $C^1$ tracers

### 5.2.1 Time integration check

Sometimes validation problems have unexpected features that interact with a method to produce higher-accuracy solutions than would occur in a more realistic problem. Of particular concern is the symmetry of the flow in time around the midpoint time, 6 days. To be sure our trajectory interpolation procedure is not interacting with this symmetry to produce artificially higher accuracy, we run a test in which the solution error is computed at the midpoint time as well as the final time.

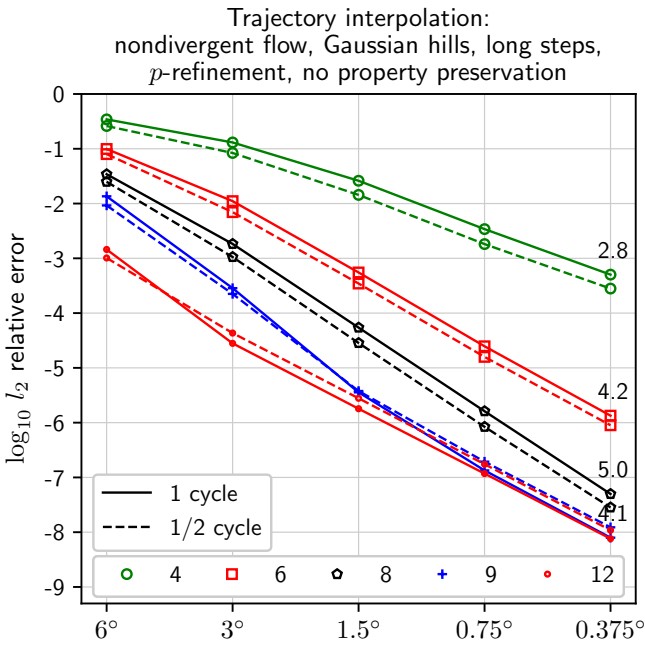

**Figure 7.** Comparison of relative errors calculated at the test simulation's midpoint time of 6 days (1/2 cycle, dashed lines) and endpoint time of 12 days (1 cycle, solid lines). Each number at the right side of the plot is the empirical OOA computed using the final two points of the 1-cycle result.





The test uses the nondivergent flow, Gaussian hills IC, and long time steps. Property preservation is turned off to expose

fully the temporal error. The midpoint reference solution is computed using one 6-day step and the natural GLL basis. The time

integrator's relative error tolerance is set to $10^{-14}$ rather than its usual $10^{-8}$ for this step. One remap step with the natural GLL

basis provides a midpoint solution much more accurate than the time-stepped case, thus serving as an appropriate reference.

Figure 7 shows results. The dashed curves show the error measured at the midpoint, half of a cycle of the 12-day problem; the

solid curves, the usual endpoint. The numbers at the right side of the plot show empirical OOA computed using the final two

points of the one-cycle results. Numbers for the half-cycle curves are omitted because each pair of curves has almost exactly

parallel lines for resolution at least as fine as $0.75°$. For fine enough resolution, all curves should converge with OOA 4 because

the tracer-grid trajectories are interpolated using the $n_p = 4$ natural interpolant from the dynamics grid, but for $n_p^t = 8$, the full

convergence regime is not reached in this plot, thus giving OOA 5 for $n_p^t = 8$. The $n_p^t = 4$ curves have OOA less than 4 due to

a spatial OOA limit of 2 in the full convergence regime. We see OOA at about 4 for $n_p^t = 9$ and 12. The $n_p^t = 9$ curve shows

at coarse resolution higher OOA, governed by the spatial error, and then a drop to 4 as the temporal error becomes dominant.

Importantly, the half- and full-cycle errors are very close for each value of $n_p^t$, demonstrating that the endpoint error metrics

are valid measurements for the Islet method.

### 5.2.2  Empirical order of accuracy

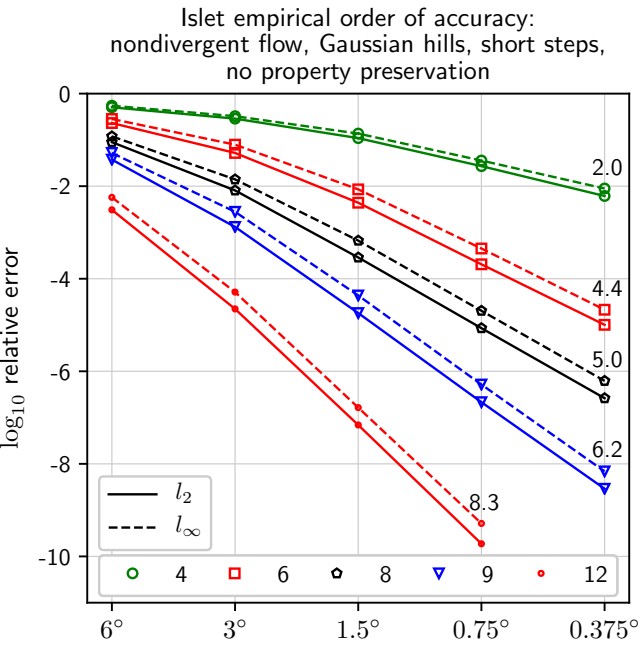

**Figure 8.** Empirical verification of the order of accuracy of the Islet GLL bases. Each number at the right side of the plot is empirical OOA computed using the final two points of the $l_\infty$ curve.





Figure 8 empirically verifies the OOA of the Islet bases. The test uses the nondivergent flow, Gaussian hills IC, and short time steps. To expose the OOA of the basis, this test does not use TTPR – the trajectories are computed nearly exactly at each quadrature point, and thus $n_p^v = n_p^t$ – and property preservation is turned off. Errors are reported in the $l_2$ (solid lines) and $l_\infty$ (dashed lines) norms. The numbers on the right side of the plot are empirical OOA calculated using the final two points of the $l_\infty$-norm curves; the values for the $l_2$-norm curves are about the same. The $n_p^t = 12$ curve has one fewer element-grid refinement point because the $0.375°$ one is slightly influenced by the machine-precision limit on relative accuracy. Each empirical OOA is at least as large as, but not much larger than, the expected OOA listed in Table 1.

### 5.2.3 Accuracy heuristic

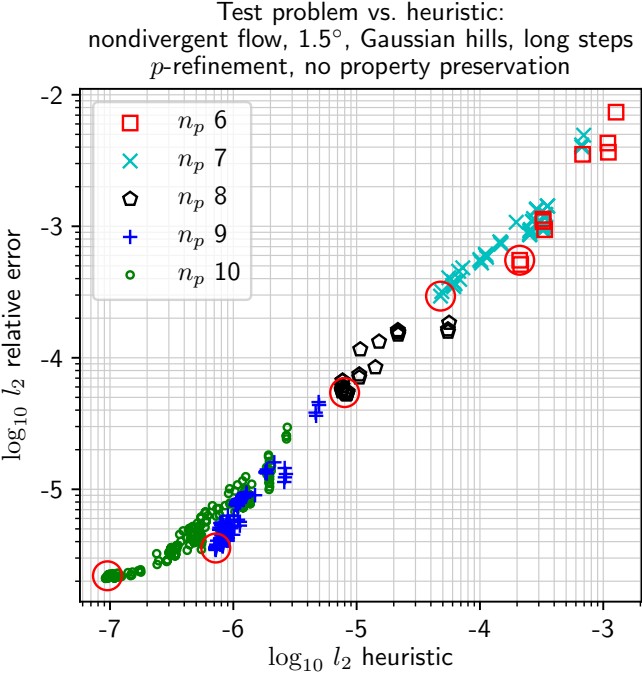

**Figure 9.** $l_2$ norm on the nondivergent flow problem using basis $\mathcal{N}_{n_p}$ vs. $a_2(\mathcal{N}_{n_p})$, for a large number of t.p.s. bases and $n_p = 6$ to 10. The legend lists the marker type for each $n_p$. Large red circles outline the bases in Table 1. The configuration uses the Gaussian hills IC and no property preservation.

To select Islet bases for each value of $n_p^t$, recall that we used accuracy heuristics $a_i$, $i \in \{1, 2, \infty\}$, to choose accurate bases, and the perturbed uniform mesh metric to filter out bases that might be unstable in practice. The accuracy heuristics are functions of the basis data only, not any specific problem. Figures 9 and 10 plot accuracy on a validation problem vs. the value of the $a_2$ heuristic for a large number of bases. Each point in a scatter plot corresponds to a basis. The $y$-axis is accuracy of bases when used to simulate the nondivergent flow at resolution $1.5°$, in two different cases: Gaussian hills with no property





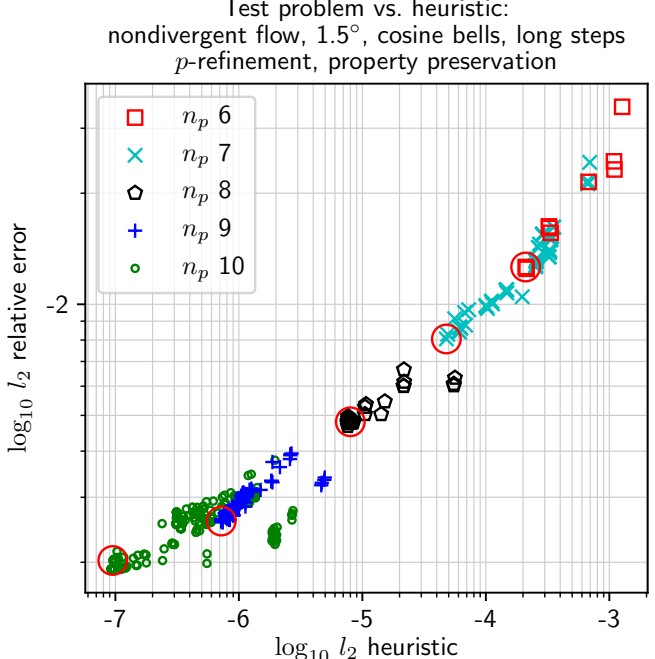

**Figure 10.** Same as Fig. 9 except that the configuration uses the cosine bells IC with property preservation.

preservation (Fig. 9) and cosine bells with property preservation (Fig. 10). The $x$-axis is the value of $a_2$ for the basis. The marker pattern corresponds to $n_p^t$. For each $n_p^t$, one red circle outlines the basis that is in Table 1. The $a_2$ accuracy heuristic is a good predictor of relative accuracy on the validation problems. We chose the bases in Table 1 using the heuristics without reference to validation problem results, as we do not want accuracy on any particular problem to influence the selection of recommended bases.

### 5.2.4 Accuracy data for other standard configurations

The next figures show accuracy for standard configurations used in TR14. Figure titles provide the test configuration details. Although we explain how each figure can be compared with corresponding ones in TR14, these figures also stand on their own as simply convergence plots for various test cases.

Figures 11 and 12 can be compared with Figs. 1, 2 in TR14. They evaluate error on an infinitely smooth IC. Figures 13 and 14 provide data that can be compared with the top panel of Fig. 3 in TR14. They evaluate error on a $C^1$ IC. The horizontal dash-dotted line provides the relative $l_2$-error-norm value of 0.033 by which the "minimal resolution" diagnostic value is determined; the coordinate of the intersection between $l_2$-norm curve and this reference line is the value. For example, with a long time step, for $n_p^t = 8$, this value is a little coarser than $3°$; for $n_p^t = 12$, approximately $6°$.



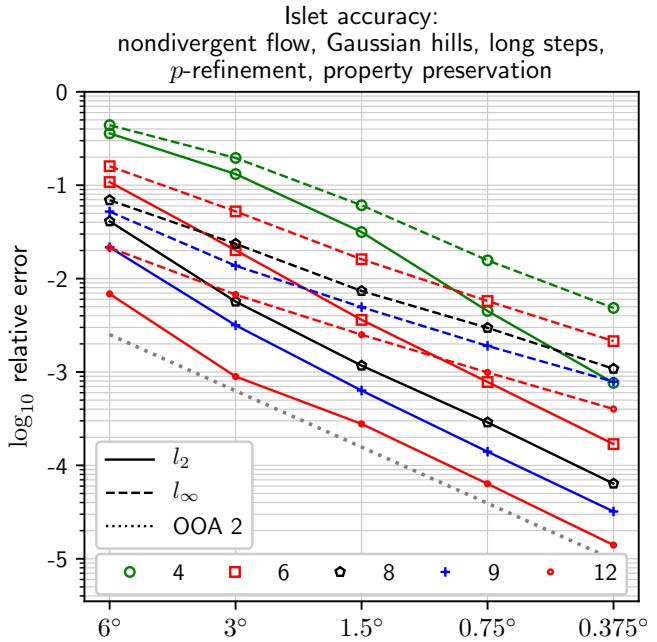

**Figure 11.** Accuracy diagnostic. Compare with Figs. 1, 2 in TR14.

The Islet method with a high-order basis compares extremely favorably with the best of the methods in TR14. For example, the most accurate shape-preserving method in TR14 for the nondivergent flow with Gaussian hills IC is HEL-ND-CN1.0 (cyan curves in Fig. 1, bottom right, of TR14). The $n_p^t = 12$ Islet scheme with long time steps, Fig. 11, has even more accuracy than HEL-ND-CN1.0 in the $l_2$ norm.

Figures 15 and 16 provide data that can be compared with the top two panels in Fig. 16 of TR14. They are like Figs. 13 and 14, but here the divergent flow is used.

### 5.2.5 Filament diagnostic

Figure 17 shows results for the filament diagnostic described in Sect. 3.3 of TS12, to compare with Fig. 5 in TR14. We used the code distributed with TS12 to compute the results. The diagnostic uses the nondivergent flow with cosine bells IC. In this
test, the midpoint solution is analyzed to determine the quality of the filamentary structure. See Fig. 20 for an illustration of the filamentary structure at the simulation midpoint, although with the slotted cylinders IC. For each value of the mixing ratio at the initial time, $\tau \in [0.1, 1]$, the area over which the mixing ratio is at least $\tau$ at the midpoint time is computed. The diagnostic is then this area divided by the correct area, which for nondivergent flow is the area at the initial time. The perfect diagnostic value is 100% for all $\tau \in [0.1, 1]$ and 0 otherwise. In each plot in Fig. 17, the $x$-axis is $\tau$ and the $y$-axis is the diagnostic value.
The diagnostic is computed for the dynamics-grid resolutions and time step lengths listed in the legend. Note that the $y$-axis limits are tighter with increasing $n_p^t$.



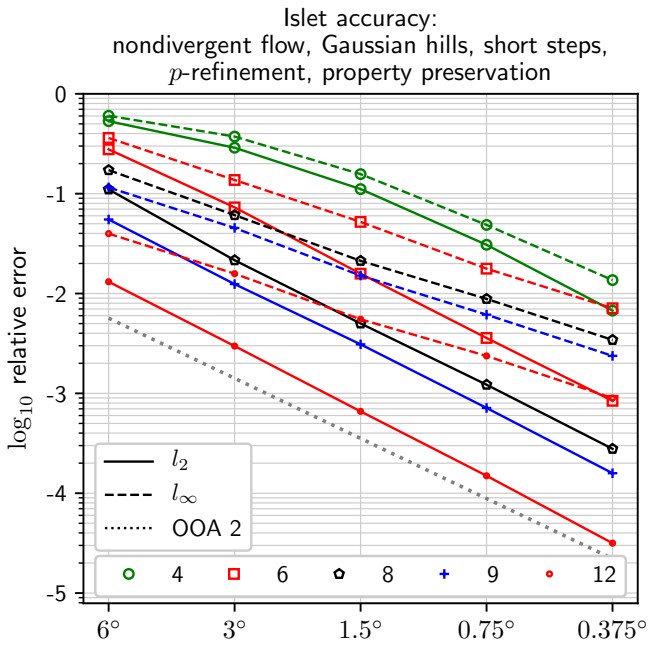

**Figure 12.** Accuracy diagnostic. Compare with Figs. 1, 2 in TR14.

A subtlety with this diagnostic is that the area calculation must use the quadrature method of the discretization. On the dynamics grid, the resulting curve is noisier than is implied by the underlying solution on the tracer grid. Thus, in Fig. 17, we show the diagnostic as computed on the dynamics grid in the top row of plots; on the tracer grid, in the bottom row.

### 5.2.6 Mixing diagnostic

Figures 18 and 19 show results for the mixing diagnostic described in Sect. 3.5 of TS12, to compare with Figures 11–14 in TR14. Like the filament diagnostic, the analysis is done at the solution midpoint rather than the endpoint. Also as for the filament diagnostic, we used the code distributed with TS12 to compute the results. The problem is nondivergent flow with two ICs: cosine bells on the $x$-axis of each plot, and the correlated cosine bells field on the $y$-axis. Each dot in a plot is a grid-point sample from the dynamics grid of the mixing ratios. A perfect diagnostic has the red dots all on the convex-upward black reference curve. $n_p^t$ and time step length is printed in each plot. Diagnostic values $l_r$ and $l_u$ are printed in each plot; the parenthesized "(v)" suffix means the value is measured on the dynamics grid, while the un-suffixed values are measured on the tracer grid. Because we show results only for simulations with property preservation on, $l_o = 0$ always, and so we do not print those values. We omit a description of these scalar diagnostics and refer the reader to TS12.

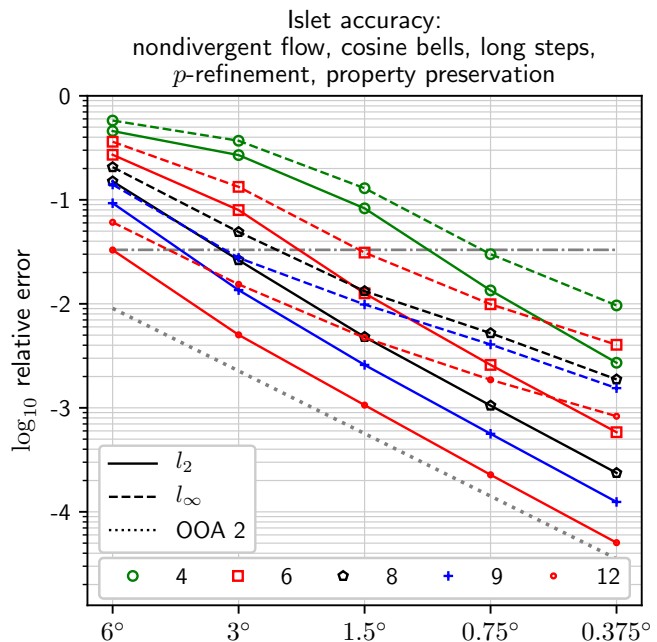

**Figure 13.** Accuracy diagnostic. Compare with Fig. 3 in TR14.

## 5.3 Slotted cylinders

We observe that in both the filament and mixing diagnostics of TS12, $n_p^t \geq 6$ gives excellent results; $n_p^t = 12$, nearly perfect. Figures 20 and 21 show solution quality using latitude-longitude images and further underscore these observations. The problem is nondivergent flow with the slotted cylinders IC, at resolutions $1.5°$ and $0.75°$, with long and short time steps. The text in the individual images in Fig. 21 provides normwise accuracy at the end of one cycle and deviation from the initial extrema. $\phi_{\min} \geq 0$ and $\phi_{\max} \leq 0$ are consistent with no global extrema violation. Compare Fig. 20 with Figs. 7–10 in TR14 and both figures with Fig. 7 in TS12.

## 5.4 Source terms

Now we move to validation problems that include a source term.

### 5.4.1 Accuracy

The first test validates the property-preserving remaps between physics and tracer grids. The test is constructed as follows. Two tracers, a source tracer $q_1 = s$ and a manufactured tracer $q_2 = m$, are paired. $m(0)$ is set to 0. A tendency $\Delta m$ is applied to $m$ on the physics grid: $\Delta m(t) = -[\cos(2\pi(t + \Delta t)/T) - \cos(2\pi t/T)]s(t)/2$, so that the exact solution is $m(t) = (1 - \cos(2\pi t/T))s(t)/2$ and, in particular, $m(T/2) = s(T/2)$. To compute the tendency, the state $s(t)$ must be remapped to the



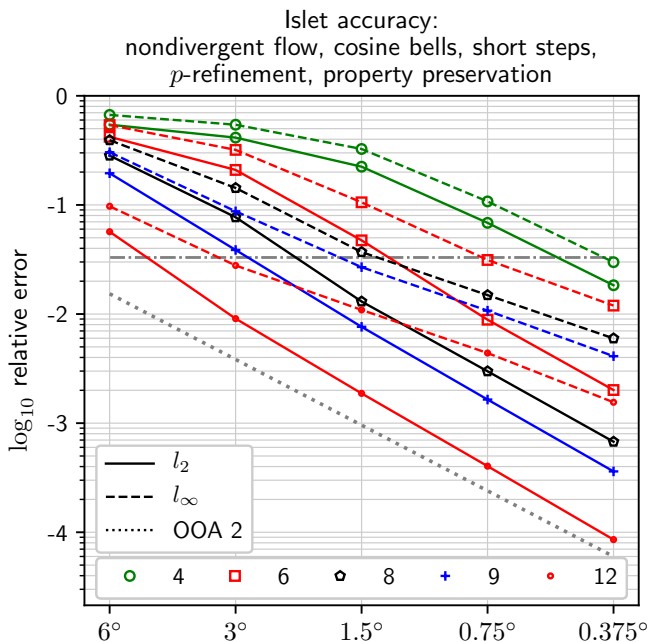

**Figure 14.** Accuracy diagnostic. Compare with Fig. 3 in TR14.

physics grid. Thus, this test depends on accuracy in both grid-transfer directions. We measure the error at time $T/2$, on the

745 dynamics grid as usual, as in Fig. 7. Figure 22 shows results. We run this test with $n_f = n_p^t$ (dash-dotted lines) and $n_f = 2$ (dashed lines). The solid lines show the error in $s(T/2)$ as a reference. The dotted line provides the OOA-2 reference. We see that when $n_f = n_p^t$, the errors are nearly the same as those for $s(T/2)$; for $n_p^t = n_f = 4$, the curves overlap at the resolution of the plot. As one expects, when $n_f = 2$, the error in $m(t)$ is much larger than in $s(T/2)$, but the OOA remains 2.

### 5.4.2 Toy chemistry diagnostic

The toy chemistry validation problem is described in Lauritzen et al. (2015), subsequently TC15. The problem consists of two tracers, $q_i = X_i$, $i = 1, 2$, that interact according to kinetic equations that are nonlinear in one of them: $DX_i(\boldsymbol{y}, t)/Dt = f_i(\boldsymbol{y}, X_1, X_2)$, where $\boldsymbol{y}$ is the spatial coordinate on the sphere. The tracers are composed of a monatomic and a diagnostic molecule, respectively, of the same atomic species. The ICs are designed so that the sum over atomic mixing ratio at each point in space is a constant, $\bar{X}_T$. The source terms have this property, too, since they model chemical reactions. Thus, in the exact

solution, $\bar{X}_T$ is maintained at every point in space and time. Let $X_T$, without a bar, be the corresponding measured quantity. The toy chemistry diagnostics are then $c_2(t)$, the $l_2$ norm of $X_T - \bar{X}_T$ at time $t$, normalized by $\bar{X}_T$, and $c_\infty$, the same but for the $l_\infty$ norm.



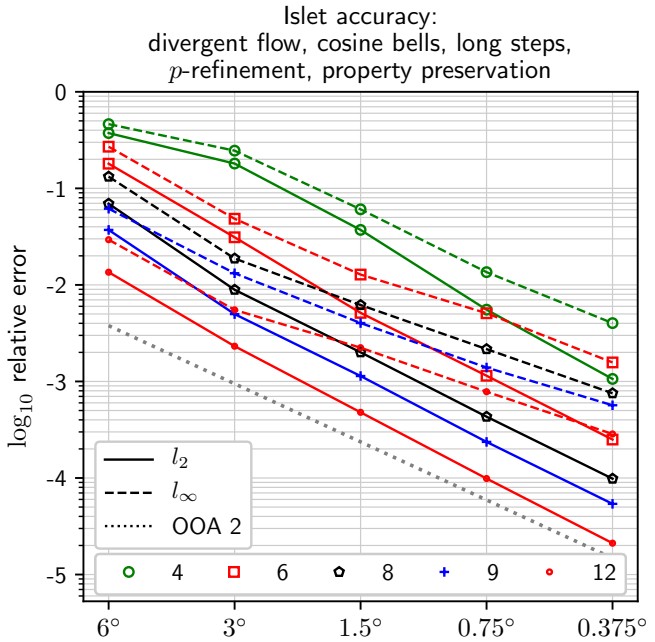

**Figure 15.** Accuracy diagnostic. Compare with Fig. 16 in TR14.

As explained in the context of equation 14 in TC15, any advection operator that is *semi-linear* will produce a perfect diagnostic value of 0 when using exact arithmetic. Linear operators are semi-linear; the CEDR algorithms we use are, as well, as explained in Bradley et al. (2019); and a composition of semi-linear operators is, too. Thus, the Islet method is semi-linear.

We compute the diagnostics, as usual, on the dynamics grid. Following the Islet tracer transport method described in Sect. 4.2, the source terms are computed on the physics grid using states remapped from the tracer grid, and then the computed tendencies are remapped to the tracer grid.

It is already known that the Eulerian spectral element tracer transport method yields poor values for this diagnostic due to finite-precision effects of the limiter (Lauritzen et al., 2017). Islet with property preservation using CAAS-CAAS does, as well. In exact arithmetic, each of these methods would produce perfect values. The poor diagnostic values are due to quickly accumulating machine-precision truncation errors that break semi-linearity in finite precision. The interaction of the chemistry source term with exact bounds in element-local limiter applications is responsible for this fast accumulation. In contrast, Islet with property preservation using CAAS-point and relaxed-bound, element-local CAAS applications produces good diagnostic values because the relaxed bounds in the element-local part make unnecessary many of the mixing ratio adjustments that lead to loss of semi-linearity in finite precision. Recall that CAAS-point, applied at the end of the step, imposes the exact bounds, so at the end of a time step, shape preservation still holds to machine precision. Clips to bounds must still occur, but adjustments to other grid points to compensate are smaller because the adjustments are spread over many more grid points.



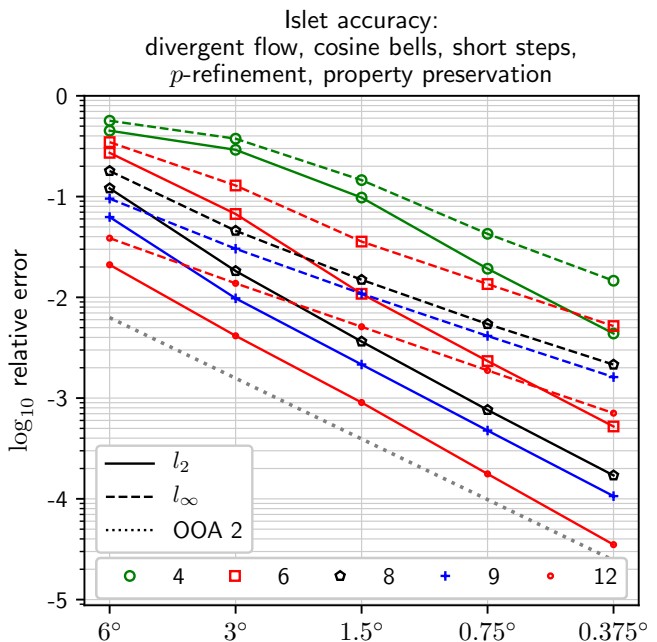

**Figure 16.** Accuracy diagnostic. Compare with Fig. 16 in TR14.

Figure 23 shows the diagnostic values for the case of nondivergent flow, $1°$ dynamics-grid resolution, and a 30-minute time step, where these configuration details are prescribed in TC15. For the $n_p^t = 4$ case, we use CAAS-point rather than CAAS-CAAS as previously, since we already know that CAAS-CAAS will produce poor values. The diagnostic is usually plotted over the course of one cycle (12 days) of the flow, but it is useful to view it over multiple cycles. Figure 23 shows ten cycles on the $x$-axis, for a total of 120 days. The $y$-axis is the diagnostic value. Solid lines plot $c_2$; dashed, $c_\infty$. Markers are placed on the curves at the start of each cycle to help differentiate the curves. In each case, $n_f = n_p^t$. We choose this value of $n_f$ because the toy chemistry source term has nearly a singularity at the terminator – the terminator is clearly seen in Fig. 24 – and thus it makes sense to compute the physics tendencies at high spatial resolution. The $n_p^t = 4$ case with CAAS-point is greatly improved relative to Eulerian spectral element results shown in Fig. 7 of TC15, even after ten cycles instead of the one shown in that figure. For $n_p^t > 4$, the growth in error is very small, with $c_2$ less than $10^{-10}$ through ten cycles.

To illustrate what these diagnostics measure, Figure 24 shows latitude-longitude images of the monatomic tracer at the end of the first cycle for $n_p^t = n_f = 4$ with CAAS-CAAS (left) and $n_p^t = n_f = 8$ with CAAS-point (right). Note that the images in TS12 and TR14 are plotted with longitude ranging from 0 to $2\pi$; those in TC15, from $-\pi$ to $\pi$. In our latitude-longitude figures so far, we have chosen the convention used in TS12 and TR14, and we continue to use it in these toy-chemistry images. Thus, these images are circularly shifted horizontally by half the image width relative to those in TC15. The globally extremal tracer values are printed in the upper-right quadrant of each image. The correct maximum is $4 \times 10^{-6}$ and the correct minimum is at least 0. The right image is free of noise and satisfies these bounds. The left image shows substantial noise, as we expect





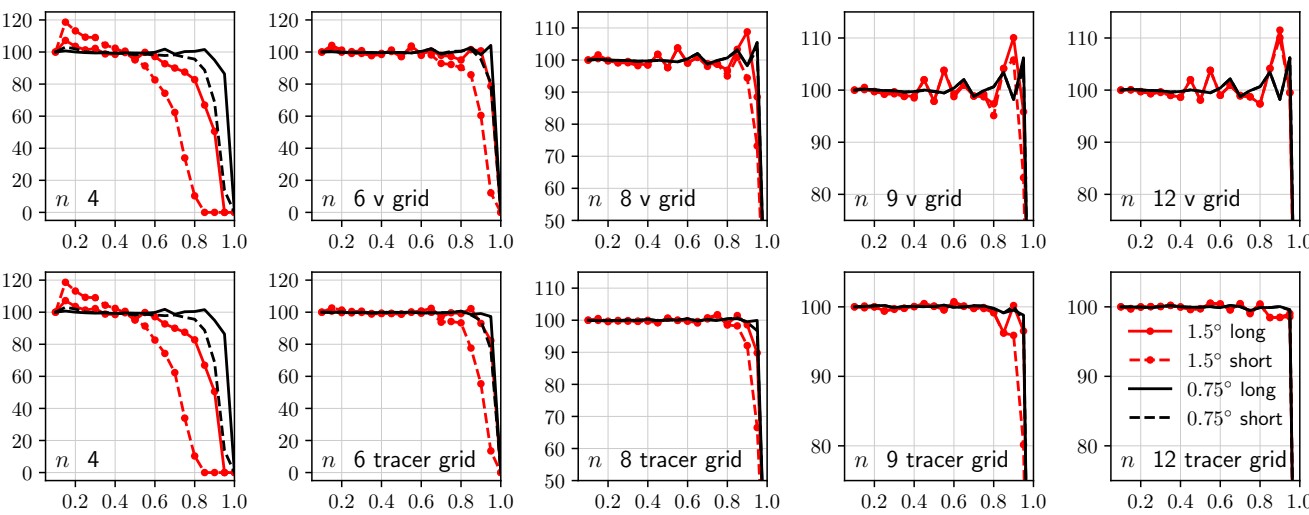

**Figure 17.** Filament diagnostic, following Sect. 3.3 of TS12. Compare with Fig. 5 in TR14. The top row shows the diagnostic measured on the $n_p^v = 4$ dynamics grid; the bottom row, on the tracer grid. The legend describes the dynamics-grid resolution and the time step length. The prescribed validation problem is the nondivergent flow with cosine bells IC. Property preservation is on. The $x$-axis is $\tau$, the mixing ratio threshold. The $y$-axis is the percent area having mixing ratio at least $\tau$ relative to that at the initial time.

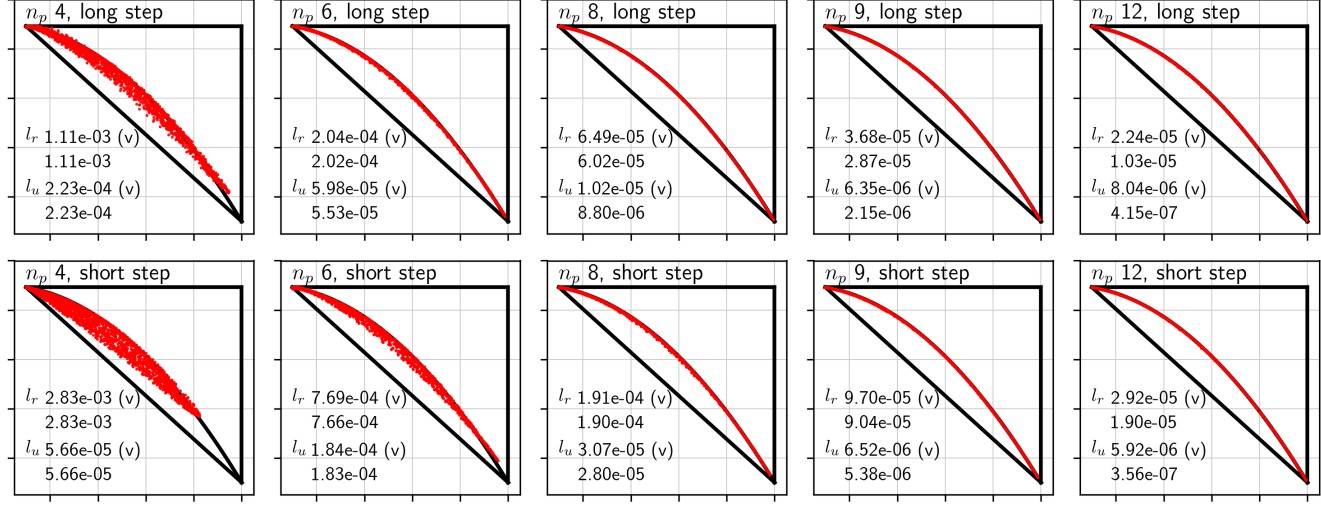

**Figure 18.** Mixing diagnostic, following Sect. 3.5 of TS12. Compare with Figs. 11–14 in TR14. This figure shows results for dynamics-grid resolution of $1.5°$. $l_o$ is exactly 0 in all cases because shape preservation is on, and so is not shown. See text for further details.

when using exact bounds in the local property preservation problems, and consistent with previous observations about spectral element transport. Other than noise and some filaments that grow from the noise, the two images are qualitatively similar.





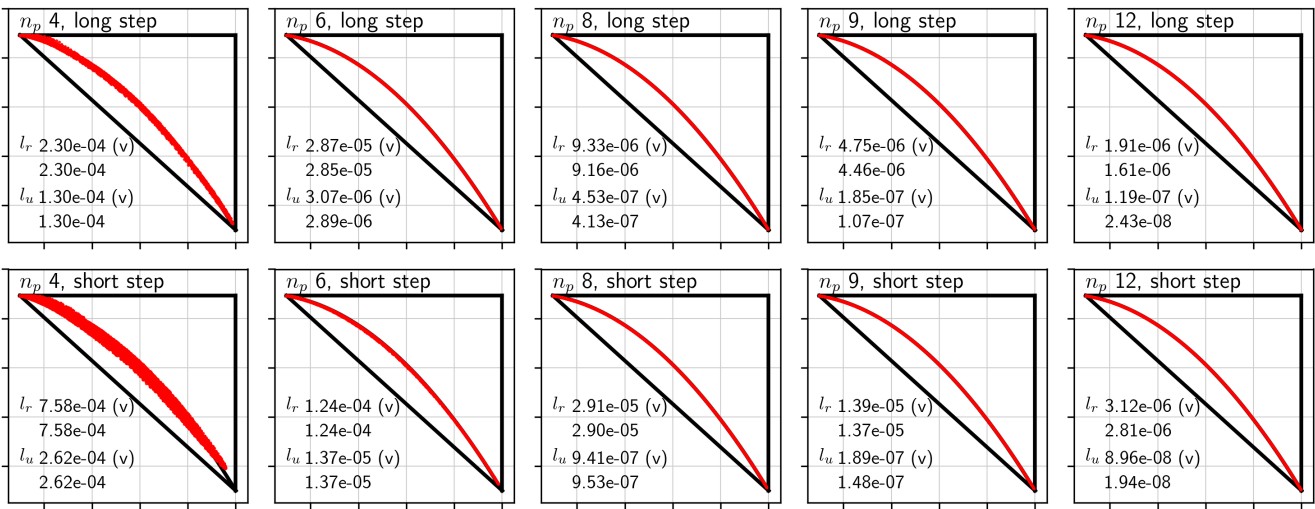

**Figure 19.** Same as Fig. 18 but with dynamics-grid resolution $0.75°$.

Figure 25 shows images in the same format, but the quantity is now $(X_T - \bar{X}_T)/\bar{X}_T$ at the end of the first cycle. The correct value is 0 everywhere. In the right image, the pointwise relative error is a little better than $10^{-11}$, consistent with the $l_\infty$-norm diagnostic value for $n_p^t = 8$ at the end of the first cycle in Fig. 23.

## 6  Performance results

### 6.1  Communication volume

Following the discussion in Sect. 4.3.2, Figure 26 plots the number of real scalars transmitted in $q$-messages per tracer per element per time step ($y$-axis) vs. time in days of the simulation ($x$-axis), in the case of one element per process. The configurations use nondivergent flow, $1°$ resolution, long and short time steps, and $n_p^t \in \{4, 8, 12\}$. Statistics include the maximum (dashed line) and median (solid line) over all target elements. A dotted horizontal line provides the reference value $(n_p^t)^2$. As discussed in Sect. 4.3.2, because the volume of extrema data is almost independent of $n_p^t$, the measured volume decreases relative to the reference $(n_p^t)^2$ with increasing $n_p^t$.

### 6.2  GPU performance

E3SM Atmosphere Model's (EAM) nonhydrostatic dynamical core, HOMME (Dennis et al., 2005, 2012; Taylor et al., 2020), has been ported to C++ to run on GPU-based supercomputers (Bertagna et al., 2020), and performance results are reported in Bertagna et al. (2020) for the NGGPS benchmark (Whitaker, 2016; Michalakes et al., 2016). At that time, only the original Eulerian tracer transport module was available. In related work, a quasiuniform 3.25km convection-permitting configuration of EAM, called SCREAM, has been developed (Caldwell et al., 2021); this work uses SL transport with $n_p^v = n_p^t = 4$. Other



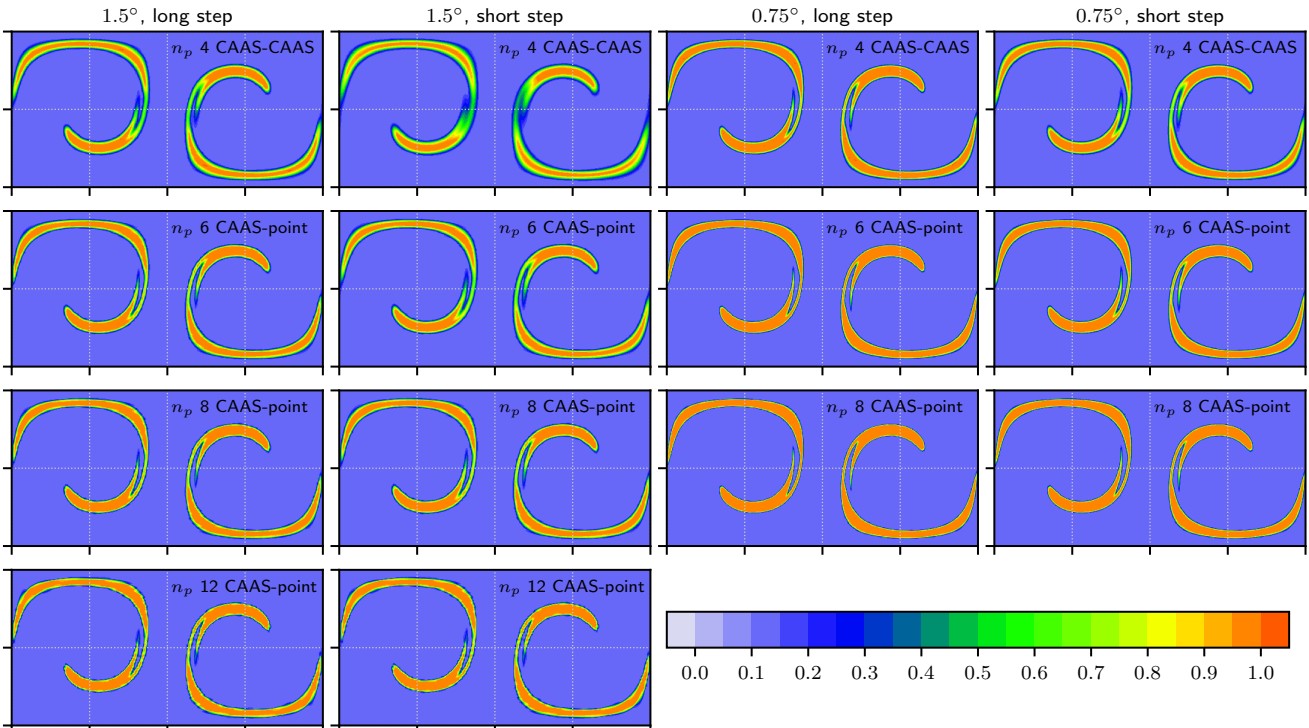

**Figure 20.** Images of the slotted cylinders IC advected by the nondivergent flow at the simulation's midpoint. Each column corresponds to a spatial resolution and time step length configuration, as stated at the top of each column. Each row corresponds to a particular value of $n_p^t$, as stated in the text at the top-right of each image. We omit $n_p^t = 12$ results for the $0.75°$ resolution because they are essentially identical at the resolution of the figure to the $n_p^t = 8$ images.

than this difference, Bertagna et al. (2020) and Caldwell et al. (2021) use the same dycore configurations. We omit description of these configuration details because they are not relevant to this article; see the cited articles for details.

Our SL transport implementation in HOMME does not yet permit $n_p^t > n_p^v$, but otherwise it follows, in particular, the communication procedures described in Sect. 4.3. We re-ran the benchmark in Bertagna et al. (2020) with SL transport configured as in Caldwell et al. (2021) and report results in Fig. 27. The $x$-axis is number of NVIDIA V100 GPUs on the Oak Ridge

Leadership Computing Facility's Summit supercomputer used in a run; Summit has just over 27,600 V100 GPUs. The $y$-axis is dycore throughput reported in simulated years per wallclock day (SYPD). Some data points have SYPD numbers above the point for additional precision.

First, we established that we could roughly reproduce the results in Bertagna et al. (2020). The black dotted line shows the data we obtained using Eulerian transport; compare this curve with the data from Bertagna et al. (2020), plotted as the black

solid line with circular markers. Our results are a little less than 10% slower, which is acceptable given different compiler and MPI versions, different job profiles running at the same time, and a slightly different code base. We then switched to using SL



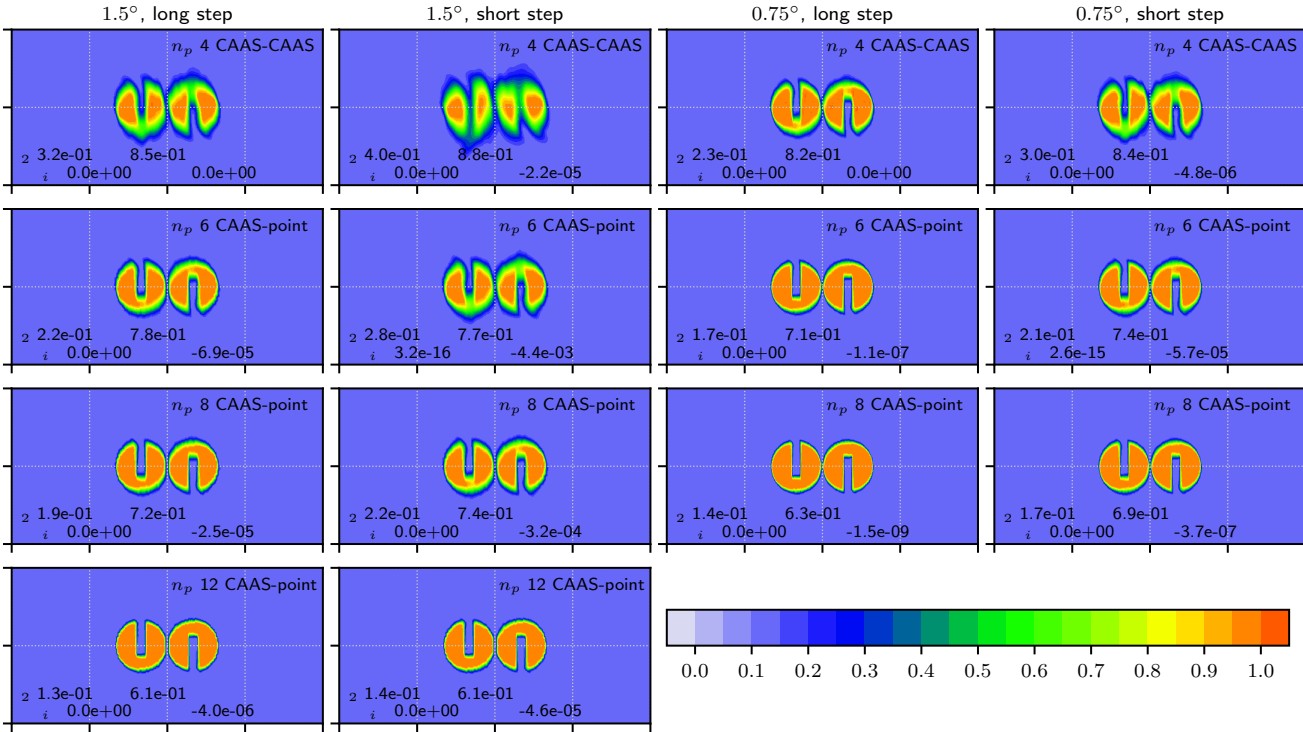

**Figure 21.** Same as Fig. 20 but for the simulation final point. Error measures are printed at the bottom-left of each image; see text for details.

transport and obtained the solid red line with square markers. These simulations used ten tracers. We repeated them with 40 tracers, the number used in E3SMv1 and v2 models; the corresponding curves are dashed. Comparing the new SL data with the data in Bertagna et al. (2020), SL transport speeds up the nonhydrostatic dycore by a factor of over 1.4 when 10 tracers are

transported and a factor of 2.8 when 40 tracers are transported.

# 7  Conclusions

We have presented a set of compact bases, the Islet bases, for very accurate interpolation semi-Lagrangian element-based tracer transport. We then described algorithms to support a three-grid atmosphere model, one with a shared element grid but separate subelement grids for physics parameterizations, dynamics, and tracer transport using the Islet bases. This configuration permits

the modeler to create a dynamics grid with a tolerable CFL-limited time step independent of the other two subcomponents, while physics parameterizations and tracer transport can run at resolutions potentially substantially higher.

Future work includes the following. First, in Sect. 3.2 we described an infinite set of potential bases, but then in Sect. 3.3 we restricted our attention to a finite set of these. It is possible that there are much better bases outside of the subset we considered. Searching for these is one line of work.



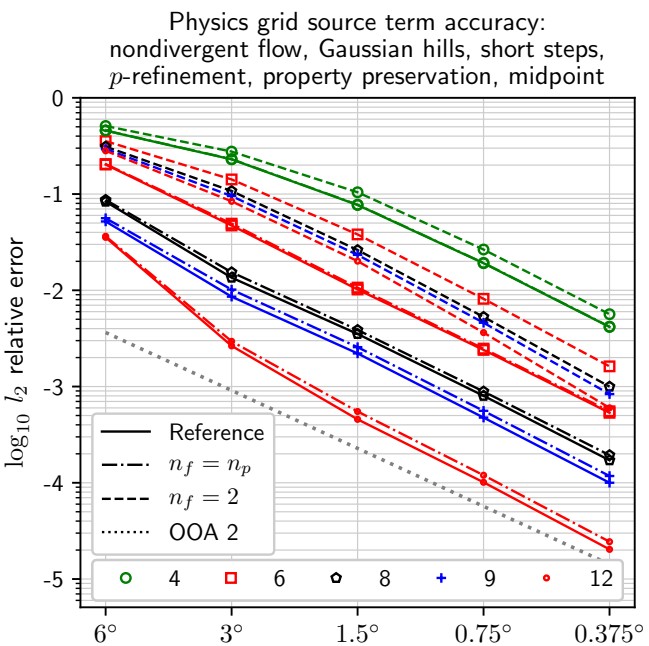

**Figure 22.** Validation of the remap of tendencies from physics to tracer grids and state from tracer to dynamics grids. See Sect. 5.4 for a description of the problem.

Second, because all operations except the DSS are either local to the element or act on element-level scalars, both the tracer and the physics grids permit various types and subsets of spatially and temporally adaptive and tracer-dependent refinement and derefinement, possibly in combination with already existing E3SM regionally refined models (RRM). In the vertical direction, we believe there is an opportunity to combine Islet with FIVE (Yamaguchi et al., 2017), or a method like it, to transport highly resolved vertical data while still running the dynamics solver on the coarser background vertical grid.

Third, it is possible to recover local mass conservation in an ISL method by modifying the space-time operator's coefficients so that the sum of mass over all target points associated with a source element is consistent with the source element's total mass (Kaas, 2008). This optional coefficient modification step is compatible with the Islet method and should be explored.

Finally, we predict that applications related to, in particular, aerosols will benefit from the Islet method. In future work, we intend to integrate the Islet method into the E3SM Atmosphere Model to investigate its impact on science applications.

*Code and data availability.   Current URLs are preliminary; this section will be finalized with Zenodo URLs if the article is accepted.* Code and scripts for the algorithms and figures presented in this paper are available at https://github.com/E3SM-Project/COMPOSE/releases/tag/ v1.1.1. In this repository, read `methods/islet/readme.txt` for further instructions. Data for the figures will be uploaded to GMD. Code for the Summit performance results is available at https://github.com/ambrad/E3SM/releases/tag/islet-2d-paper-summit-sl-gpu-timings,



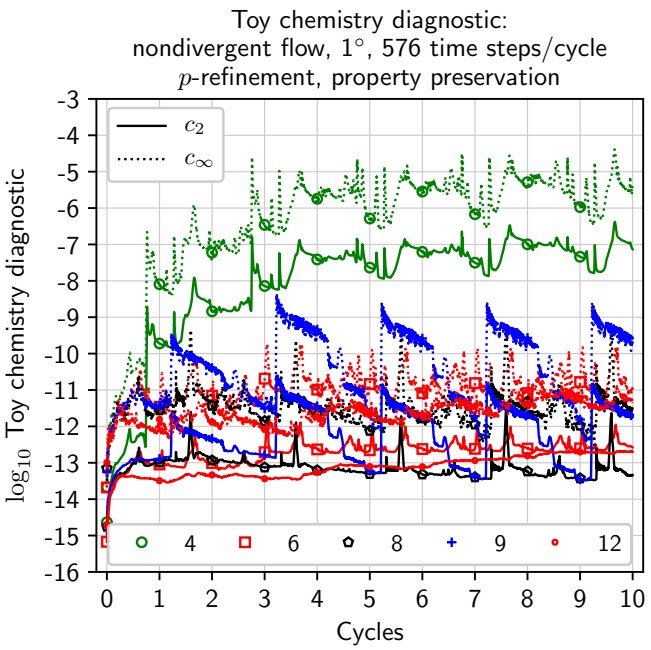

**Figure 23.** Toy chemistry diagnostic values as a function of time for ten cycles of the nondivergent flow. Time is on the $x$-axis and measured in cycles. Diagnostic values $c_2$ (solid lines) and $c_\infty$ (dashed lines) are on the $y$-axis. Markers as listed in the bottom legend are placed at the start of each cycle to differentiate the curves.

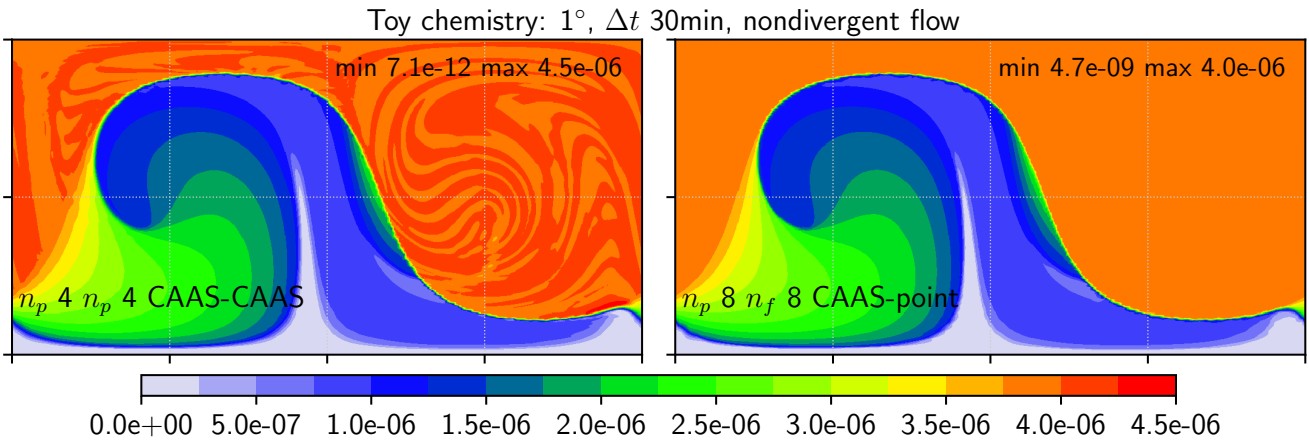

**Figure 24.** Images of the monatomic tracer at the end of the first cycle. Text at the lower left of each image states the configuration. Text at the upper right reports global extremal values.



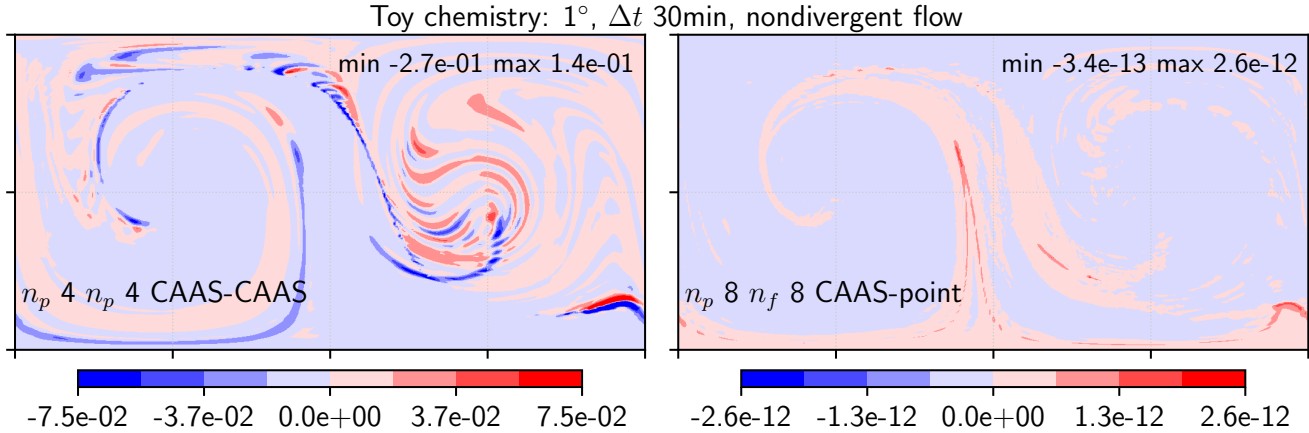

**Figure 25.** Same as Fig. 24, but now the images are of $(X_T - \bar{X}_T)/\bar{X}_T$.

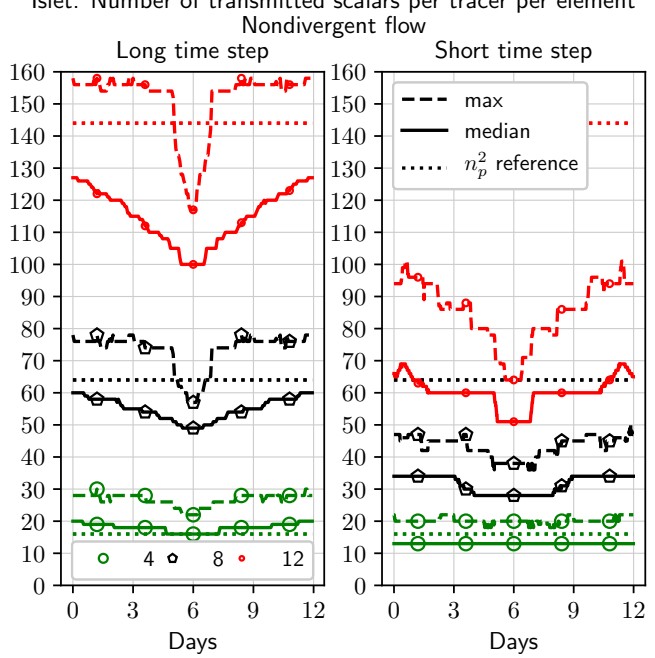

**Figure 26.** Communication volume, in number of real scalars transmitted in $q$-messages per tracer per element per time step ($y$-axis) vs. time in days of the simulation ($x$-axis), in the case of one element per process, for the nondivergent flow, with long (left) and short (right) time steps. Statistic and $n_p^t$ line patterns are stated in the legends.

principally in the directory `components/homme/src/share/compose`. Data and scripts for the Summit performance results are available at https://github.com/E3SM-Project/perf-data/tree/main/nhxx-sl-summit-mar2021.



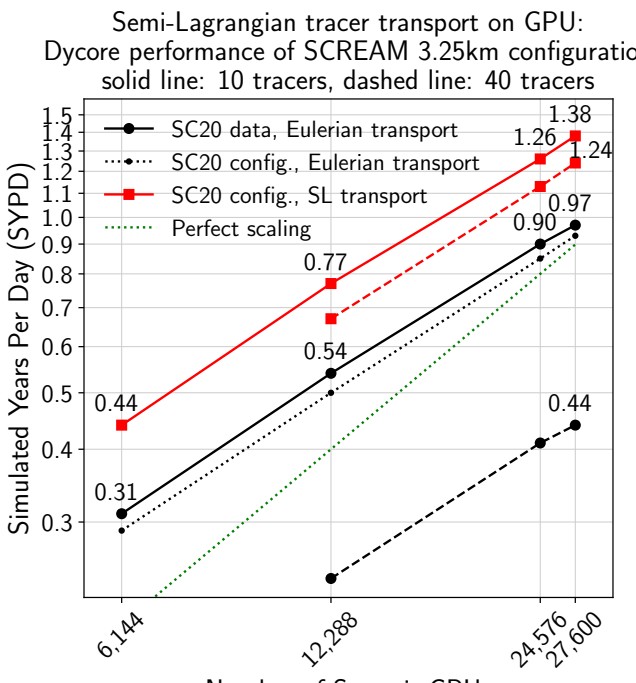

**Figure 27.** Performance comparison of SL transport with $n_p^{\mathrm{v}} = n_p^{\mathrm{t}} = 4$ vs. Eulerian transport in the E3SM Atmosphere Model's dynamical core on the Summit supercomputer. The $x$-axis is number of NVIDIA V100 GPUs on Summit used in a run; the $y$-axis is dycore throughput reported in simulated years per wallclock day (SYPD). The black curves are for Eulerian transport; the red, for SL. Dashed lines are for 40 tracers; solid and the dotted black line, for 10. A number above a data point reports the $y$-value of that point.

## Appendix A: Maximum eigenvalue magnitude

Let an eigenvector of $\mathbf{A}$ defined in Eq. (1) be $\boldsymbol{V}$, and write $\boldsymbol{V} \equiv \left( \boldsymbol{v}_0^T \quad \cdots \quad \boldsymbol{v}_{N-1}^T \right)^T$, where $d$-vector $\boldsymbol{v}_k$ is $d$-length block row $k$ of $\boldsymbol{V}$, $k \in \{0, \ldots, N-1\}$. We suppose $\boldsymbol{v}_k = \mu^k \boldsymbol{q}$, $\mu = e^{i2\pi j/N}$, $j \in \{0, \ldots, N-1\}$, $i$ the imaginary unit, and $\boldsymbol{q}$ an unknown $d$-vector.

Let $r \equiv (r' + \rho d) \bmod (Nd)$ for $\rho \in \{0, \ldots, N-1\}$. We need to establish some indices and permutations. Let $\bar{r} \equiv \lfloor r/d \rfloor$, $\hat{r} \equiv r \bmod d$ and

$$\mathbf{S}_r^\mu \equiv \begin{pmatrix} \mathbf{0} & \mathbf{I}^{(d-\hat{r}) \times (d-\hat{r})} \\ \mu \mathbf{I}^{\hat{r} \times \hat{r}} & \mathbf{0} \end{pmatrix}.$$





Now consider rows $(r + (0:d-1)) \bmod (Nd)$ of the eigenvalue problem $\mathbf{A}\boldsymbol{V} = \lambda \boldsymbol{V}$, and assume $\boldsymbol{v}_k = \mu^k \boldsymbol{q}$ as described above. The left hand side can be written

$$\mathbf{A}((r + (0:d-1)) \bmod (Nd),:)\boldsymbol{V}$$
$$= \mathbf{B}\boldsymbol{V}((\rho d + (0:d)) \bmod (Nd))$$
$$= \mathbf{B}\mu^\rho \begin{pmatrix} \mathbf{I}^{d \times d} \\ \mu \boldsymbol{e}_0^T \end{pmatrix} \boldsymbol{q}.$$

The right hand side can be written

$$\lambda \boldsymbol{V}((r + (0:d-1)) \bmod (Nd))$$
$$= \lambda \mu^{\bar{r}} \begin{pmatrix} \boldsymbol{q}(\widehat{r}:d-1) \\ \mu \boldsymbol{q}(0:\widehat{r}-1) \end{pmatrix}$$
$$= \lambda \mu^{\bar{r}} \mathbf{S}_r^\mu \boldsymbol{q}.$$

Together, these give the $d \times d$ eigenvalue problem

$$\mu^{-\lfloor r'/d \rfloor}(\mathbf{S}_r^\mu)^H \mathbf{B} \begin{pmatrix} \mathbf{I}^{d \times d} \\ \mu \boldsymbol{e}_0^T \end{pmatrix} \boldsymbol{q} = \lambda \boldsymbol{q}, \tag{A1}$$

where $H$ is the matrix conjugate transpose and we have used $\mu^{\rho - \bar{r}} = \mu^{-\lfloor r'/d \rfloor}$. The eigenvalues of $\mathbf{A}$ are the union of those
obtained by solving Eq. (A1) for each $\mu = e^{i2\pi j/N}$, $j = 0, \ldots, N-1$.

If every value of $N$ is of interest, then $\mu = e^{i\theta}$, $\theta \in [0, 2\pi)$. Thus, to compute the maximum eigenvalue of $\mathbf{A}$ independent of $N$, we solve the problem

$$\lambda_{\max}(\mathbf{B}) \equiv \max_{\substack{\theta \in [0, 2\pi) \\ i \in \{0, \ldots, d-1\}}} |\lambda_i|$$
$$\text{subject to } (\mathbf{S}_r^{e^{i\theta}})^H \mathbf{B} \begin{pmatrix} \mathbf{I}^{d \times d} \\ e^{i\theta} \boldsymbol{e}_0^T \end{pmatrix} \mathbf{Q} = \boldsymbol{\Lambda} \mathbf{Q},$$

where $\boldsymbol{\Lambda}$ is the diagonal matrix of eigenvalues $\lambda_i$, $i \in \{0, \ldots, d-1\}$. We omit $\mu^{-\lfloor r'/d \rfloor}$ from the eigenvalue problem because it does not affect an eigenvalue's magnitude. In practice we must limit the search to a discrete, but possibly large, subset of $[0, 2\pi)$. This problem is useful because it requires solving many separate $d \times d$ eigenvalue problems instead of one or more (since we are interested in arbitrary $N$) $(Nd) \times (Nd)$ problems. Thus, first, one can program a work- and parallel-efficient method to check a particular discretization for empirical satisfaction of the necessary condition; second, we avoid the superlinear growth
in the cost of the eigenvalue computation with $N$.





## Appendix B: Interpolants

In this appendix, we construct expressions for an infinite set of potential bases. In this article, we restricted our search to two finite subsets of these.

1. Given a departure point $x$, the interpolant is a linear operator in the values $\boldsymbol{y}$ at basis points $\boldsymbol{x}$. Anticipating additional constraints, we begin by writing this interpolant in the form

$$f(\boldsymbol{y};x) \equiv \mathcal{L}(x;\mathbf{M}^{n\times n}(x)\boldsymbol{y},\boldsymbol{x}^n).$$

2. Within each region, $M(x)$ is constant:

$$\mathbf{M}(x) = \sum_{r=0}^{d-1}\mathrm{Hat}(x;\boldsymbol{x}(r),\boldsymbol{x}(r+1))\,\mathbf{A}_r^{n\times n},$$

where

$$\mathrm{Hat}(x;a,b) \equiv \begin{cases} 1 & \text{if } a \leq x \leq b \\ 0 & \text{else.} \end{cases}$$

We call each $\mathbf{A}_r$ a *region operator*.

3. $f$ must recover a degree-$p$ polynomial with specified $p = s - 1$ and $2 \leq s \leq n$. This *order constraint* implies

$$\mathbf{A}_r\begin{pmatrix} \mathbf{I}^{s\times s} \\ \mathbf{B}^{(n-s)\times s} \end{pmatrix} = \begin{pmatrix} \mathbf{I}^{s\times s} \\ \mathbf{B}^{(n-s)\times s} \end{pmatrix},$$

where $\mathbf{B}^{(n-s)\times s} \equiv \mathcal{L}(\boldsymbol{x}(s:n-1);\mathbf{I}^{s\times s},\boldsymbol{x}(0:s-1))$.

4. $f$ must interpolate nodes $r, r+1$. This *region interpolation constraint* implies

$$\mathbf{A}_r([r,r+1],:) = \begin{pmatrix} \mathbf{0}^{2\times r} & \mathbf{I}^{2\times 2} & \mathbf{0}^{2\times(n-r-2)} \end{pmatrix}.$$

5. Consider the $n$ basis functions $\phi_i(x) \equiv f(\boldsymbol{e}_i;x)$. These must have the symmetry $\phi_i(x) = \phi_{n-1-i}(-x)$. This *symmetry constraint* implies, first, $\boldsymbol{x}$ has points symmetric around 0; second,

$$\mathbf{A}_r = \mathbf{A}_{n-1-r}(n-1:-1:0,n-1:-1:0).$$

For a non-middle-region operator, $\mathbf{A}_r$ starts with $n^2$ degrees of freedom, the order constraint removes $ns$, and the region interpolation constraint removes $2(n-s)$ more since $2s$ degrees of freedom are shared with the order constraint. This leaves $n^2 - ns - 2(n-s) = (n-2)(n-s)$ degrees of freedom. The symmetry constraint implies that a middle-region operator $\mathbf{A}_r$ has half of these: $(n-2)(n-s)/2$.

We can devise a basis $\mathcal{N}_{n_p}$ by selecting values for node points $\boldsymbol{x}^{n_p}$ and the region operators $\mathbf{A}_r$, subject to constraints 1–5.



*Author contributions.* AMB developed the algorithms, wrote the software, ran the numerical studies, and wrote the manuscript, all with contributions from PAB and OG. PAB administers the project that funded most of this work.

*Competing interests.* The authors declare that they have no conflict of interest.

*Disclaimer.* Sandia National Laboratories is a multimission laboratory managed and operated by National Technology and Engineering Solutions of Sandia, LLC., a wholly owned subsidiary of Honeywell International, Inc., for the U.S. Department of Energy's National Nuclear
Security Administration under contract DE-NA-0003525. This paper describes objective technical results and analysis. Any subjective views or opinions that might be expressed in the paper do not necessarily represent the views of the U.S. Department of Energy or the United States Government. SAND2021-10514 O.

*Acknowledgements.* We thank Mark A. Taylor for valuable discussions regarding material in this article.

This work was supported by the US Department of Energy (DOE) Office of Science's Advanced Scientific Computing Research (ASCR)
and Biological and Environmental Research (BER) Programs under the Scientific Discovery through Advanced Computing (SciDAC 4) ASCR/BER Partnership Program, and by the Energy Exascale Earth System Model (E3SM) project, funded by BER. This research used resources of the Oak Ridge Leadership Computing Facility at the Oak Ridge National Laboratory, which is supported by the Office of Science of the U.S. Department of Energy under Contract No. DE-AC05-00OR22725.





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
