# Peer review of "Islet: Interpolation semi-Lagrangian element-based transport"

_Geoscientific Model Development, 2021_

## Author Comment (AC1)

Dr. Añel:

Thank you for your instructions. I have made Zenodo DOIs for each of the repositories:

1. Methods code:

   - `https://github.com/E3SM-Project/COMPOSE/releases/tag/v1.1.2`
   - Zenodo: `https://doi.org/10.5281/zenodo.5595499`

2. Model code for performance results:

   - `https://github.com/ambrad/E3SM/releases/tag/islet-2d-paper-summit-sl-gpu-timings`
   - Zenodo: `https://doi.org/10.5281/zenodo.5595508`

3. Performance data and scripts:

   - `https://github.com/E3SM-Project/perf-data/releases/tag/islet-gmd-v0`
   - Zenodo: `https://doi.org/10.5281/zenodo.5595501`

4. Data used in the figures. These data can be recreated by following the instructions in resource 1, "Methods code", in particular files `methods/islet/readme.txt` and `methods/islet/figures/figs.tex`.

   - Zenodo: `https://doi.org/10.5281/zenodo.5595518`

In the case of resource 1, "Methods code", the URL and Zenodo DOI are for an updated version of the repo. Relative to the version at submission, I generalized the Makefile for the slmmir program to build with both Intel and GNU compiler suites and updated the readme.txt file accordingly, and I modified some data generation scripts to make output directories before running.

Thank you for bringing to my attention the broken hyperlink. In the final version of the paper, I will be sure to check that each hyperlink works correctly.

Thanks,
Andrew Bradley

---

## Author Comment (AC2)

Reviewer 1:

Thank you for a close reading of the manuscript and many valuable suggestions to clarify the presentation of the material. In this response, I focus on the technical points and questions you raised.

In the following, the reviewer's comments are italicized.

RC1: *Figures 20 and 21 shows larger diffusion for shorter time steps. Could you elaborate on this? Is the traditional SL property that longer time steps reduce diffusion (while the solution can degrade in other metrics)? How would one choose the right time step in practice?*

Yes, generally in SL methods, longer time steps reduce dissipation. It is simplest to understand this in remap-form methods. In remap-form methods, the only source of dissipation is in the remap step, which occurs once per time step. Thus, as the time step is decreased, the dissipation increases. One sees this, for example, in the SL methods in [3] that provide data for two different time steps: CSLAM and FARSIGHT.

Semi-Lagrangian passive tracer transport is simpler than semi-Lagrangian dynamical cores. Importantly, long trajectories are possible because the velocity data are provided by the dynamics. Thus, the relatively inexpensive transport trajectory algorithm can be substepped, if necessary, relative to the expensive tracer transport remap step. In this case, the advection algorithm can be robust over a large range of time steps.

When using a method that is robust over a large range of time steps, the practical time step limit is determined by the simulator in which the transport method is embedded. For example, the transport step cannot exceed the maximum permitted physics parameterization time step, since updated tracer values must be provided to the physics parameterizations. In addition, details of the dynamical core may require additional transport-dynamical-core coupling computations, e.g., to exceed a dynamical core's vertically Lagrangian vertical remap time step. Finally, selecting multiple subcomponents' time steps while satisfying substepping divisibility constraints influences final time step selection. Practical experience in E3SMv2 and SCREAM [2] with a method very close to the $n_p^t = n_p^v = 4$, $n_f = 2$ Islet method shows that a time step 5–8 times longer than the dynamics time step satisfies the various constraints. For example, in the E3SMv2 standard and Regionally Refined Model (RRM) resolutions, the factor is 6; it is 8 in [2]; and it is 5 in some other SCREAM simulations due to divisibility constraints and balancing other time steps. The Islet method with other parameter values will maintain this range of factors.

RC1: *l631: why can you not use the dynamical core to compute the density? the chosen approach seems rather ad-hoc; can you guarantee that density values are realistic?*

Thank you for this clarifying question. I should have provided more details in the manuscript. The density factor is described in the text surrounding equation 32 of reference [1]. In the manuscript, its discretization follows from approximation of the integrals in that equation by $n_p = 4$ GLL quadrature (since $n_p = 4$ is used for the total mass density field $\rho$); with this quadrature scheme, the basis function nodal values are 0 except at one quadrature point. The resulting discretization has order of accuracy 2. The experiments are run with a standalone program that is not part of the dynamical core. In addition, to run many long experiments, we need a fast method to compute density; thus, an interpolation semi-Lagrangian method makes sense.

Thanks,
Andrew Bradley

**References**

[1] P. A. Bosler, A. M. Bradley, and M. A. Taylor. Conservative multimoment transport along characteristics for discontinuous galerkin methods. *SIAM J. on Sci. Comput.*, 41(4):B870–B902, 2019.

[2] P. M. Caldwell, C. R. Terai, B. R. Hillman, N. D. Keen, P. A Bogenschutz, W. Lin, H. Beydoun, M. A. Taylor, L. Bertagna, A. M. Bradley, et al. Convection-permitting simulations with the e3sm global atmosphere model. *J. Adv. Model Earth Sy.*, page e2021MS002544, 2021.

[3] P. H. Lauritzen, P. A. Ullrich, C. Jablonowski, P. A. Bosler, D. Calhoun, A. J. Conley, T. Enomoto, L. Dong, S. Dubey, O. Guba, et al. A standard test case suite for two-dimensional linear transport on the sphere: results from a collection of state-of-the-art schemes. *Geosci. Model Dev.*, 7(1):105–145, 2014.

---

## Author Comment (AC3)

Reviewer 2:

Thank you for your questions. In the following, the reviewer's comments are italicized.

RC2: *(1) The stability associated with the SL method is that the deformational Courant number (Lipschitz condition) should not exceed unity, in plain language, the trajectories should not cross intersect (see, Staniforth & Cotes 1992 MWR paper). Is the cubic ISL method (lines 115-120) unstable due to this condition? Need some explanation.*

No. In the first example, lines 118–122, the flow is uniform in space and constant in time. Thus, the trajectories are parallel lines and so cannot intersect. In addition, the deformational Courant number is 0 because the flow is uniform in space.

The second example, described in lines 123–127, is more complicated. We analyze this case in Appendix A of this letter.

RC2: *(2) The SL transport scheme can be stabilized using a limiter, filter or with an explicit diffusion (see, Ullrich & Norman, QJRMS, 2014). You can use the native high-order SE interpolation (basis function) for the SL transport combined with the limiter which you are already using for the Islet method. It will be interesting to see how the Islet method compares with this simple SL-SE scheme employing 4x4 GLL grid (I guess that is the SE grid choice made for the operational E3SM).*

A limiter or filter makes a linear discretization nonlinear. A method that assures bounded-ness of the solution leads to stability because the solution is bounded in norm. However, for a nonlinear discretization, consistency and stability are not enough to assure convergence. In contrast, our method satisfies a necessary condition for stability of a linear discretization; because the discretization is linear, consistency and stability assure convergence.

The dash-dotted curves in Figure 6 of the manuscript illustrate what happens when "*the native high-order SE interpolation (basis function) for the SL transport [is] combined with the limiter.*" The curves for $n_p^{\mathrm{t}} > 4$ diverge within one cycle of the test problem. The $n_p^{\mathrm{t}} = 4$ case only starts to diverge, but with multiple cycles or greater refinement, the divergence continues to be as substantial as in the $n_p^{\mathrm{t}} > 4$ cases. This divergence is a result of the fact that the linear discretization's maximum eigenvalue magnitude is above 1 at almost every value of $\Delta t$, as illustrated by the red curve in Figure 2 of the manuscript for the case $n_p = 8$.

We discuss the reference Ullrich & Norman, 2014 [6], in Appendix B of this letter.

RC2: *(3) It is not convincing to have 3 grid systems (physics: FV, dynamics: GLL, transport: tweaked GLL) in a SE modeling framework. The Fig.5 shows such a grid configuration, and it appears to be very challenging. At a very high (NH) resolution the data movement is a major issue for an element-based Galerkin model (DG/SE). A typical climate model may have O(100) tracers, an additional tracer grid with more DOF than the dynamic grid can exacerbate this problem. This will limit the use of Islet scheme, how do you address it?*

The Energy Exascale Earth System Model version 2 (E3SMv2) uses a two-grid system as a result of work described in [1]. Semi-Lagrangian transport, with $n_p^{\mathrm{v}} = n_p^{\mathrm{t}} = 4$, and this two-grid system, with $n_f = 2$, together make the E3SMv2 Atmosphere model twice as fast as E3SMv1. Figure 1 shows a representative strong-scaling study comparing version 2 to version 1. I believe the success of this two-grid system motivates further work along these lines.

Regarding many tracers, in the case of $n_p^{\mathrm{t}} = n_p^{\mathrm{v}} = 4$, Figure 27 of the manuscript is illustrative. This figure shows dynamical core performance for the full 3D method on a 3.25km cubed-sphere grid having 128 model levels. The throughput of the dynamical core with SL transport is only slightly decreased when going from 10 tracers to 40 (1.38 to 1.24 SYPD on 27,600 Summit V100 GPUs), while the throughput of the dynamical core with Eulerian transport is decreased substantially (0.97 to 0.44 SYPD with the same number of GPUs). Section 4.3 of the manuscript discusses

[Figure]

Figure 1: Performance of the Energy Exascale Earth System Model (E3SM) version 2 Atmosphere Model (EAMv2) on the ANL LCRC Chrysalis cluster. (a) Strong-scaling study of the standard-resolution version 1 (v1, blue) and version 2 (v2, red) models. Each model has 40 tracers. Tracer transport is over 6 to over 8 times faster than in v1 due to use of semi-Lagrangian transport in v2. The overall model ("Total") is 2 to 2.5 times faster than v1, with most of the remaining speedup due to the use of separate physics parameterizations and dynamics grids. (b) Proportion of time spent in key subcomponents, with the total v1 time normalized to 1, for the models run on 85 Chrysalis nodes. Tracer transport, red with circles, is sped up by almost 6.5. The green region encapsulates the part of the model that uses the physics grid; it is decreased by a factor 1.8 in v2.

the communication details for $n_p^{\mathrm{t}} > n_p^{\mathrm{v}}$. Importantly, only the fundamental step of the method,
obtaining interpolated values, scales in communication volume proportionally to $(n_p^{\mathrm{t}})^2$; a scaling of
this sort is true of any method as basis order increases. Equally importantly, there are no additional
communication rounds when $n_p^{\mathrm{t}} > n_p^{\mathrm{v}}$.

RC2: *(4) With real data you have velocity information only available at the GLL (dynamics)*
*grid, the way you find the 2D trajectory information using the 3D Cartesian coordinates leads to*
*additional computational overhead when the method is extended to the 3D application (line 470-*
*475). This needs some justification, why not use the spherical (u,v) components or corresponding*
*contravariant vectors?*

Figure 27 of the manuscript shows results for the full 3D application, and Figure 1 of this letter
also uses the full 3D solver. $(u, v)$ should not be used because of the poles. Contravariant compo-
nents can be used but require details to handle coordinate systems between elements. Cartesian
coordinates avoid these details at the cost of one extra variable. However, the trajectory compu-
tation is a negligible part of the overall transport step even at just 10 tracers, making this extra
variable also negligible. Finally, as either or both of $n_p^{\mathrm{t}}$ and tracer count increase, the relative cost
of the trajectory computation decreases.

RC2: *It is not clear that the maximum eigenvalue required for the interpolation is the tracer data*
*dependent, in that case you have a serious computational overhead for the multi-tracer applications,*
*Please clarify!*

The maximum eigenvalue computation is used only in deriving basis sets. It is not used in a
simulation. I am sorry that this point was not clear in the manuscript.

RC2: *What is the computational halo requirement for an SE stencil with NxN GLL points, when the shape preserving limiter is applied?*

The limiter requires extrema data from adjacent elements. The extrema data per adjacent element is two scalars, independent of $n_p^{\text{t}}$. Thus, the relative cost decreases with increasing $n_p^{\text{t}}$. This communication pattern is the same as for the original Eulerian SE method used in the SE dynamical core.

RC2: *(5) What is the special advantage of using Islet method? It seems you have introduced a complex numerical method for a relatively simple linear transport problem.*

Although the transport problem may be simple, it can have a large computational cost in Earth system models because of the large number of tracers. Thus, this work focuses on improving computational efficiency, like other projects that develop tracer transport methods for passive tracers, e.g., those in [3]. Computational efficiency is the ratio of a measure of solution accuracy to a measure of computational work. The Islet method combines a very efficient class of SL methods, the remap-form interpolation class, with details specific to element-based methods. It provides two parameters, $n_p^{\text{t}}$ and $n_f$, to efficiently trade between accuracy and speed. Importantly, it does not require the dynamics solver to be modified, regardless of parameter values, making it possible to tune tracer transport parameters without also having to modify the dynamics solver.

Similar reasons led to the work to separate the physics and dynamics grid [2, 1]. In both these papers, additional complexity was introduced to transfer data between grids. But the result was an increase in computational efficiency in each Earth system model.

RC2: *If mass-fixing is the way to go, one could use the RBF-based (Kriging type) interpolator which provides very accurate solution, and no need for the expensive search for max eigenvalue etc.*

Interpolation methods require careful stability analysis. This is in contrast to exactly integrated $L^2$ projection methods, in which the projection provides stability. Interpolation methods can be substantially more efficient than projection methods, justifying the effort to do this analysis and to derive methods satisfying linear stability conditions. Because the Islet bases provide element-local interpolation, the stability analysis can focus on the element. An interpolation method that uses data beyond a single element would require mesh-dependent stability analysis. Its implementation would require greater communication volume than an element-local method. An RBF-based method may be effective, but work would need to be done to analyze and implement it. The manuscript focuses on an element-local interpolation basis.

Regarding the "expensive search for max eigenvalue," I am sorry that it was not clear in the manuscript that this search is done only to find basis functions. Once the basis function sets are found based on analysis of an element, e.g. those in Table 1, they can then be used directly in simulations like any other basis function set.

Thanks,
Andrew Bradley

**Appendix A**

By equation A6 of [5], two computed trajectories, which are line segments, intersect when there is a point in space-time at which these computed trajectories have the same position. A bound of one on deformational Courant number is a sufficient condition to assure that these computed trajectories do not cross when they are computed according to certain algorithms. In addition, the deformational Courant number is used in, e.g., [5, 4] to bound the time step when solving iteratively for the departure points. In the examples I provide in section 2.1, trajectories are exact, and thus no condition governing the solution of the departure point is needed. Importantly, given exact trajectories, the deformational Courant number is not a means to assess stability of the space-time operator, as demonstrated in the first example.

In the second example, the flow is shear and constant in time. The deformational Courant number is $\max_{x,y} \max(|u_x|, |u_y|, |v_x|, |v_y|)\Delta t = 2\pi\Delta t$; thus, with $\Delta t = 0.2761$, the number exceeds 1. However, trajectories in this flow cannot cross, in two senses, as follows. First, the exact trajectories do not cross. Second, line segments connecting arrival points to exact departure points do not cross. The second follows from the first, which we shall demonstrate in a moment, because the exact trajectories in this flow are lines. For in this flow, (i) velocities are constant along lines, (ii) a velocity vector points along its constant-velocity line, and (iii) these lines of constant velocity are parallel to each other. Consider any two points in the flow and a time increment $\Delta t$. By (ii) a point must stay on the line on which it started. By (i) a point on a line cannot overtake another point on that line. Finally, by (iii), if the two points start on separate lines, since their lines do not cross, the trajectories cannot, either. For this flow, we should also verify that each departure element is neither self-intersecting nor has reversed in orientation, both of which are possible in a discretization even if the continuum flow does not permit intersecting trajectories. I verified using orientation and convexity checks, where the latter is a sufficient condition for non-intersection, that no deformed element has reversed orientation or is self-intersecting. Finally, whereas the time step 0.2761 in this example leads to an unstable operator, time steps 0.273 and 0.279 do not.

Between the two examples, we see that a unit bound on the deformational Courant number, given exact trajectories and so exact departure points, is neither a necessary nor a sufficient condition for stability of the space-time operator. Similarly, the violation of the bound is neither a necessary nor a sufficient condition for instability.

**Appendix B**

Ullrich and Norman present a scheme that is third-order accurate when a monotonicity scheme is not applied, requires hyperdiffusion for stability in 2D, and is CFL-limited by hyperdiffusion [6]. The left panel of Figure 11 of [6] suggests that if a positivity filter is applied, the method has an empirical order of accuracy of approximately 2.5. Positivity—the tracer must have values at least 0—is a weaker condition than monotonicity, which requires that a tracer value in a region $R$ is bounded below and above by values in the domain of dependence of the region $R$, where the definition of $R$ depends on the discretization. In the manuscript, results for the Islet method are shown with a monotone filter rather than just a positivity filter. Hyperdiffusion requires additional communication rounds. As the basis order increases, the hyperdiffusion-limited CFL number roughly decreases, from 2.44 for cubic to below 1 for quintic. Islet does not require a filter or hyperdiffusion for stability, regardless of basis order, and we find bases providing order of accuracy up to 9.

**References**

[1] W. M. Hannah, A. M. Bradley, O. Guba, Tang Q., J.-C. Golaz, and W. Wolfe. Separating physics and dynamics grids for improved computational efficiency in spectral element earth system models. *J. Adv. Model Earth Sy.*, 13(7):e2020MS002419, 2021.

[2] A. R. Herrington, P. H. Lauritzen, K. A. Reed, S. Goldhaber, and B. E. Eaton. Exploring a lower resolution physics grid in CAM-SE-CSLAM. *J. Adv. Model Earth Sy.*, page 2019MS001684, 5 2019.

[3] P. H. Lauritzen, P. A. Ullrich, C. Jablonowski, P. A. Bosler, D. Calhoun, A. J. Conley, T. Enomoto, L. Dong, S. Dubey, O. Guba, et al. A standard test case suite for two-dimensional linear transport on the sphere: results from a collection of state-of-the-art schemes. *Geosci. Model Dev.*, 7(1):105–145, 2014.

[4] J. Pudykiewicz, R Benoit, and A. Staniforth. Preliminary results from a partial LRTAP model based on an existing meteorological forecast model. *Atmosphere-ocean*, 23(3):267–303, 1985.

[5] Piotr K. Smolarkiewicz and Janusz A. Pudykiewicz. A class of semi-Lagrangian approximations for fluids. *Journal of Atmospheric Sciences*, 49(22):2082–2096, 1992.

[6] Paul A. Ullrich and Matthew R. Norman. The flux-form semi-Lagrangian spectral element (FF-SLSE) method for tracer transport. *Quarterly Journal of the Royal Meteorological Society*, 140(680):1069–1085, 2014.

---

## Author Comment (AC4)

Dr. Kelly:

Thank you for your consideration of this manuscript. In separate letters, I have responded to the technical points and technical questions raised in RC1 and RC2. I like GMD and thus want to ask whether, after reading these letters, your opinion on the suitability of this article for possible publication in GMD has changed. If it has, I will prepare a new draft that is responsive to, in particular, the many excellent suggestions for clarification that reviewer 1 has provided. If it has not, then I will withdraw the article as you have suggested.

Thanks,
Andrew Bradley

---

## Author Comment (AC5)

Dr. Kelly and Reviewers:

Thank you for your consideration of this manuscript and reviews. After discussing details with Dr. Kelly, I will carry out the following plan:

1. Move sections 2 and 3 of the manuscript to another paper, and submit this new paper to another journal.

2. Revise sections 1, 4, and 5 according to the suggestions in the reviews, and submit the result as a revision to this manuscript.

Thanks,
Andrew Bradley

---

## Author Response (AR1)

Reviewers 1 and 2 and Associate Editor:

Thank you for your reviews. In previous responses, I addressed a number of questions and technical comments:

- to reviewer 1: https://doi.org/10.5194/gmd-2021-296-AC2;

- to reviewer 2: https://doi.org/10.5194/gmd-2021-296-AC3.

In this response accompanying the submission of a revised manuscript, I respond to points raised about the text and figures.

I carried out a major revision to the original manuscript. First, as suggested, I have divided the original manuscript into two. Most, but not all, of the material from Sects. 4 and 5 are in this revised manuscript. The material from Sects. 2 and 3 is used in another manuscript.

Second, to clarify material, I have added much more background information. In the new manuscript, Sect. 1 discusses very high-level matters. Section 2 provides details about all the algorithmic subcomponents that are then assembled into the overall method in Sect. 3. Each of the subsections of Sect. 2 has two parts: high-level explanations followed by mathematical details. In a first reading, the mathematical details can be skipped.

Third, Sect. 4 contains fewer figures and more comparisons to other methods in the text.

Fourth, Appendix A summarizes the Islet bases for completeness, while omitting almost all the details of their derivation; details are now in a second manuscript.

Fifth, in addition to removing material about the derivation of the Islet bases, I have also removed computer implementation details and performance figures; these are in the second manuscript. However, to provide clear information about the high performance of the Islet method, I have added new text regarding performance in the recently released E3SM version 2 early in the introduction (lines 18–26).

In the following, the reviewer's comments are italicized. Revised text from the manuscript is blue.

RC1: *The paper presents a new set of basis functions for semi-Lagrangian advection method in spectral element models. The presented method is novel and its performance in atmospheric test cases is encouraging. The paper therefore warrants to be published. The main drawback of the paper is that it is quite tedious to read. This paper essentially presents two things: (1) a novel optmized basis functions for 1D semi-Lagrangian advection schemes, and (2) its implementation in an atmo-spheric model using three different (sub-element) grids. Both of these are quite complex topics, and their discussion is intertwined (e.g. in section 2) and the reader is easily lost in details. The overall presentation should be improved before the paper can be accepted for publication. Considering the amount of work, I recommend a major review.*

Thanks. In reworking the presentation, I have tried to reduce the tedium by making each subsection in Sect. 2 have a high-level discussion followed by mathematical details that can be skipped. Section 3 now has no low-level details; instead, references to the mathematical details in subsections of Sect. 2 are provided.

RC1: *For clarity, I suggest that you clearly define the two considered problems, 1D SL advection method, and the implementation in 3-grid atmospheric models, already in the introduction. The introduction is now quite short and actually does not mention many important concepts relevant for the paper.*

Thanks. Sections 1 and 2 are now an extended introduction. In particular, both the 3-grid model and the advection method are discussed at length, first at a high level and referenced to

Figs. 1 and 2, then with mathematical details. Figure 2 is new and illustrates the interpolation SL method on spectral elements.

RC1: *Section 2 is rather difficult to follow, consider revising. I suggest to start by defining the 1D discretization with N elements, the interpolant functions within each element, and the properties of the interpolant functions (e.g. basis functions, continuity and symmetry). Presently, the interpolant functions first appear only in section 3.1. A figure could clarify the concepts, including the source and target elements/nodes. I would also define a symbol for the basis functions themselves, instead of using L from section 3.1 (L is the interpolant function itself). The discussion of stability becomes comprehensible only after the discretization has been introduced.*

Thanks. The stability material will appear in a second paper. I have attempted to build up the rest of the ideas piece by piece in Sect. 2. The new Fig. 2 illustrates target and source elements, GLL nodes on the dynamics and tracer grids, and 1D and 2D basis functions.

RC1: *As I understand, the 3 axioms of advection methods are: (1) global conservation, (2) preservation of constant tracers (sometimes called local conservation, or tracer consistency), and (3) monotonicity (i.e. no spurious overshoots appear). These properties apply to both the SL tracer advection scheme and the remap operators between grids. In section 1.2 these concepts seem to be mixed and referred to by different names (plausibly due to historical reasons; the so called "property preservation" is just a combination of 1, 2, and 3). Consider revising.*

Local mass conservation is not the same as mass-tracer consistency. A solution can be locally mass conserving but not mass-tracer consistent, as well as the opposite. Mass-tracer consistency means that the transport solver and the dynamics solver produce or use the same air density field $\rho$ to machine precision.

I have rewritten that section as follows:

(48–57) An approximate numerical solution for $q$ of Eq. 4 is said to be *property preserving* if (possibly just a subset of) properties that hold for the exact solution also hold for the approximate one. Equation 5 implies that advection cannot introduce new extrema in the mixing ratio; advection is said to be *shape preserving*. Equation 2 with $f = 0$ implies the global mass is conserved. Although the focus of this article is not the continuity equation, we note that the Lagrangian form of the continuity equation, Eq. B4 in Appendix B, implies that the total mass in a Lagrangian parcel, which is a parcel of fluid that moves with the flow, is constant. A final property that is a special case of the shape preserving property relates to coupling a solver for Eq. 4 to a dynamics solver: mass-tracer consistency. This property means that if $q$ is constant in space at time $t_0$, then it remains constant in space at every other time. In other words, the dynamics solver and transport solver use the same air density. The methods in this article conserve global mass, do not introduce new nodal extrema, and provide mass-tracer consistency when coupled to a dynamics solver.

Regarding local mass conservation, I now write:

(167–170) Local mass conservation means that one can identify numerical, possibly Lagrangian, fluid parcels on the grid that have constant tracer mass. *Local* is in contrast to *global* mass conservation; the latter means that the mass of the tracer fluid is conserved over the whole domain but not necessarily in any identifiable parcels smaller than the domain.

Finally, In the new Sect. 2.2, which I do not quote here because of its length, I write the specific property preservation problem the Islet method solves in detail, thus resolving any ambiguities that may arise from different terminology.

RC1: *Figures 20 and 21 shows larger diffusion for shorter time steps. Could you elaborate on this? Is the traditional SL property that longer time steps reduce diffusion (while the solution can degrade in other metrics)? How would one choose the right time step in practice? In the case of Eulerian transport it is easy: take the maximum stable one, and you are guaranteed to satisfy all the necessary properties.*

To address the first part of this question, I added the following.

(549–552) Essentially all SL methods, particularly when given exact trajectory data, exhibit
greater error with smaller time step, e.g., CSLAM in TR14. This is because the only source of
error, given exact trajectories, is the remap error. Smaller time steps correspond to more remaps
to reach a fixed simulation time.

I answered the second part of this question in the original response to reviewer 1. I have not
added material to the paper to answer this part because the answer is complex and somewhat
orthogonal to the primary material. All SL transport methods must contend with the matter of
time step. In the standard-resolution EAM version 2 configurations, the time step is limited by the
physics time step to six times the dynamics time step.

RC1: *Throughout the manuscript the authors use the terms "mixing ratio" and "tracer" for the*
*advected quantity $q_i$, seemingly interchangeably. For the sake of clarity I would prefer just to use*
*"tracer".*

Thanks. A tracer is a trace species, such as $CO_2$. A mixing ratio is a dimensionless quantity.
In meteorology the mixing ratio is typically the mass mixing ratio, e.g., grams of a species per
kilograms of wet air. Thus, a tracer is not the same as a mixing ratio; the tracer is the substance
whose quantity can be given by a mixing ratio or a density.

To address this point, I have added the following:

(33–34) The tracer transport equation in continuity form and with a source term for a tracer
mixing ratio $q$ and corresponding tracer density $\rho q$ is...

I have also been careful to use *tracer mixing ratio* ($q$), *tracer density* ($\rho q$) and *air density* ($\rho$)
consistently throughout the manuscript.

RC1: *l35: what is the definition of "local conservation" here?*

I have added the following text:

(167–170) Local mass conservation means that one can identify numerical, possibly Lagrangian,
fluid parcels on the grid that have constant tracer mass. *Local* is in contrast to *global* mass
conservation; the latter means that the mass of the tracer fluid is conserved over the whole domain
but not necessarily in any identifiable parcels smaller than the domain.

RC1: *l155: "This structure arises as follows. Consider a continuous discretization using a*
*nodal $n_p$-basis, $n_p = d + 1$, with $n_p$ the number of nodes. The grid has N elements. Each row of the*
*space-time matrix corresponds to a target node." This description is too brief to be understandable,*
*please elaborate. This is my interpretation of the discretization: The 1D domain is divided into N*
*elements. The solution in each element is approximated by a continuous function, defined by $n_p$*
*basis functions. Thus a function f in an element e can be written as $f_e(x) = \sum_{i=1}^{n_p} f_i \psi_i(x)$ where*
*$\psi_i$ and $f_i$ denote the i-th basis function and its corresponding coefficient. Each basis function is*
*associated with a node $x_i$ within the element; The basis is Lagrangian (a.k.a. nodal), i.e. $f_i(x_i) = 1$.*
*Furthermore, the discretization of the function is continous across element interfaces, implying that*
*the neighboring elements share a (exactly one?) basis function. Furthermore, the basis is assumed*
*to be symmetric about the center point of the element.*

This passage has been removed, as the topic will appear in a second paper. However, I have
added text following your suggestions in Sect. 2 to clarify relationships among elements, nodes, and
bases.

RC1: *l156: "Each row of the space-time matrix corresponds to a target node." You should*
*define the space-time matrix for this statement to be comprehensible.*

I have removed references to the space-time matrix in this manuscript, as the matrix is needed
only for stability analysis, which will appear in a companion paper.

RC1: *l177: Have you defined the basis to be symmetric somewhere?*

In the revised manuscript, I write that the basis is symmetric in Sect. 2.1.2 (background material) and Appendix A (precise description of the Islet bases):

(186–187) Each basis is a *nodal* basis: a basis function has value 1 at one node and 0 at every other node. Thus, each basis function is associated with a node. For example, in Fig. 2(b), the blue basis function is associated with the third node of six. The basis is symmetric; basis function $k \in \{0, \dots, n_p - 1\}$ is the mirror image of basis function $n_p - k - 1$. Thus, the blue and cyan functions are mirror images around reference coordinate 0.

(779–782) In addition, the basis is symmetric, meaning basis function $\phi_i^{n_p}(x) = \phi_{n_p-1-i}^{n_p}(-x)$. Thus, first, support nodes are specified for regions 0 through $\lfloor n_p/2 \rfloor - 1$, and the support nodes for the remaining regions are determined by symmetry. Second, if $n_p$ is even, then the middle region, $r = n_p/2 - 1$, has support nodes $\mathcal{I}_r^{n_p}$ that are symmetric around reference coordinate 0.

RC1: *l195: "L provides a basis for degree-d polynomials." I would say that the basis functions are the $\Pi_i$ functions defined in the equation of L; L itself is the interpolant function defined by the basis and the specific nodal values $y(i)$.*

Thanks. This material will appear in a companion paper.

RC1: *l195: "These are supported by $n = d + 1$ points, each an element in the n-vector xn." To be consistent with the literature I would use the term "node" instead of "point".*

In the revised manuscript, I use "node" when referring specifically to GLL nodes but continue to use "point" when referring to general grid points.

RC1: *l204: "Given a departure point x" These properties define the interpolant functions, thus there's no need to say that x is a departure point, it can be any point within the element.*

Thanks. This material will appear in a companion paper.

RC1: *l208: I think this constraint is equivalent to saying that the basis must Lagrangian or nodal?*

This material will appear in a companion paper. The basis need not be Lagrangian (e.g., it turns out to be piecewise polynomial), but it must admit a nodal representation.

RC1: *l246: only here you define a d-degree polynomial. This would be useful already in section 2.*

This material will appear in a companion paper.

RC1: *l257: what is Runge's phenomenon? what is Lebesque constant? help the reader to understand the rationale behind your work.*

This material will appear in a companion paper.

RC1: *Section 3.6: the description of the search algorithm is quite technical and could perhaps be moved to the appendix; it is not necessary to follow the main storyline of the paper.*

This material will appear in a companion paper.

RC1: *Section 4: mention TTPR already in the introduction as it seems to be relevant for the entire Islet method.*

Thanks. I have moved the definition to Sect. 1.3:

(104–106) We refer to this approach as *tracer transport p-refinement* (TTPR). In the finite element method, *p*-refinement means increasing the basis polynomial degree. In the Islet method, we increase $n_p^{\text{t}}$ relative to $n_p^{\text{v}}$ to represent the mixing ratio fields at higher resolution.

RC1: *l439: earlier tracer tendencies were denoted by $f_i \Delta t$*

Thanks. I have now written the defintion explicitly:

(410–412) Second, $\Delta\mathbf{q} \equiv \mathbf{f}\Delta t$ is remapped from the physics grid to the tracer grid, where $\Delta t$ is the physics parameterization time step. Either of $\mathbf{f}$ or $\Delta\mathbf{q}$ is sometimes called a *tendency*.

RC1: *l442: "immediate element neighbors" Are these neighbors that share an edge or vertex?*

Thanks. I now write:

(426–427) Let an element's neighborhood contain itself and every other element that shares a vertex with it.

RC1: *l464: "In contrast, …" Meaning unclear, please revise.*

Thanks. I have rewritten the DSS discussion and moved it to its own subsection, Sect. 2.4. Section 2 builds up notation and concepts step by step, so the new explanation of the DSS and the generalized DSS hopefully is clearer.

RC1: *l507: What is "tracer density"? Is it just $\rho$? Then, for clarity, I'd call it "density"*

$\rho$ is the air density, where "air" is the total content of the parcel. The tracer density is $\rho q$, where $q$ is the tracer mixing ratio. In a previous comment, I addressed how I have clarified these concepts in the text.

RC1: *section 4.3: while computational efficiency is important, I would move this section to the appendix, as it*

This material will appear in a companion paper.

RC1: *l570: "is proportional to the number of grid points" Should read: "proportional to the square of number of grid points"*

This material will appear in a companion paper. However, by "grid points" I mean general points in a grid, regardless of spatial dimension. In Sect. 2, I explain the tensor-product grid in greater detail, hopefully clarifying that essentially none of the new manuscript is concerned with 1D grids.

RC1: *l572: what is a naive h-halo exchange and how does it differ from what is proposed here?*

This material will appear in a companion paper, and I will explain the concepts in greater detail. "Naive" means deterministically exchanging data in all elements that *might* have a source-target relationship. Instead, in my implementation, I exchange only the data that are needed to determine the solution based on the flow-dependent domain of dependence in each time step, substantially reducing the communication volume. Thus, in general, the structure of the communicated data changes in each communication round.

RC1: *section5, l606: please mention the problem domain (full sphere?) and the equations that are being solved (pure advection on the tracer grid?)*

Thanks. I now write:

(502–503) Except in Sect. 4.3, the equation is the sourceless advection equation, Eq. 4. Two-dimensional, time-dependent flow, $\mathbf{u}(\mathbf{x}, t)$, is prescribed on the sphere.

RC1: *l631: why can you not use the dynamical core to compute the density? the chosen approach seems rather ad-hoc; can you guarantee that density values are realistic?*

Thanks. The new Appendix B derives the ISL discretization of the continuity equation, starting with the Reynolds transport theorem and then describing the details of the numerical quadrature, step by step, that lead to the discretization.

The experiments are run with a standalone program that is not part of the dynamical core. In addition, to run many long experiments, which are used in stability verification, I need a fast method to compute density; thus, an interpolation semi-Lagrangian method makes sense. Note that the air density is used for only property preservation calculations, so the details of its solution
are not impactful.

RC1: *scaling figures 6-8, 11-16: for easier readability do not use red line twice for np=6 and*
*np=12. x axis label is missing.*

Thanks. I have added the label and used a different color for each value of $n_p^{\mathrm{t}}$.

RC1: *l703: "has even more accuracy" Can you quantify this? Is the difference significant?*

Thanks. I have added a number of quantitative comparisons, as follows, where the first passage
addresses the particular line in your question.

(590–599) For example, the most accurate shape-preserving method in TR14 for the nondiver-
gent flow with Gaussian hills IC is HEL-ND-CN1.0, by a substantial margin (cyan curves in Fig. 1,
bottom right, of TR14). The $n_p^{\mathrm{t}} = 12$ Islet scheme with the long time step, Fig. 4, is approximately
three times more accurate than HEL-ND-CN1.0 in the $l_2$ norm at resolution $0.375°$ and approxi-
mately twice as accurate at resolution $3°$. Yet HEL-ND is, quoting TR14, an "unphysical" method.
It is run for comparison with the practically useful HEL scheme. After HEL-ND, the next most
accurate method in the $l_2$ norm at $0.375°$ resolution is CSLAM-CN5.0. The Islet method with the
long time step, the same as that of CSLAM-CN5.0, is at least as accurate for $n_p^{\mathrm{t}} \geq 8$. With the
short time step, the same as that of CSLAM-CN1.0, the Islet method is at least as accurate as
CSLAM-CN1.0 for $n_p^{\mathrm{t}} \geq 6$. At $3°$ resolution, no method other than HEL-ND-CN1.0 provides $l_2$
norm below $10^{-2}$; the Islet method does for $n_p^{\mathrm{t}} \geq 8$ with the long time step and $n_p^{\mathrm{t}} \geq 10$ (only
$n_p^{\mathrm{t}} = 12$ is shown) with the short time step.

(609–610) For example, with a long time step, for $n_p^{\mathrm{t}} = 8$, this value is a little coarser than $3°$;
for $n_p^{\mathrm{t}} = 12$, approximately $6°$. For comparison, no model in TR14 reports a value larger than $2.5°$.

(627–628) There is no summary number that can be compared directly with the results in Fig. 5
of TR14, but, visually, the curves for $n_p^{\mathrm{t}} \geq 8$, resolution $1.5°$, and on the tracer grid are among the
best of those in TR14.

(642–644) In Figs. 11–14 in TR14, the smallest value of $l_r$ at $1.5°$ among the property-preserving
methods is $2.15 \times 10^{-4}$, by the UCISOM-CN5.5 method, except for a value of 0 by HEL-ND, which,
again, cannot be used in practice. For the long time step, the Islet method gives at least as small
a value for $n_p^{\mathrm{t}} \geq 6$; for the short time step, $n_p^{\mathrm{t}} \geq 8$.

(648–656) This diagnostic is more difficult to compare than $l_r$ because very dissipative methods
tend to have a large value of $l_r$ and consequently a very small value for $l_u$. In contrast, a very
accurate method, for which $l_r$ is small, can have a larger $l_u$ value than a very dissipative method.
One means of comparison is to consider the best $l_u$ values among methods that obtain, say, $l_r \leq$
$5 \times 10^{-4}$. In Figs. 11–14 in TR14, the smallest value of $l_u$ at $1.5°$ under this restriction is 0, obtained
by the HEL-CN1.0 and HEL-CN5.5 methods. These HEL variants are practically usable, unlike
HEL-ND, and are designed to preserve tracer correlations exactly. Other than the HEL methods,
the next best value is $4.80 \times 10^{-5}$, again by the UCISOM-CN5.5 method. For both the long and
short time steps, at $1.5°$, the Islet method gives at least as small a value for $n_p^{\mathrm{t}} \geq 8$. However,
even with the constraint on $l_r$, comparison is not straightforward, as the UCISOM methods are not
strictly shape-preserving and so have $l_o > 0$.

(653–655) Although TR14 does not provide error norm values for this problem, those in Fig. 7
of TS12 can be compared with the Islet method's values at $1.5°$ resolution and the long time step
(left column of Fig. 14). The Islet method's values of $l_2$, $l_{\mathrm{inf}}$, $\phi_{\min}$, and $\phi_{\max}$ are at least as good
as those in Fig. 7 of TS12 for $n_p^{\mathrm{t}} \geq 6$.

RC1: *l711: "For each value ..." unclear sentence, please revise.*

I have revised this passage as follows:

(618–621) For each possible value $\tau$ of the tracer mixing ratio at the initial time, the area over which the mixing ratio is at least $\tau$ at the midpoint time is computed. For the cosine bells IC, $\tau \in [0.1, 1]$. The diagnostic is then this area divided by the correct area, which for nondivergent flow is the area at the initial time. The perfect diagnostic value is 100% for all $\tau \in [0.1, 1]$ and 0 otherwise.

RC1: *l741: 0 in m(0) stands for time t=0?*

Thanks. Yes. I have now made clear the dependence on space and time:

(670–674) Two tracer mixing ratios, a source $q_1 = s$ and a manufactured tracer $q_2 = m$, are paired. At time $t = 0$, $m(\mathbf{x}, t)$ is set to 0, where $\mathbf{x}$ is position on the sphere. A tendency $\Delta m$ is applied to $m$ on the physics grid: $\Delta m(\mathbf{x}, t) \equiv -[\cos(2\pi(t + \Delta t)/T) - \cos(2\pi t/T)]s(\mathbf{x}, t)/2$, so that the exact solution is $m(\mathbf{x}, t) = (1 - \cos(2\pi t/T))s(\mathbf{x}, t)/2$ and, in particular, $m(\mathbf{x}, T/2) = s(\mathbf{x}, T/2)$.

RC1: *l780: what is a terminator?*

I have added the definition to the text:

(684–686) The reactions are extremely sensitive to solar insolation. The sun's position is held fixed with respect to the grid. As a result, the largest-scale spatial pattern one sees in the fields is the boundary dividing nonzero (day) and zero (night) solar insolation, the solar *terminator*; this boundary is particularly visible in the right image of Fig. 17...

RC1: *l897: Here you define the basis functions $\phi_i$. This notation would be useful throughout the manuscript, already in Sect 2 and 3.1.*

Thanks. I now introduce the basis functions with the symbol $\phi$ in Sect. 2.

RC1: *abstract and intro: you define an abbreviation "dycore". I would omit it as it does not really save space, and it is not used frequently in the paper.*

Thanks; I have.

RC1: *l745: "shows the results"*

Thanks; fixed.

RC1: *l815: SYPD numbers printed above ...*

Thanks; fixed.

RC2: *This manuscript describes an interpolation-based semi-Lagrangian (SL) method for the transport problem on spectral-element (SE) domains. The SL transport schemes are widely used for multi-tracer transport in atmospheric models due to their accuracy and computational efficiency. The classical SL method employs interpolation at the upstream locations of the backward trajectories to estimate the advecting scalar values at the new time level. However, such an approach is not conservative per se, for practical applications an arbitrary procedure known as the "mass-fixing" usually employed for global conservation — which may have an adverse effect for climate-scale (long term) integration due to the local mass drifting. On the other hand, a finite-volume formulation of the SL method is conservative by design, where the upstream interpolation over the Lagrangian element is replaced by integration constrained to be locally (hence globally) mass conservative. The conservative data transfer from regular Eulerian grid to the deformed Lagrangian grid often referred to as the remapping (re-zoning), a limiter or shape-preserving scheme is usually employed for physically realizable solutions. A wide body of literature is available for both conservative and classical SL methods.*

Re: "'mass-fixing' ... may have an adverse effect for climate-scale (long term) integration due to the local mass drifting": I have attempted to clarify in a number of spots the relationship between local mass conservation and computational efficiency. In the introduction, I write:

(61–65) Another quality of a transport method is important: its computational efficiency. Computational efficiency is some measure of solution accuracy for a given set of computational resources.

Thus, when developing a new transport method, the objective is to obtain high efficiency, as measured by diagnostic values and computational cost, constrained by the need to couple to specific dynamics and physics grids. Our objective in this article is to extend our highly efficient tracer transport method for EAM version 2. . .

Then, later:

(173–175) Our objective in this work is to use the freedom provided by giving up local, but not global, mass conservation to maximize computational efficiency.

To address the viability of our method in long time integrations, I cite a model that uses one variant of it:

(22–24) For EAM version 2, we developed a new tracer transport method that is 6.5 to over 8 times faster than in EAM version 1, in the cases of, respectively, low and high workload per computer node [4] (Fig. 3).

RC2: *Implementation of conservative SL method on spherical domains tiled with high-order spectral-elements are very challenging. Authors have proposed an interpolation-based SL method Islet for the SE discretization. Instead of using the unstable native high-order interpolator (basis function) they have devised a cumbersome numerical procedure which employs an alternative grid system within each spectral element, adding another layer of complexity. The Islet method is not conservative, nevertheless, the global conservation is achieved by mass-fixing. The authors argue that the Islet method can handle tracer transport as well as the remapping between physics & dynamics grids, and incorporate shape-preservation filters.*

While the method is complex, EAM version 2 successfully uses two grids, incorporating the new physics grid described in this paper and [5]. I have added the following text to support this point, where the text refers to EAM version 2:

(24–26) In addition, we developed remap operators to remap data between separate grids for physics parameterizations and dynamics, permitting the physics parameterization computations to run on a coarser grid and thus 1.6 to 2.2 times faster in version 2 than in version 1 [5, 4].

To clarify that the new basis functions are not any more difficult to use than the natural basis functions, I added this text to the first paragraph describing the Islet basis functions in Appendix A:

(764–765) This article can be understood equally well by assuming the standard GLL bases are used; only the numerical results depend on the details of the basis functions.

RC2: *The manuscript is very long, the Islet interpolation as described by the authors is extremely complex. Authors failed to explain the core interpolation algorithm with clarity, there are many statements in the manuscript which leads to ambiguity. The numerical analysis part is very intense maybe more suitable for a computational math journal (e.g., SIAM / JCP) than the GMD. The subject covered could be split into a two-part paper, one describing the basic algorithm and analysis with more details and rigor, and the second part for implementation and validation with standard tests. This would be helpful for better reading. The current manuscript is written in an awkward manner and is unacceptable for publication.*

Thanks. I have split the original manuscript into two. To improve clarity, I wrote the new Sect. 2. Section 2 describes each algorithmic subcomponent, with the following outline corresponding to subsections:

- Semi-Lagrangian transport
    - Types of SL methods
    - Spectral element ISL transport

Each subsection provides a text overview followed by mathematical details. Section 3 then directly
references each subsection from Sect. 2 as it discusses each step of the Islet method.

RC2: *Recommendation: Major revision, possibly resubmit as a two-part manuscript. Authors*
*should address the following questions.*

I have separated the manuscript into two.

RC2: *(1) The stability associated with the SL method is that the deformational Courant number*
*(Lipschitz condition) should not exceed unity, in plain language, the trajectories should not cross*
*intersect (see, Staniforth & Cotes 1992 MWR paper). Is the cubic ISL method (lines 115-120)*
*unstable due to this condition? Need some explanation.*

I addressed this comment in my previous response to reviewer 2. This material is used in
another manuscript.

RC2: *(2) The SL transport scheme can be stabilized using a limiter, filter or with an explicit*
*diffusion (see, Ullrich & Norman, QJRMS, 2014). You can use the native high-order SE interpo-*
*lation (basis function) for the SL transport combined with the limiter which you are already using*
*for the Islet method. It will be interesting to see how the Islet method compares with this simple*
*SL-SE scheme employing 4x4 GLL grid (I guess that is the SE grid choice made for the operational*
*E3SM).*

I addressed this comment in my previous response to reviewer 2. Since the material on stability
now is used in a second manuscript, I have moved the figure and text demonstrating the effect of
instability on the method using "the native high-order SE interpolation (basis function) for the SL
transport combined with the limiter" to Appendix A3.

RC2: *(3) It is not convincing to have 3 grid systems (physics: FV, dynamics: GLL, transport:*
*tweaked GLL) in a SE modeling framework. The Fig.5 shows such a grid configuration, and it*
*appears to be very challenging. At a very high (NH) resolution the data movement is a major issue*
*for an element-based Galerkin model (DG/SE). A typical climate model may have O(100) tracers,*
*an additional tracer grid with more DOF than the dynamic grid can exacerbate this problem. This*
*will limit the use of Islet scheme, how do you address it?*

Thanks. I addressed this comment in my previous response to reviewer 2. I have also added
the following text to the introduction, where Fig. 3 in [4] is Fig. 1 in my response to reviewer 2
(https://doi.org/10.5194/gmd-2021-296-AC3):

(18–26) Because of the large number of tracers in climate models, tracer transport can be
computationally very expensive. For example, in the Dept. of Energy's Energy Exascale Earth
System Model (E3SM) [2] Atmosphere Model (EAM) version 1 [3], configured with the default 40
tracers, tracer transport takes approximately 75% of the total dynamical core wall clock time and approximately 23% of the total atmosphere model wall clock time on a typical computer cluster [4] (Fig. 3). For EAM version 2, we developed a new tracer transport method that is 6.5 to over 8 times faster than in EAM version 1, in the cases of, respectively, low and high workload per computer node [4] (Fig. 3). In addition, we developed remap operators to remap data between separate grids for physics parameterizations and dynamics, permitting the physics parameterization computations to run on a coarser grid and thus 1.6 to 2.2 times faster in version 2 than in version 1 [5, 4].

RC2: *(4) With real data you have velocity information only available at the GLL (dynamics) grid, the way you find the 2D trajectory information using the 3D Cartesian coordinates leads to additional computational overhead when the method is extended to the 3D application (line 470-475). This needs some justification, why not use the spherical (u,v) components or corresponding contravariant vectors?*

I addressed this comment in my previous response to reviewer 2.

RC2: *It is not clear that the maximum eigenvalue required for the interpolation is the tracer data dependent, in that case you have a serious computational overhead for the multi-tracer applications, Please clarify! What is the computational halo requirement for an SE stencil with NxN GLL points, when the shape preserving limiter is applied?*

I addressed this comment in my previous response to reviewer 2.

RC2: *(5) What is the special advantage of using Islet method? It seems you have introduced a complex numerical method for a relatively simple linear transport problem. If mass-fixing is the way to go, one could use the RBF-based (Kriging type) interpolator which provides very accurate solution, and no need for the expensive search for max eigenvalue etc.*

I addressed this comment in my previous response to reviewer 2.

RC2: *(6) The results are looking good, authors should limit the number of figures and make an effort to compare the results with that of other high-order element-based schemes. Why the results from your own previous papers (Bosler et. al. 2019, SIAM J Sci. Computing; Guba et al. 2014, JCP) discussed? These results should be compared and the relative merits should be discussed.*

Thanks. I have added a number of comparisons to the text, as documented above in a response to reviewer 1.

Re: Guba et al. 2014, JCP, results from that method appear in TR14 and so are included in the above comparisons. In addition, the performance of our ISL method is compared in lines 18–26 of the manuscript quoted above.

Re: Bosler et. al. 2019, SIAM J Sci. Computing, I have added the following text to make clear why we are pursuing an *interpolation* method instead of an *exactly cell-integrated* method:

(152–171) Interpolation is in contrast to *exactly cell-integrated* methods, which accurately integrate the basis of a target (e.g. Lagrangian) element against those of the source; see, e.g., [1]. (In some cases, an inaccurate cell-integrated method can be interpreted as an interpolation method; see Appendix B for an example.) Exactly cell-integrated methods have substantially greater cost than interpolation methods for three reasons.

First, to obtain smoothness in the integrand, integration is over facets computed by geometric intersection of a target element against source elements; intersection calculations are not needed in interpolation methods. Typically, to minimize computational geometry complexity, departure cell edges are approximated by great arcs rather than flow-distorted curves, limiting the method to second-order accuracy; however, [6] describe a higher-order edge reconstruction that yields a third-order accurate advection method. In contrast, achieving arbitrarily high order in an ISL method's linear advection operator does not entail any additional complexity.

Second, accurate integration has a larger computational cost because it requires sphere-to-reference point calculation and interpolant evaluations at many quadrature points.

Third, an exactly cell-integrated method requires a larger communication volume because all data from a source element are used to integrate against each target basis function.

In trade for these additional costs, exactly cell-integrated methods are locally mass conserving, and the fact that they are $L^2$ projections can be used to prove stability. Local mass conservation means that one can identify numerical, possibly Lagrangian, fluid parcels on the grid that have constant tracer mass. *Local* is in contrast to *global* mass conservation; the latter means that the mass of the tracer fluid is conserved over the whole domain but not necessarily in any identifiable parcels smaller than the domain. Although an exactly cell-integrated method is locally mass conserving, coupling it to a dynamics solver still generally requires additional measures to obtain mass-tracer consistency.

RC2: *(7) There are many undefined terms (e.g. CAAS) and notations which I am going to list, this should be fixed.*

Thanks. CAAS in particular is now explained in detail in Sect. 2.2.1. I have attempted to make the notation very clear in Sect. 2 to make Sect. 3 easier to read.

Thanks,
Andrew Bradley

**References**

[1] P. A. Bosler, A. M. Bradley, and M. A. Taylor. Conservative multimoment transport along characteristics for discontinuous galerkin methods. *SIAM J. on Sci. Comput.*, 41(4):B870–B902, 2019.

[2] E3SM Project. Energy Exascale Earth System Model (E3SM). [Computer Software] https://dx.doi.org/10.11578/E3SM/dc.20180418.36, April 2018.

[3] J.-C. Golaz, P. M. Caldwell, L. P. Van Roekel, M. R. Petersen, Q. Tang, J. D. Wolfe, G. Abeshu, V. Anantharaj, X. S. Asay-Davis, D. C. Bader, et al. The DOE E3SM coupled model version 1: Overview and evaluation at standard resolution. *J. Adv. Model Earth Sy.*, 11(7):2089–2129, 2019.

[4] J.-C. Golaz et al. The doe e3sm model version 2: Overview of the physical model. *in preparation for J. Adv. Model Earth Sy.*, 2022.

[5] W. M. Hannah, A. M. Bradley, O. Guba, Tang Q., J.-C. Golaz, and J. Wolfe. Separating physics and dynamics grids for improved computational efficiency in spectral element earth system models. *J. Adv. Model Earth Sy.*, 13(7):e2020MS002419, 2021.

[6] P. A. Ullrich, P. H. Lauritzen, and C. Jablonowski. Some considerations for high-order incremental remap-based transport schemes: edges, reconstructions, and area integration. *Int. J. for Numer. Meth. in Fl.*, 71(9):1131–1151, 2013.

---

## Author Response (AR2)

Dr. Kelly:

Thanks for your work as Topical Editor for this manuscript.

*1. In the abstract and the intro, please spell out the acronym Islet (Interpolation Semi-Lagrangian Element-based Transport).*

In the abstract (lines 6–8), the sentence now reads: "We describe a finite-element ISL transport method that we call the Interpolation Semi-Lagrangian Element-based Transport (Islet) method, such as for use with atmosphere models discretized using the spectral element method."

In the introduction (lines 28–29), we now write: "In this article, we describe the Interpolation Semi-Lagrangian Element-based Transport method, with the acronym stylized as "Islet" rather than "ISLET" to minimize distracting all-capitalized text."

*2. Section 2.1.2: Please update the reference for Bradley (2022) if you have submitted it for publication. GMD wants reference "Works cited in a published manuscript should be published already, accepted for publication, or available as a preprint with a DOI."*

Yes, that makes sense. I had hoped to have submitted the other paper by now, but another project has claimed my attention for the last several months. I have changed the text as follows to use "forthcoming article" as the means to refer to that work:

Lines 176–178: "We develop an ISL method that uses the *Islet bases*, summarized in Appendix A and derived in Bradley et al. (2021, Sect. 2 and 3) and a forthcoming article based on that material, to satisfy a necessary condition for stability."

Lines 766–767: "Bradley et al. (2021, Sect. 2 and 3) and a forthcoming article based on that material detail the derivation of the Islet bases, while this appendix summarizes the results of that derivation for completeness."

*3. Line 137: Replace "operators" with "operations".*

Thanks for catching that typo.

*4. Section 4: You refer to "validation problems". "Validation" generally refers to comparison with physical data/observations, while "verification" refers to testing the math/implementation for correctness. Hence, "verification problems" is a better term. Same comment in the Conclusions.*

I have replaced all uses of the word "verification" with "validation" and made equivalent replacements for other parts of speech.

*5. Line 920: If you have a DOI for this reference, please include it.*

Thanks. I have updated the reference to the ESSOAr preprint citation, with DOI.

Thanks,
Andrew Bradley